# Molecular and circuit mechanisms underlying avoidance of rapid cooling stimuli in *C. elegans*

Chenxi Lin[1,2,4], Yuxin Shan[1,2,4], Zhongyi Wang[1,2,4], Hui Peng[1], Rong Li[3], Pingzhou Wang[3], Junyan He[1], Weiwei Shen[1], Zhengxing Wu[3] & Min Guo[1,2]

The mechanisms by which animals respond to rapid changes in temperature are largely unknown. Here, we found that polymodal ASH sensory neurons mediate rapid cooling-evoked avoidance behavior within the physiological temperature range in *C. elegans*. ASH employs multiple parallel circuits that consist of stimulatory circuits (AIZ, RIA, AVA) and disinhibitory circuits (AIB, RIM) to respond to rapid cooling. In the stimulatory circuit, AIZ, which is activated by ASH, releases glutamate to act on both GLR-3 and GLR-6 receptors in RIA neurons to promote reversal, and ASH also directly or indirectly stimulates AVA to promote reversal. In the disinhibitory circuit, AIB is stimulated by ASH through the GLR-1 receptor, releasing glutamate to act on AVR-14 to suppress RIM activity. RIM, an inter/motor neuron, inhibits rapid cooling-evoked reversal, and the loop activities thus equally stimulate reversal. Our findings elucidate the molecular and circuit mechanisms underlying the acute temperature stimuli-evoked avoidance behavior.

The ability of the nervous system to trigger appropriate behavioral responses to cold stimuli is critical for animal survival and fitness. Animals have evolved complex and multifaceted thermosensory systems to respond to cold stimuli, especially acute cold stimuli, which may lead to tissue damage and even be life-threatening[1–3]. Although some cold sensors, such as the transient receptor potential (TRP) ion channels TRPM8[4,5] and TRPA1[6] and the glutamate receptor GluK2[7], have been identified in endotherms, such as mammals, their cold-elicited behavioral responses and the underlying neural circuits remain largely unexplored.

In ectotherms, *Caenorhabditis elegans* has an extremely attractive system that can be used to investigate the molecular and neural basis of temperature sensation due to its higher temperature sensitivity[8], and simple and compact neural circuit[9,10]. Previous studies have confirmed that worms employ distinct behavioral strategies to respond to temperature changes[11,12]. For example, as worms crawl on a spatial temperature gradient within an innocuous temperature range

(-13–26 °C), they utilize a sophisticated thermotaxis strategy driven by AFD, AWC and ASI neurons to navigate and accumulate around a preferred temperature, which depends on the past cultivation temperature (Tc)[8,13–16]. Measurements of the temperature-evoked activities of these neurons suggest that they are Tc-dependent responses[8,14,15]. In addition to thermotaxis behavior, worms also avoid rapid heating within or beyond an innocuous temperature range[17], and these rapid heating-evoked avoidance behaviors are similar to those evoked by other noxious agents[18]. Notably, the avoidance of rapid heating by worms depends not only on the absolute temperature, but also on the rate of heating. For example, heating at an absolute temperature of -33 °C using electronically heated metal wire[19], -31 °C using a thermal barrier[20], 20–32 °C using a Peltier element-based device[21], -33.5–38 °C[19,22] and 23–33 °C[23] using infrared laser could elicit rapid withdrawal behavior. Similarly, heating rates of 0.8–18 °C/s[24], 1.5–5.9 °C/s[25] and 1–5 °C/s[26] using infrared laser pulses could also evoke a rapid avoidance behavior that includes pause, reversal and turn. The

[1]College of Life Science and Technology, Huazhong Agricultural University, Wuhan, Hubei 430070, China. [2]College of Biomedicine and Health, Huazhong Agricultural University, Wuhan 430070, China. [3]Key Laboratory of Molecular Biophysics of Ministry of Education, Institute of Biophysics and Biochemistry, and Department of Biophysics and Molecular Physiology, College of Life Science and Technology, Huazhong University of Science and Technology, Wuhan 430074, China. [4]These authors contributed equally: Chenxi Lin, Yuxin Shan, Zhongyi Wang. ✉e-mail: minguo@mail.hzau.edu.cn

results from laser ablation and calcium imaging have shown that various sensory neurons, including AFD[22], FLP[22] and AWC[23] neurons in the head, PVD[25] neurons in the midbody and PHC[22] neurons in the tail, mediate heating-evoked avoidance behavior. These findings indicate that worms can respond to multimodal temperature stimulation.

In nature, how do worms respond to cold stimuli? In contrast to the many studies that have investigated thermotaxis and heat avoidance, few studies have examined cold-elicited behavioral responses in worms. Previous studies found that acute cold stimulation induced by changing the buffer temperature from 20 to 15 °C, triggers a robust increase in the frequency of omega turn[7,27]. Electrophysiologic and calcium imaging results have confirmed that the TRPA-1 in PVD neurons[27] and the kainate-type glutamate receptor GLR-3 in ASER neuron[7], regulate cold-evoked omega turn. However, the cold-elicited behaviors of a freely moving worm, which are similar to the heating-evoked behaviors in *C. elegans*, remain largely unknown.

To dissect the cold-elicited behavioral response of a freely moving worm, we developed an approximate linear cooling control device based on the semiconductor control principle (Fig. 1a). Because worms would be paralyzed in the presence of a cooling temperature below -13 °C, we tested the locomotor behavior of the worms using a temperature range from -14.5 °C to 25.5 °C. Using this device, we examined the locomotor activity of a freely behaving worm under stimulation with different cooling rates. By combining quantitative behavioral analyses, neuronal manipulation, genetic analysis and in vivo calcium imaging, we identified a pair of polymodal sensory ASH neurons that sense and mediate rapid cooling-evoked avoidance behavior within the physiological temperature range. By dissecting circuits, we uncovered multiple parallel circuits act downstream of ASHs to regulate rapid cooling-elicited avoidance behavior.

## Results

### ASHs are needed for cooling-elicited avoidance behavior

To quantify the strength of cooling stimuli, we stimulated worms using an approximate linear cooling rate generated by the semiconductor chilling plate controlled with a programmable power source (Fig. 1a). A cooling stimulus within the physiological temperature range (-14.5 °C–25.5 °C) was used to stimulate the worm only when it

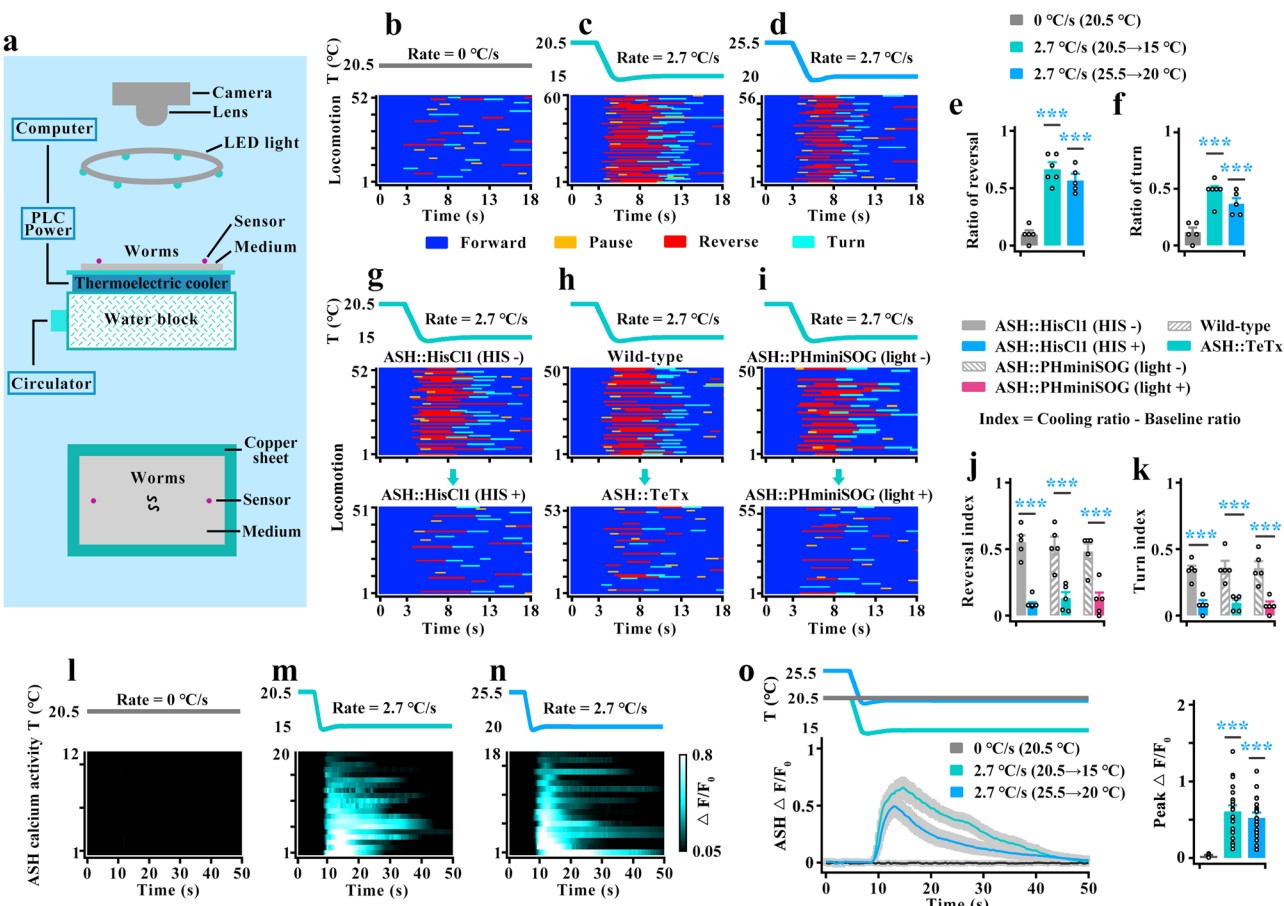

**Fig. 1 | ASH sensory neurons respond to rapid cooling. a** Schematic drawing of the rapid cooling system (RCS). Upper panel: whole RCS. Lower panel: Vertical view of RCS. **b**–**d** Locomotion behavior induced by cooling rates of 0 °C/s at 20.5 °C (**b**), 2.7 °C/s with range 20.5-15 °C (**c**), 2.7 °C/s with range 25.5-20 °C (**d**) in a forward-moving worm. In each figure, the top is temperature trace, the bottom is locomotion map. Forward (dark blue), pause (yellow), reverse (red) and turn (light blue). **e**, **f** Ratio of reversal (**e**) and turn (**f**) in the denoted worms. A cooling-evoked reversal or turn is defined by stopping forward movement and initiating a reversal with at least half a head swing or initiating a turn, respectively, within 3–8 s in the locomotion map. An omega turn following reversal without interruption was also defined as a cooling-evoked turn. n = 5, 6, 5 groups for each bar in (**e**, **f**) ≥10 worms/group. **g**–**i** Rapid cooling-evoked locomotion behaviors in ASH-silenced (**g**), ASH-blocked (**h**), and ASH-killed (**i**) worms. **j**, **k** Calculation of reversal (**j**) and turn (**k**) index for (**g**–**i**). The reversal or turn index was calculated by the ratio of reversal or turn elicited by cooling minus the ratio of baseline reversal or turn without cooling, respectively. The baseline reversal or turn were showed in Supplementary Fig. 3. n = 5 groups for each bar in (**j**, **k**) ≥10 worms/group. **l**–**o** ASH calcium transients induced by 0 °C/s at 20.5 °C (**l**), 2.7 °C/s with range 20.5–15 °C (**m**), 2.7 °C/s with range 25.5–20 °C (**n**). **l**–**n** Heatmaps. **o** Representative traces. In (**o**), the colored lines and the light gray area surrounding them indicate the mean values and SEM, respectively. n = 12, 20, 18 worms for each bar. Data are expressed as the mean ± SEM. Student's *t* test or Mann–Whitney rank sum tests (two-sided) in (**j**, **k**). One-way ANOVA followed by Dunnett's multiple comparisons in (**e**, **f**, **o**). ***p < 0.001.

exhibited forward locomotion, and the evoked locomotion behaviors consisting of forward, pause, reverse and turn were then recorded (Fig. 1b–d). Cooling-elicited reversal was defined as stopping forward movement and initiating a reversal with at least half a head swing within 5 s of stimulation initiation (duration of 3 to 8 s in the locomotion map), whereas a cooling-elicited turn was defined as any of the following: (1) stopping forward movement and initiating a turn within a 5-s period and (2) an omega turn following the reversal without interruption[16,21]. Consequently, we found that the wild-type worms cultivated at 20.5 °C showed cooling rate-dependent avoidance behavior within the range from 20.5 °C to 15 °C (Supplementary Fig. 1a–e, g–j, Fig. 1b, c, e, f, Supplementary movie 1, 2), suggesting that a faster cooling rate induced a stronger cooling-evoked avoidance response. To confirm whether the avoidance behavior is dependent on the cooling rate, we examined the locomotion behaviors evoked by different cooling ranges with the fastest rate of 2.7 °C/s. Worms cultivated at 20.5 °C also showed cooling-elicited avoidance behavior when stimulated by temperatures in the range from 25.5 °C to 20 °C (Fig. 1d–f), or from 20.5 °C to 17 °C (Supplementary Fig. 1f–j). These results indicate that worms respond to cooling rate and even avoid rapid cooling within the physiological temperature range.

To explore sensory neurons that are essential for rapid cooling-evoked avoidance behavior, we employed the *Drosophila* histamine-gated chloride channel (HisCl1) plus exogenous histamine to acutely silence amphid sensory neurons activities[28]. Due to the potential impact of the inhibition of neuronal activities on reversal or turn behavior, we recorded the locomotion behaviors of a worm initiating forward movement for at least 8 s without application of cooling stimulation. A reversal or turn that occurred within 3–8 s was considered a baseline reversal or baseline turn, and an omega turn following the baseline reversal without interruption was also defined a baseline turn. We then calculated the reversal or turn index as the ratio of reversal or turn elicited by cooling minus ratio of baseline reversal or turn without cooling, respectively. Based on this standard, worms in which ASH sensory neurons were silenced displayed severe defects in both cooling-evoked reversal and turn, whereas the silencing of other three sensory neurons AWB, AWC and ASJ, resulted in weaker defects in cooling-elicited reversal (Supplementary Fig. 2a–o, Supplementary Fig. 3a–d, Fig. 1g, j, k). To collect more evidences, we used two additional methods to inhibit neuronal activity. First, expression of the light chain of tetanus toxin (TeTx) blocks neurotransmission[29], and second, expression of pleckstrin homology domain membrane-targeted miniature singlet oxygen generator (PHminiSOG), which has been proven to have a higher cell killing efficiency than mitochondrial membrane miniSOG (mito-miniSOG), kills neurons with application of periodic blue light illumination[30]. Similarly, worms displayed severe defects in cooling-evoked avoidance behavior after inhibition of ASH activity using the abovementioned methods (Fig. 1h–k). These findings suggest that ASHs are essential for rapid cooling-elicited avoidance behavior.

We subsequently tested the ASHs calcium transients induced by rapid cooling using the genetically encoded calcium sensor GCaMP6f. The same temperature control protocol as that used for the abovementioned behavioral test was used. Notably, when worms were cultivated at 20.5 °C, ASH calcium transients could be elicited by different cooling rates within the temperature range from 20.5 °C to 15 °C (Supplementary Fig. 1k, Fig. 1l, m, o, Supplementary movie 3, 4). Furthermore, the magnitude of the calcium response depends on the cooling rate (Supplementary Fig. 1k). To further confirm that ASH is dependent on the cooling intensity, we examined various ASH calcium transients in different cooling ranges with the fastest rate of 2.7 °C/s. We found that ASH in worms cultivated at 20.5 °C also showed strong calcium transients in the temperature range from 25.5 °C to 20 °C (Fig. 1n, o) and from 20.5 °C to 17 °C (Supplementary Fig. 1l). Together, these results suggest that worms sense the cooling intensity and avoid

rapid cooling within the physiological temperature range. Sensory neuron ASHs respond to the cooling rate and mediate rapid cooling-evoked avoidance behavior.

## ASH senses the cooling intensity

Although ASH shows robust calcium transients in response to rapid cooling, and weaker calcium transients[14] and stochastic calcium events[23] in response to the slow temperature increases, ASH mainly acts as a polymodal nociceptive neuron to sense a variety of aversive stimuli[31–34]. Thus, whether ASHs respond to the cooling rate directly or indirectly stimulated by other temperature sensory neurons is unclear. ASH may be stimulated either directly or indirectly by numerous temperature sensory neurons, namely, AFD, ASER, ASG, AWC[ON], AWC[OFF], or both AWC[ON+OFF] etc. We first examined the ASH calcium transients in these neuron-inhibited worms. We found that the calcium activities of ASH did not change significantly after inhibition of these sensory neurons (Supplementary Fig. 4a–f, Supplementary Fig. 5a–f). In addition, we examined the ASH calcium transients in AIZ-inhibited worms, for which AIZ have been reported to play a dominant role in cryophilic behavior[12]. Similarly, we did not find a significant change in the ASH calcium transients after inhibition of AIZ neurons (Supplementary Fig. 6a, b). These results suggest that ASH neurons are not stimulated by the abovementioned potential temperature sensory neurons, and imply that ASH may directly respond to rapid cooling.

To confirm this speculation, we tested the ASH calcium transients in *unc-13*, *unc-31*, *unc-7*, and *unc-9* loss-of-function (LOF) mutants. The UNC-13 protein, encoded by the *unc-13* gene, is required for the release of small and clear synaptic vesicles that carry small molecular neurotransmitters[35]; the UNC-31 protein, encoded by the *unc-31* gene, is a critical component of the release machinery for neuropeptide-containing dense core vesicles[36]; and both the UNC-7 and UNC-9 innexins, encoded by the *unc-7* and *unc-9* genes, respectively, are important structural components of gap junctions that affect locomotion[37,38]. Therefore, these null mutant worms are considered to exhibit weak information flow among neurons. We found that the cooling-evoked calcium transients in these LOF worms were as strong as those in the wild-type worms (Fig. 2a–c), although all these worms displayed serious defects in cooling-evoked avoidance behavior (Fig. 2e, f, Supplementary Fig. 7a, b). In addition, we tested the ASH calcium transients in *osm-6* mutant worms, in which all sensory neurons were silenced due to defects in functional sensory cilia[39]. We found that OSM-6 is expressed in ASH neurons (Supplementary Fig. 7c), and the cooling-elicited calcium transients in ASH and the avoidance behaviors of worms showed severe defects in *osm-6* mutant worms (Fig. 2d–f, Supplementary Fig. 7a, b). Moreover, these defects in *osm-6* mutants could be partially rescued by specific expression of *osm-6* cDNA in ASH (Fig. 2d–f, Supplementary Fig. 7a, b). Together, these results suggest that ASH could sense and respond to rapid cooling stimulation directly.

## Both AIZ and AIB interneurons mediate cooling-evoked avoidance behavior

In *C. elegans*, interneurons receive most of the synaptic output from amphid sensory neurons and regulate a wide range of aversive stimuli that evoke avoidance behaviors[40–43]. According to the reconstructions of electron micrographs of the worms[9,10,40,44], the first layer has four interneurons, namely, AIA, AIB, AIY and AIZ (Fig. 3a). We first examined the locomotion behavior of interneuron-inhibited worms (Supplementary Fig. 8a, b), and found that the cooling-evoked reversal and turn were reduced in AIZ-inhibited or AIB-inhibited worms (Fig. 3b, c). Previous studies have shown that AIZ exhibits reversal-correlated calcium activity in both cell body and axon[42,45]. We then simultaneously recorded the calcium activity of the cell body and axon in the AIZ when these are in the same focal plane, and found that the magnitude of the calcium response in the cell body was stronger than that in the axon

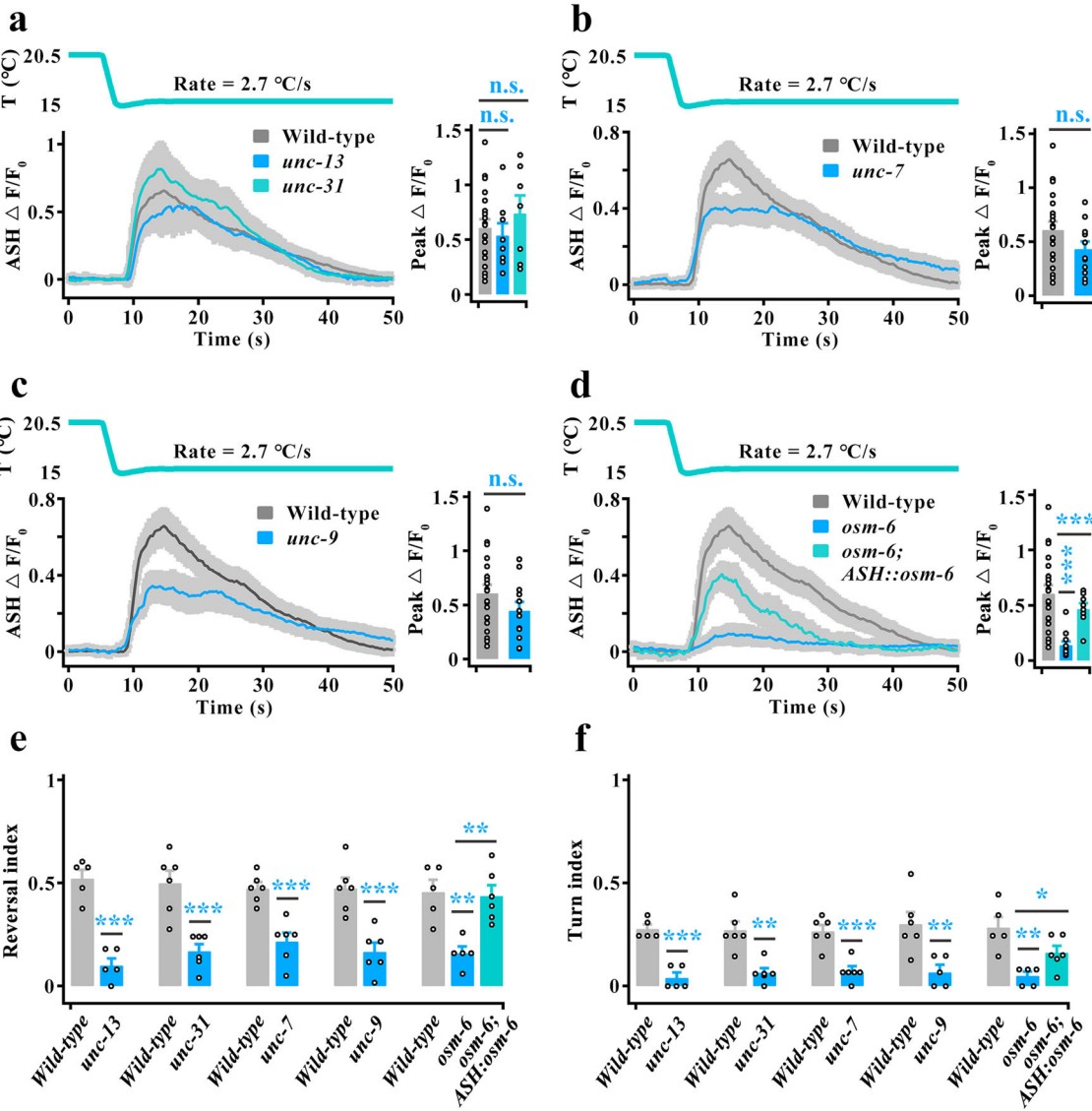

**Fig. 2 | ASH sensory neurons sense rapid cooling stimulation. a–d** ASH calcium transients induced by rapid cooling in chemical synapse loss-of-function (LOF) mutants of *unc-13* and *unc-31* (**a**), gap junction LOF mutants of *unc-7* (**b**) and *unc-9* (**c**), cilia LOF mutants of *osm-6*, and ASH-specific rescue of *osm-6* worms (**d**). $n = 20$, 8, 7 worms for each bar in (**a**). $n = 20$, 12 worms for each bar in (**b**). $n = 20$, 12 worms for each bar in (**c**). $n = 20$, 12, 8 worms for each bar in (**d**). **e, f** Calculation of reversal index (**e**) and turn index (**f**) elicited by rapid cooling in the worms denoted in (**a–d**). $n = 5, 5, 6, 6, 6, 6, 6, 6, 5, 5$ and 6 groups for each bar in (**e**, **f**) and ≥10 worms/group. Data are expressed as the mean ± SEM. Student's *t* test or Mann–Whitney rank sum tests (two-sided) in (**b**, **c**, **e**, and **f**). One-way ANOVA followed by Dunnett's multiple comparisons in (**a**, **d**, **e**, **f**). *$p < 0.05$, **$p < 0.01$, ***$p < 0.001$, $p > 0.05$ denotes not significant (n.s.).

during rapid cooling stimulation (Fig. 3d, Supplementary Movie 5). We thus used the AIZ cell body calcium transients for the following research. Moreover, we examined the calcium transients in the AIB cell body and found rapid cooling-evoked calcium activity (Fig. 3h). These behavioral and calcium results suggest that both AIZ and AIB mediate rapid cooling-evoked avoidance behavior.

To explore whether AIZ and AIB functionally act downstream of ASH to regulate rapid cooling stimulation, we examined their calcium activities after inhibition of ASH. Because only electrical synapses exist between ASH and AIZ (Fig. 3a), and the TeTx mainly blocks neurotransmission in chemical synapse. Here, we only used HisCl1 to acutely silence ASH or PHminiSOG to kill ASH (Supplementary Fig. 9a). We found that AIZ calcium activity was significantly reduced in ASH-inhibited worms (Fig. 3e), indicating that AIZ is stimulated by ASH. To further confirm this notion, we used a chemogenetic method to activate ASH but not cooling stimulation by expressing transient receptor potential vanilloid 1 (TRPV1), a mammalian cation channel that can be

activated by exogenous application of the ligand capsaicin. As previously reported[34,46], ASHs expression of mammalian TRPV1 could be activated by exogenous application of capsaicin (Supplementary Fig. 9b). Simultaneously, AIZ showed an obvious calcium spike after stimulation of ASH neurons (Fig. 3f). These data suggest that AIZ acts downstream of ASHs to regulate cooling-evoked avoidance behavior. To test whether gap junctions mediate signal transmission from ASH to AIZ, we recorded AIZ calcium transients in gap junction LOF mutant worms. We found that AIZ calcium activity was significantly decreased in both *unc-7* and *unc-9* null mutants (Fig. 3g). Furthermore, the calcium signal defects in *unc-7* or *unc-9* mutants could be partially rescued by specifically expressing the genomic DNA of *unc-7* or *unc-9* in both ASH and AIZ neurons (Supplementary Fig. 9c, Fig. 3g). These results suggest that AIZ could be stimulated by ASH via gap junctions during rapid cooling stimulation.

We then confirmed that the other interneuron, AIB, acts downstream of ASH to mediate cooling sensations using the same strategy

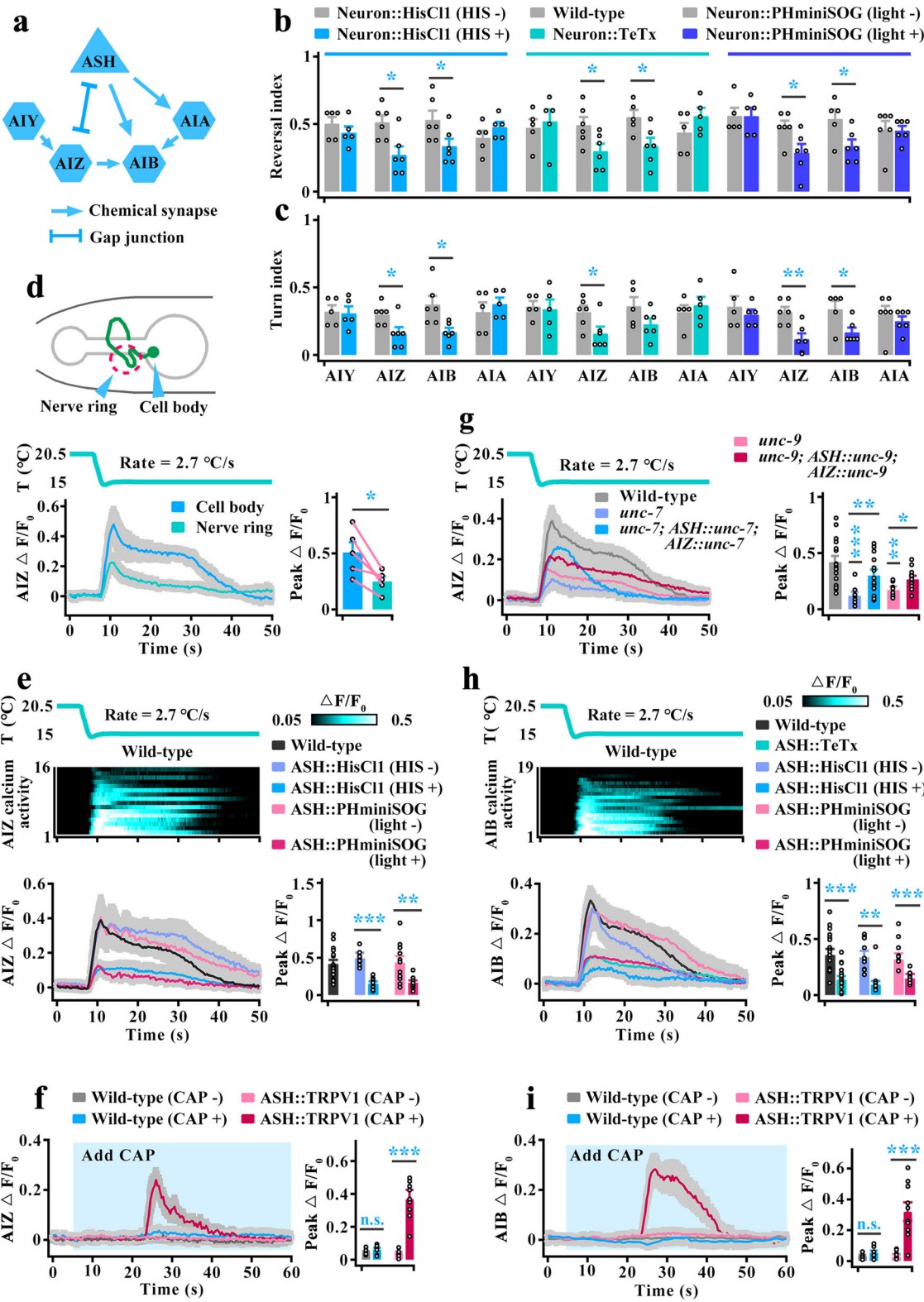

as AIZ neurons. Briefly, the cooling-evoked calcium transients in AIB were reduced after inhibition of ASH (Supplementary Fig. 9d, Fig. 3h), and AIB neurons could be excited by activation of ASH (Fig. 3i). Notably, AIZ neurons have strong chemical synapses onto AIB (Fig. 3a), implying that AIB may indirectly receive inputs from ASH via AIZ. We thus examined AIB calcium activity in AIZ-inhibited worms. We found AIB calcium transients were not significantly changed (Supplementary Fig. 9e), suggesting that AIB is mainly stimulated by ASH but not

indirectly via AIZ. Together, these results indicate that both AIZ and AIB interneurons act downstream of ASH to modulate rapid cooling stimulation.

## AIZ stimulates RIA to respond to rapid cooling

We next identified the neurons that receive inputs from AIZ to mediate cooling-evoked avoidance behavior. Based on the wiring diagram (Fig. 4a), SMB, RIA and RIM neurons are the primary downstream inter/

**Fig. 3 | Both AIZ and AIB interneurons act downstream of ASH to regulate rapid cooling-evoked avoidance behavior. a** Schematic of chemical and electrical synapses among ASH and first-layer interneurons (AIY, AIZ, AIB, and AIA). **b, c** Calculation of reversal index (**b**) and turn index (**c**) of worms in which the first-layer interneurons were inhibited. $n = 5, 5, 6, 6, 6, 6, 5, 5; 5, 5, 6, 6, 6, 6, 5, 5; 5, 5, 6, 6, 5, 6, 6$ and $6$ groups for each bar in (**b** or **c**) ≥10 worms/group. **d** Comparing cooling-evoked calcium transients between cell body and axon of wild-type AIZ interneurons. Upper panel: Schematic diagram of AIZ interneurons. Lower panel: Calcium transients in cell body and nerve ring. $n = 5$ assays. **e** Heatmaps depicting AIZ calcium transients induced by rapid cooling in wild-type worms (upper panel) and curve graphs depicting the cooling-evoked calcium activities of AIZ in ASH-inhibited worms (lower panel). $n = 16, 9, 11, 9$ and $11$ worms for each bar. **f** AIZ calcium transients following chemogenetic activation of ASH. ASH that specifically

expresses TRPV1 was activated by exogenous capsaicin (CAP 100 μM). Light blue shading denotes the period of CAP application. The same applies hereinafter in this manuscript. $n = 7, 8, 8$ and $9$ worms for each bar. **g** AIZ calcium transients elicited by rapid cooling in *unc-7*, *unc-9* mutants, and expression of *unc-7* or *unc-9* genomic DNA specific to both ASH and AIZ neurons genetically rescued worms. $n = 16, 9, 14, 8$ and $12$ worms for each bar. **h** Heatmaps depicting AIB calcium transients in wild-type worms (upper panel), and curve graphs depicting AIB calcium activities in ASH-inhibited worms (lower panel). $n = 19, 15, 11, 8, 8$ and $8$ worms for each bar. **i** AIB calcium transients following chemogenetic activation of ASHs. $n = 9, 8, 8$ and $10$ worms for each bar. Data are showed as the mean ± SEM. Student's *t* test or Mann–Whitney rank sum tests (two-sided) in (**b–i**). *$p < 0.05$, **$p < 0.01$, ***$p < 0.001$, $p > 0.05$ denotes not significant (n.s.).

motor neurons[40,44]. We thus tested these inter/motor neuron functions in cooling-evoked avoidance behavior (Supplementary Fig. 10a, b). Notably, we found an opposite behavioral response between RIA-inhibited and RIM-inhibited worms: RIA-inhibited worms showed obvious decreases in cooling-elicited reversal and turn, whereas RIM-inhibited worms displayed increases in cooling-elicited reversal (Fig. 4b, c). These results indicate that both may act downstream of AIZ to regulate cooling-evoked avoidance behavior.

To further confirm this speculation, we examined RIA and RIM calcium activities in AIZ-inhibited worms. RIA interneurons exhibit compartmentalized axonal calcium activity that integrates sensory input and motor feedback[47,48]. Thus, we tested the calcium activity in both the cell body and axon of RIA and found that both showed similar calcium activity during rapid cooling (Fig. 4d, Supplementary Movie 6). These results suggest that both the cell body and axon of RIA exhibit the same modulation in response to cooling stimulation, and consequently, the cell body calcium activity was used in subsequent research. Based on this standard, RIA calcium transients were significantly reduced after inhibition of AIZ neurons (Fig. 4e, Supplementary Fig. 10c). We also tested RIM calcium transients in AIZ-inhibited worms. Notably, RIM neuron in wild-type worms displays a probabilistic calcium response, which means that RIM could be excited or exhibit rest in response to the activation of ASH by rapid cooling (Supplementary Fig. 10d, Fig. 7d). We thus calculated all RIM calcium responses in ASH-excited worms, and found that the RIM calcium activities did not change after inhibition of AIZ (Supplementary Fig. 10d). To collect further evidence showing that RIA is stimulated by AIZ, we tested the RIA calcium signal by artificial activation of AIZ neurons. Because capsaicin is liposoluble and cannot act on non-cilia interneurons directly, we used optogenetic methods to stimulate AIZ by expressing channelrhodopsin-2 (ChR2) with 460-nm blue light illumination and used the genetically encoded $Ca^{2+}$ sensor R-GECO1 to measure RIA calcium activity. We found that RIA exhibited robust calcium activity upon artificial stimulation of AIZ neurons (Fig. 4f).

The above-described data showed that RIA was activated by AIZ and that AIZ was stimulated by ASH during rapid cooling. Therefore, RIA activity is modulated by ASH. We thus tested RIA calcium transients in ASH-inhibited worms and found that the calcium activities in RIA were significantly reduced during rapid cooling stimulation (Fig. 4g). Furthermore, RIA calcium activity could be stimulated under artificial stimulation of ASH (Supplementary Fig. 10e). Together, these results suggest that RIA interneurons, but not RIM, receive excitatory signals from AIZ during rapid cooling stimuli.

## GLR-3 and GLR-6 are essential for AIZ-mediated activation of RIA

We then explored the molecules involved in the stimulation of RIA by AIZ. Glutamate is broadly expressed in vertebrate and invertebrate nervous systems[49]. Glutamatergic signaling reportedly plays a critical role in the regulation of temperature sensation in *C. elegans*[7,50,51]. Therefore, we first tested the behavior and calcium activity of worms

with mutation in the *eat-4* gene, which encodes a vesicular glutamate transporter that concentrates glutamate into synaptic vesicles[52]. The cooling-evoked avoidance behavior and calcium activity of *eat-4* null mutants were clearly reduced, and their phenotypes were similar to the behaviors in RIA-silenced worms (Supplementary Fig. 11a, b, Fig. 5a, b), and calcium signals in AIZ-silenced worms (Fig. 5c). Furthermore, these defects in behavior and calcium signaling could be rescued by expression of *eat-4* cDNA driven by its own promoter (Supplementary Fig. 11a, b, Fig. 5a–c). These data suggest that glutamatergic signaling modulates cooling-evoked avoidance behavior. However, because *eat-4* exhibits a wide range of expression patterns in the nervous system, despite its expression in AIZ neurons[53] (Supplementary Fig. 12a), we are not entirely sure that this glutamatergic signaling in the modulation of RIA originated from AIZ. Therefore, a neuron-specific RNAi method was used to specifically knock down *eat-4* expression in AIZ neurons[54]. RNAi of *eat-4* in AIZ in wild-type worms generated obvious decreases in avoidance behavior and RIA calcium activity, which exhibited similar phenotypes to those in the *eat-4* mutant background (Supplementary Fig. 11a, b, Fig. 5a–c). Together, these results indicate that glutamate released from AIZ mediates the stimulation of RIA during rapid cooling.

We next screened candidate glutamate receptors that mediate the stimulation of RIA by AIZ, including the cation-selective ionotropic and the metabotropic glutamate receptors[55]. We found that the cooling-evoked avoidance behaviors in *glr-1*, *glr-3* and *glr-6* mutants were similar to those of RIA-silenced worms, as revealed by a significant decrease (Supplementary Fig. 12b–e). However, only the defects in *glr-3* and *glr-6* mutants could be rescued by respective expression of *glr-3* or *glr-6* cDNA in specific RIA neurons (Fig. 5d, e), whereas the behavioral defects in *glr-1* mutants could not be rescued by expression of *glr-1* cDNA in RIA neurons (Supplementary Fig. 12d, e). Calcium activity in RIA was hardly changed in the *glr-1* mutant (Supplementary Fig. 12f), but exhibited significant declines in both *glr-3* and *glr-6* mutants (Fig. 5f, g). Furthermore, the defects of calcium activity in *glr-3* or *glr-6* mutants could be recovered by respective expression of *glr-3* or *glr-6* cDNA driven by an RIA-specific promoter (Fig. 5f, g). These results suggest that both GLR-3 and GLR-6 are essential for the activation of RIA.

However, the defects in cooling-evoked reversal (Fig. 5d) and calcium activities in single *glr-3* or *glr-6* mutants (Fig. 5f, g) appeared to be weaker than the those in the behavior in RIA-silenced worms and the calcium signals in AIZ-silenced worms, respectively. We thus constructed a *glr-3;glr-6* double mutant worm. Notably, the cooling-evoked reversal and calcium activities in *glr-3;glr-6* double mutant worms showed more serious defects, as revealed by behaviors similar to those of RIA-silenced worms (Fig. 5d), and the calcium signals were similar to those of AIZ-silenced worms (Fig. 5h). These results suggest that GLR-3 and GLR-6 synergistically mediate RIA activity in response to rapid cooling. Together, our findings reveal an excitatory circuit that forms with ASH, AIZ and RIA neurons during rapid cooling stimulation (Fig. 5i).

## GLR-1 mediates ASH stimulation of AIB

AIB interneurons also regulate cooling-evoked avoidance behavior in the above-described data (Fig. 3). Thus, we explored the molecules that transduce the stimulation of AIB by ASH. The ionotropic glutamate receptor GLR-1 has been reported to mediate various aversive stimuli in AIB[16,41,43,45]. In addition, the cooling-evoked avoidance behaviors of *glr-1* mutants displayed a significant decrease, but could not rescue the defects in RIA neurons (Supplementary Fig. 12d, e). Therefore, we rechecked whether these behavioral defects in *glr-1* null mutants could be rescued in specific AIB neurons. Notably, the cooling-evoked behavioral defects in *glr-1* mutants could be rescued by expression of *glr-1* cDNA driven by its own promoter or an AIB-specific promoter (Supplementary Fig. 13a, b, Fig. 6a, b). Simultaneously, the calcium activities of AIB in *glr-1* null mutants display a significant decrease, which were similar to those in ASH-silenced worms (Fig. 6d). Furthermore, those calcium signal defects in *glr-1* mutants could be recovered by expression of *glr-1* cDNA driven by its own promoter or an AIB-specific promoter (Fig. 6c, d). We also tested AIB calcium transients and reversal initiation in artificially activated of ASH worms. Compared with wild-type worms, AIB calcium activity (Fig. 6e) and the reversal initiation (Fig. 6f) in *glr-1* mutants were significantly reduced after artificial stimulation of ASH. These results suggest that GLR-1 mediates the activation of AIB by ASH during cooling stimulation.

We next tested whether glutamate, the ligand of GLR-1, is essential for the stimulation of AIB. We found the calcium activities of AIB in *eat-4* null mutants were significantly decreased during cooling stimulation (Fig. 6g). Furthermore, these defects could be rescued by expression of *eat-4* cDNA driven by its own promoter or an ASH-specific promoter (Fig. 6g). These results suggest that glutamate, which is released from ASH, acts on GLR-1 in AIB to respond to cooling stimulation (Fig. 6h).

## RIM receives signals from AIB to respond to rapid cooling

We then explored which downstream neurons receive signals flowing from AIB. Based on the wiring diagram, the second-layer RIB and RIM interneurons are the primary downstream interneurons to which AIB sends synaptic outputs (Fig. 7a). The RIB-inhibited worms showed no obvious change in rapid cooling-evoked avoidance behaviors (Supplementary Fig. 14a, b, Fig. 7b, c), but the RIM-inhibited worms displayed a significant increase in cooling-evoked reversal and a weak increase in cooling-elicited turn (Figs. 4b, c and 7b, c), indicating that RIM inhibits cooling-evoked avoidance behavior. Thus, to identify

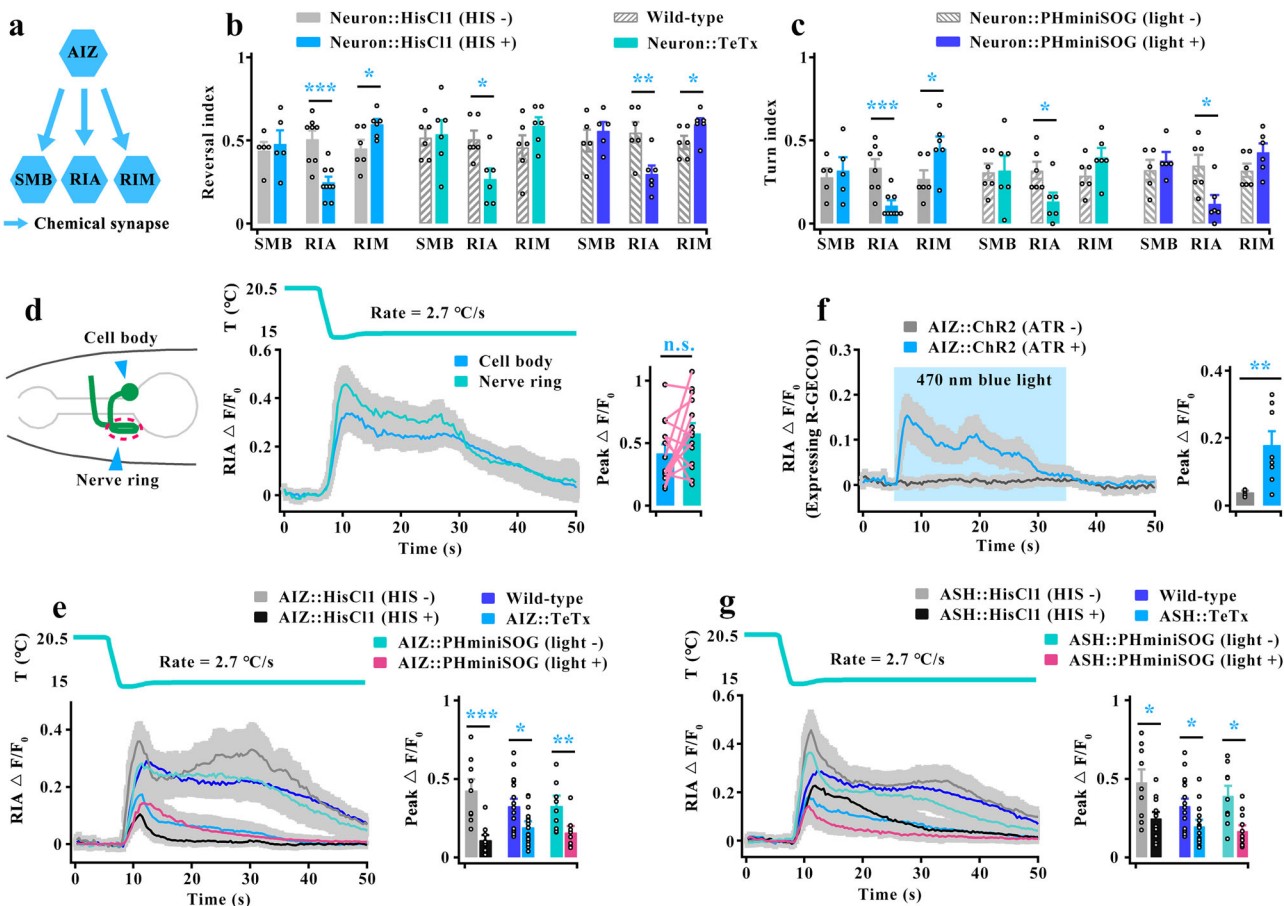

**Fig. 4 | AIZ transduces a cooling signal by stimulating its postsynaptic neuron RIA. a** Schematic showing the major downstream synaptic partners of AIZ. **b, c** Quantification of reversal index (**b**) and turn index (**c**) in silencing of postsynaptic neurons of AIZ. $n = 5, 5, 8, 9, 6, 6, 6, 6, 6, 6, 6, 6, 5, 5, 6, 6$ and 6 groups for each bar in (**b** or **c**) ≥10 worms/group. **d** Schematic diagram of the nerve ring and cell body of RIA interneurons (left panel), and the cooling-evoked calcium transients in the cell body and axon of RIA interneurons in wild-type worms (right panel). $n = 14$ assays. **e** RIA calcium transients evoked by rapid cooling stimuli in AIZ-silenced, AIZ-blocked and AIZ-killed worms. $n = 9, 10, 16, 16, 9$ and 8 worms for each bar. **f** RIA calcium transients following optogenetic activation of AIZ. RIA is expressed by the genetically encoded Ca$^{2+}$ sensor R-GECO1. AIZ that specifically expressed ChR2 was activated by 460-nm blue light illumination plus All-Trans-Retinal (ATR) at a final concentration of 5 μM. Light blue shading denotes the period of blue light illumination. The same applies hereinafter in this manuscript. $n = 6, 8$ worms for each bar. **g** RIA calcium transients elicited by rapid cooling stimuli in ASH-silenced, ASH-blocked and ASH-killed worms. $n = 9, 11, 16, 16, 9$ and 11 worms for each bar. Data are showed as mean ± SEM. Student's *t test* or Mann–Whitney rank sum tests (two-sided) in (**b**–**g**). *$p < 0.05$, **$p < 0.01$, ***$p < 0.001$, $p > 0.05$ denotes not significant (n.s.).

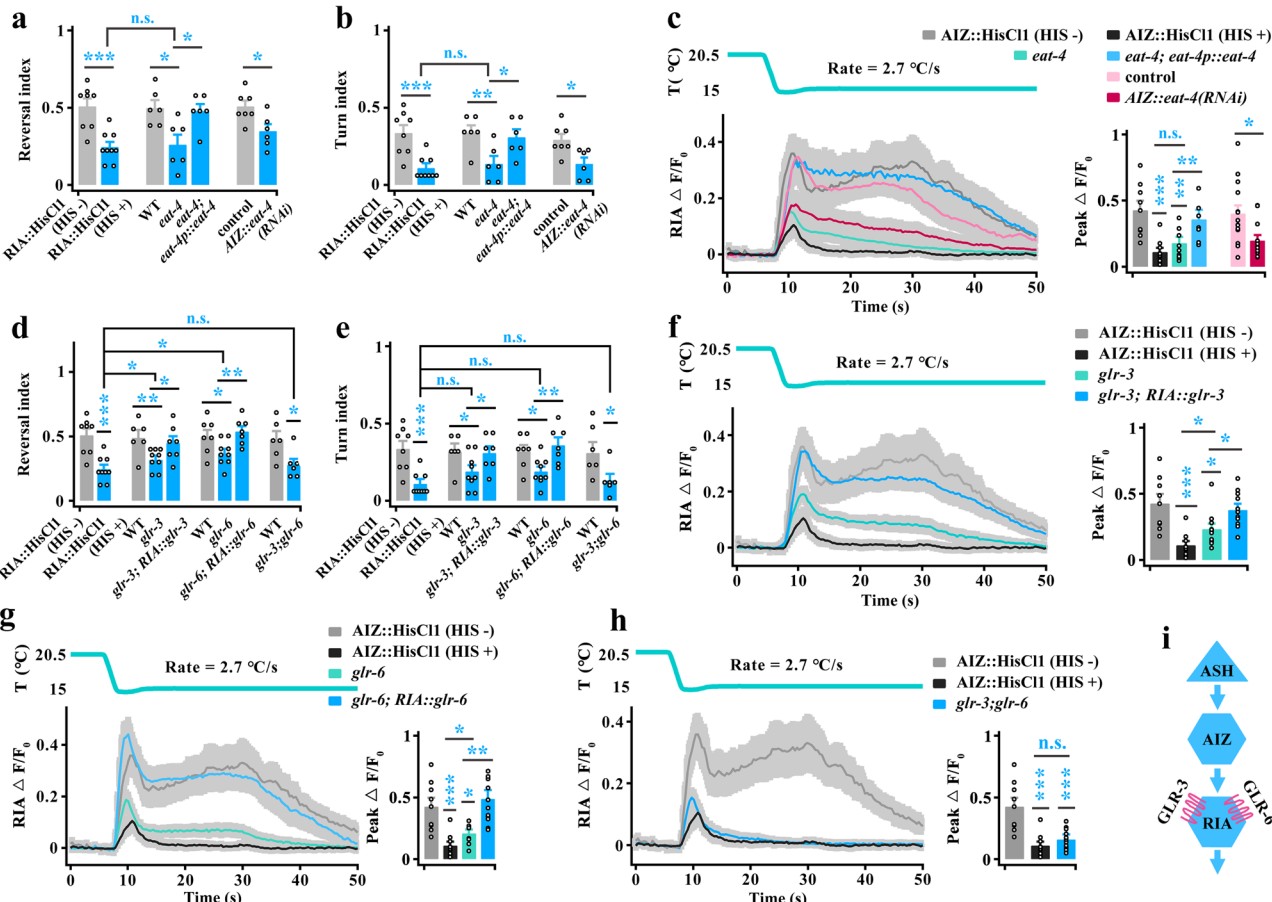

**Fig. 5 | Both glutamate receptors of GLR-3 and GLR-6 in RIA are needed for cooling-evoked avoidance behavior. a, b** Calculation of reversal (**a**) and turn (**b**) index elicited by rapid cooling stimulation in wild-type, RIA-silenced, *eat-4* mutant, *eat-4; eat-4p::eat-4* genetically rescued and AIZ::*eat-4(RNAi)* worms. The expression of AIZ::GCaMP6f in wild-type worms was used as the control of AIZ::*eat-4(RNAi)*, and the same applies in figure (**c**). *n* = 8, 9, 6, 6, 6, 7 and 6 groups for each bar in (**a** or **b**) ≥10 worms/group. **c** RIA calcium transients elicited by rapid cooling in the worms denoted in (**a, b**). *n* = 9, 10, 9, 8, 15 and 12 worms for each bar. **d, e** Calculation of reversal (**d**) and turn (**e**) index of *glr-3* and *glr-6* mutants, *glr-3;glr-6* double mutants, and RIA::*glr-3* and RIA::*glr-6* genetically rescued worms. *n* = 8, 9, 6, 9, 7, 7, 9, 7, 6 and

6 groups for each bar in (**d** or **e**) ≥10 worms/group. **f–h** Rapid cooling-evoked RIA calcium transients in AIZ-silenced, *glr-3* mutant and RIA::*glr-3* genetically rescued worms (**f**, *n* = 9, 10, 11 and 11), *glr-6* mutant and RIA::*glr-6* genetically rescued worms (**g**, *n* = 9, 10, 9 and 12), and *glr-3;glr-6* double mutant worms (**h**, *n* = 9, 10 and 14). **i** Schematic model showing an excitatory circuit composed of ASH, AIZ and RIA neurons in response to rapid cooling stimulation. Data are showed as mean ± SEM. Student's *t* test or Mann–Whitney rank sum tests (two-sided) in (**a–e**). One-way ANOVA followed by Dunnett's multiple comparisons test in (**a, b, d–h**). Kruskal–Wallis test with Dunnett's multiple comparisons in (**c**). *$p < 0.05$, **$p < 0.01$, ***$p < 0.001$, $p > 0.05$ denotes not significant (n.s.).

whether RIM is modulated by AIB during rapid cooling stimulation, we silenced both AIB and RIM neurons and found that the cooling-evoked avoidance behaviors were similar to those of AIB-silenced worms (Supplementary Fig. 14a, b, Fig. 7b, c). These results suggest that RIM acts downstream of AIB and may be inhibited by AIB during rapid cooling stimulation, which is consistent with the multimode regulation between AIB and RIM described in previous studies[41,43,56].

We then tested RIM calcium transients in AIB-inhibited worms. Notably, RIM in wild-type worms displays a probabilistic calcium response (approximately 40%) during rapid cooling stimulation (Supplementary Fig. 10d), in which a calcium response event in RIM depends on the excitation of ASH (Fig. 7d). We found that cooling-evoked calcium activities of RIM were significantly increased in AIB-inhibited worms (Fig. 7e–g), indicating that RIM is suppressed by AIB during rapid cooling stimulation. We also examined RIM calcium activity in AIB-activated worms, and found the calcium activity of RIM was reduced after stimulation of AIB by both the optogenetic method and rapid cooling (Fig. 7h, i), even if AIB calcium activity was strengthened (Supplementary Fig. 14c). These data obtained after inhibition of AIB and stimulation of AIB suggest that RIM is suppressed by AIB during rapid cooling.

The above-described data showed that AIB is stimulated by ASH (Fig. 3h, i), and that AIB inhibits RIM. Thus, RIM should be modulated by ASH. Using the same manipulation methods as those used for AIB, we found that RIM calcium activities were increased after inhibition of ASH (Fig. 7j–l), and reduced after stimulation of ASH (Supplementary Fig. 14d, Fig. 7m, n). Together, these data reveal a circuit consisting of ASH, AIB and RIM neurons during rapid cooling stimulation. In this circuit, RIM suppresses avoidance behavior. AIB stimulated by ASH could inhibit RIM activity, which is equivalent to impairing the inhibitory effect of RIM, and ultimately promotes worms to avoid rapid cooling. This modulation circuit is formed by AIB and RIM and can thus be considered as the disinhibitory circuit[41].

### The glutamate-gated chloride channel AVR-14 mediates inhibition of RIM by AIB

We then explored the molecular mechanism underlying the AIB-mediated inhibition of RIM. Previous studies have shown that AIB releases glutamate to act on both excitatory and inhibitory glutamate receptors on RIM to regulate aversion-evoked reversal[41,57,58]. We thus first screened glutamate-gated chloride channel mutant worms

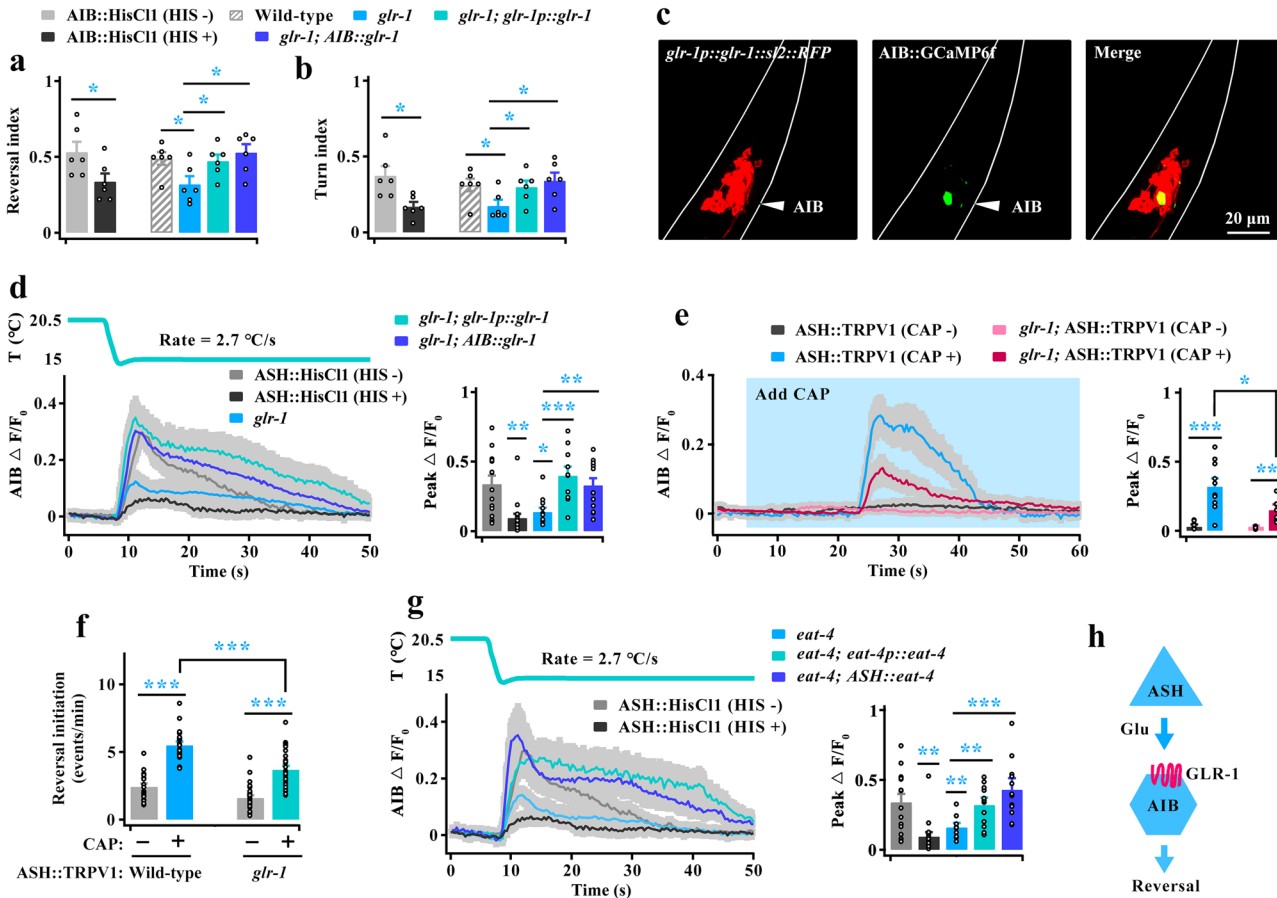

**Fig. 6 | The glutamate receptor GLR-1 on AIB mediates excitatory input from ASH. a, b** Calculation of reversal (**a**) and turn (**b**) index induced by rapid cooling in wild-type, AIB-silenced, *glr-1* mutant, *glr-1; glr-1p::glr-1* and *glr-1; AIB::glr-1* genetically rescued worms. *n* = 6 groups in (**a** or **b**) ≥10 worms/group. **c** Expression pattern of *glr-1p::glr-1::sl2::RFP* and AIB::GCaMP6f in *glr-1* mutants. The results show that GLR-1 is expressed in AIB neurons. Scale bar, 20 μm. **d** AIB calcium transients induced by rapid cooling stimuli in the worms denoted in (**a**). *n* = 11, 8, 12, 10 and 13 worms for each bar. **e, f** Comparison of AIB calcium transients (**e**, *n* = 8, 10, 5 and 7) and reversal initiation in free locomotion (**f**, *n* = 18, 19, 26 and 30) between wild-type

and *glr-1* mutant worms after chemogenetic activation of ASH neurons. **g** AIB calcium transients evoked by rapid cooling in ASH-silenced, *eat-4* mutant, *eat-4; eat-4p::eat-4* and *eat-4; ASH::eat-4* genetically rescued worms. *n* = 11, 8, 10, 12 and 12 worms for each bar. **h** Schematic model showing that ASH stimulates AIB neurons by releasing glutamate to act on GLR-1 in AIB. Data are showed as mean ± SEM. Student's *t* test or Mann–Whitney rank sum tests (two-sided) in (**a, b, e, f**). One-way ANOVA test followed by Dunnett's multiple comparisons in (**a, b, d, g**). *$p < 0.05$, **$p < 0.01$, ***$p < 0.001$.

because these channels usually have an inhibitory function on neuronal activity[59]. The cooling-evoked reversal (Supplementary Fig. 15a, b), and calcium activity in RIM (Supplementary Fig. 15c) were significantly increased in both *avr-14* and *avr-15* null mutant worms, and their phenotypes were similar to the behaviors in RIM-silenced and calcium activity in AIB-silenced worms respectively. Moreover, these defects in *avr-14* mutants could be rescued by expression of *avr-14* cDNA driven by its own promoter or an RIM-specific promoter (Fig. 8a–d), but not rescued in *avr-15* mutants by expression of *avr-15* cDNA in RIM-specific neurons (Supplementary Fig. 15a–c) because the AVR-15 was not expressed in RIM neurons (Supplementary Fig. 15d). These findings suggest that the glutamate receptor AVR-14 on RIM regulates cooling-evoked avoidance behavior.

We subsequently examined whether glutamate, the ligand of AVR-14, is essential for the AIB-mediated inhibition of RIM. Briefly, the cooling-evoked calcium activities of RIM in *eat-4* mutants were increased (Fig. 8e, f), and these defects in *eat-4* mutants could be rescued by expression of *eat-4* cDNA driven by its own promoter (Fig. 8e, f). In addition, the calcium activity of RIM and the cooling-evoked reversal were increased in AIB-specific knockdown *eat-4* worms, which were similar to the calcium signals in AIB-silenced (Fig. 8e, f) and behaviors in RIM-silenced worms (Fig. 8g), respectively. Together, these results indicate that glutamate released from AIB

mediates the inhibition of RIM by acting on its AVR-14 receptor (Fig. 8h).

## Role of command interneurons in response to rapid cooling

The above-described data showed that two circuits, the stimulatory circuit (AIZ, RIA) (Figs. 3 and 5) and the disinhibitory circuit (AIB, RIM) (Figs. 7 and 8), act downstream of ASH to regulate cooling-evoked avoidance behavior. We thus tested the avoidance behavior in which both types of circuits were inhibited worms, and found that cooling-evoked reversal and turn were significantly reduced (Supplementary Fig. 16a, b, Fig. 9a, b). Notably, fewer defects in cooling-evoked reversal were found in both RIA- and AIB-silenced worms compared with ASH-silenced worms (Fig. 9a). In addition, the abovementioned data showed that both GLR-3 and GLR-6 in RIA neurons (Fig. 5) and GLR-1 in AIB neurons (Fig. 6), modulate cooling-evoked avoidance behavior. Therefore, we constructed a *glr-1;glr-3;glr-6* triple mutant and tested its avoidance behavior, and found the cooling-evoked reversal and turn were also significantly decreased in the triple mutant worms (Supplementary Fig. 16a, b, Fig. 9a, b). Similarly, fewer defects in cooling-evoked reversal were observed in the triple mutants compared with the ASH-silenced worms (Fig. 9a). Thus, these behavioral differences between ASH-silenced and (RIA + AIB)-silenced worms, or between ASH-silenced and *glr-1;glr-3;glr-6* triple mutant worms suggest that the

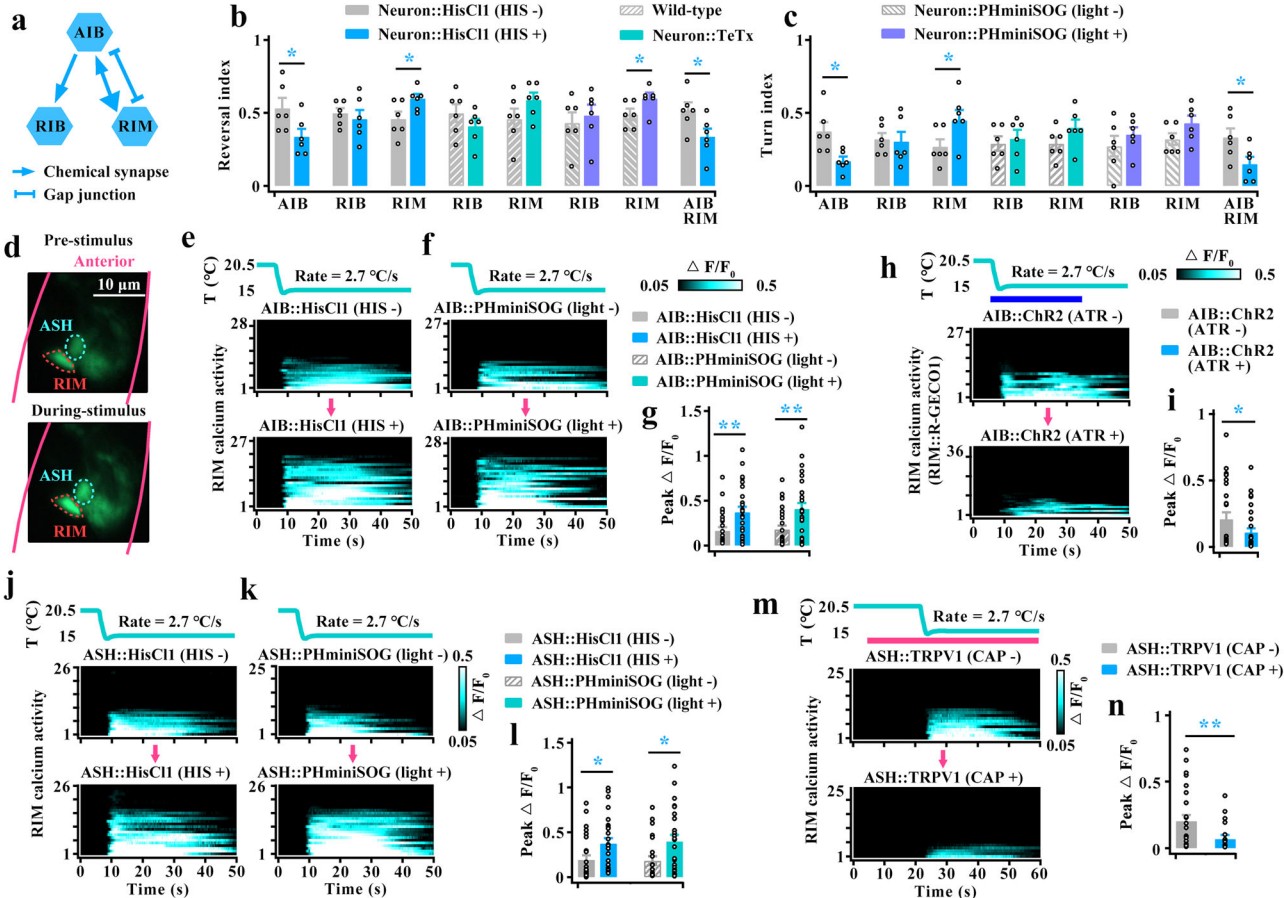

**Fig. 7 | AIB promotes cooling-evoked avoidance behavior by inhibiting RIM neurons. a** Schematic showing the major synaptic partners of AIB neurons. **b**, **c** Calculation of the cooling-evoked reversal (**b**) and turn (**c**) index of neuron-inhibited worms. $n = 6$ groups for each bar in (**b** or **c**) ≥ 10 worms/group. **d** Criteria for determining the RIM calcium response during rapid cooling. The upper panel is the pre-cooling and lower panel is the during cooling stimulation. **e–g** Heatmaps of the RIM calcium transients induced by rapid cooling stimuli in AIB-silenced (**e**) and AIB-killed (**f**) worms. **g** Comparison of the calcium activity in the worms denoted in (**e–f**). $n = 28, 27, 27$ and $28$ worms for each bar. **h** Heatmaps of the RIM calcium response to a rapid cooling stimulus in optogenetics activation of AIB worms. AIB that specifically expressed ChR2 was activated by blue light illumination. The dark

blue line indicates the period of blue light illumination. **i** Comparison of the peak of calcium signal change in the worms denoted in (**h**). $n = 27$ and $36$ worms for each bar. **j**, **k** Heatmaps of cooling-evoked calcium activities in RIM neurons in ASH-silenced (**j**) and ASH-killed (**k**) worms. **l** Comparison of the calcium activity in the worms denoted in (**j**, **k**). $n = 26, 26, 25$ and $26$ worms for each bar. **m** Heatmaps of RIM calcium response to rapid cooling in chemogenetic activation of ASH worms. For both rapid cooling and capsaicin stimulation, capsaicin was delivered to the worm nose 17 s before cooling. The red line indicates the period of capsaicin application. **n** Comparison of calcium activity in the worms denoted in (**m**). $n = 25$ worms. Data are showed as mean ± SEM. Student's *t* test or Mann–Whitney rank sum tests (two-sided) in this figure. *$p < 0.05$, **$p < 0.01$.

potential existence of other parallel circuits that regulate rapid cooling stimulation.

Based on the wiring diagram and previous reports, a group of command interneurons (AVA, AVD, and AVE) act downstream of ASH or first- and second-layer interneurons to initiate aversive-elicited reversal[41,60–62]. Thus, we tested the avoidance behavior of these command interneuron-inhibited worms, and found that cooling-elicited reversal and turn were significantly impaired when these command interneurons were inhibited (Supplementary Fig. 16a, b, Fig. 9c, d). Calcium imaging showed that the calcium activities of these command interneurons were reduced to a certain degree after inhibition of ASH neurons (Fig. 9e–h). To identify whether these command interneurons synergistically modulate avoidance behavior, we compared the avoidance behaviors of AVA-inhibited worms and (AVA + AVE + AVD)-inhibited worms. In particular, the avoidance behaviors of AVA-inhibited worms were not significantly different from those of (AVA + AVE + AVD)-inhibited worms (Fig. 9c, d), indicating that AVA is the dominant neuron in response to rapid cooling stimulation among command interneurons.

In addition, we examined other interneurons calcium activities, including the RIC and RMG, which have direct synaptic connections

with ASH. The calcium activities of these neurons were not changed in ASH-inhibited worms, indicating that these neurons are not involved in rapid cooling-evoked avoidance behavior (Supplementary Fig. 16c–f).

## Multiple parallel circuits regulate cooling-evoked avoidance behavior

Based on the neural connectome[9,10,44], the command AVA interneurons receive inputs from upstream interneurons and directly from sensory neurons. Thus, we tested the AVA calcium transients in interneuron-silenced worms. We found that cooling-evoked calcium activity in AVA was reduced after inhibition of AIZ, or AIB, or both of them (Fig. 10a, b), but not changed in RIA- or RIM-silenced worms (Supplementary Fig. 17a, b). Furthermore, AVA still showed obvious calcium activity in these interneuron-inhibited worms, suggesting that AVA also receives excitatory signals from AIZ and AIB in addition to ASH during rapid cooling.

The abovementioned data showed that multiple circuits, including stimulatory circuits (AIZ, RIA, AVA) and disinhibitory circuits (AIB, RIM), act downstream of ASH to modulate rapid cooling stimulation.

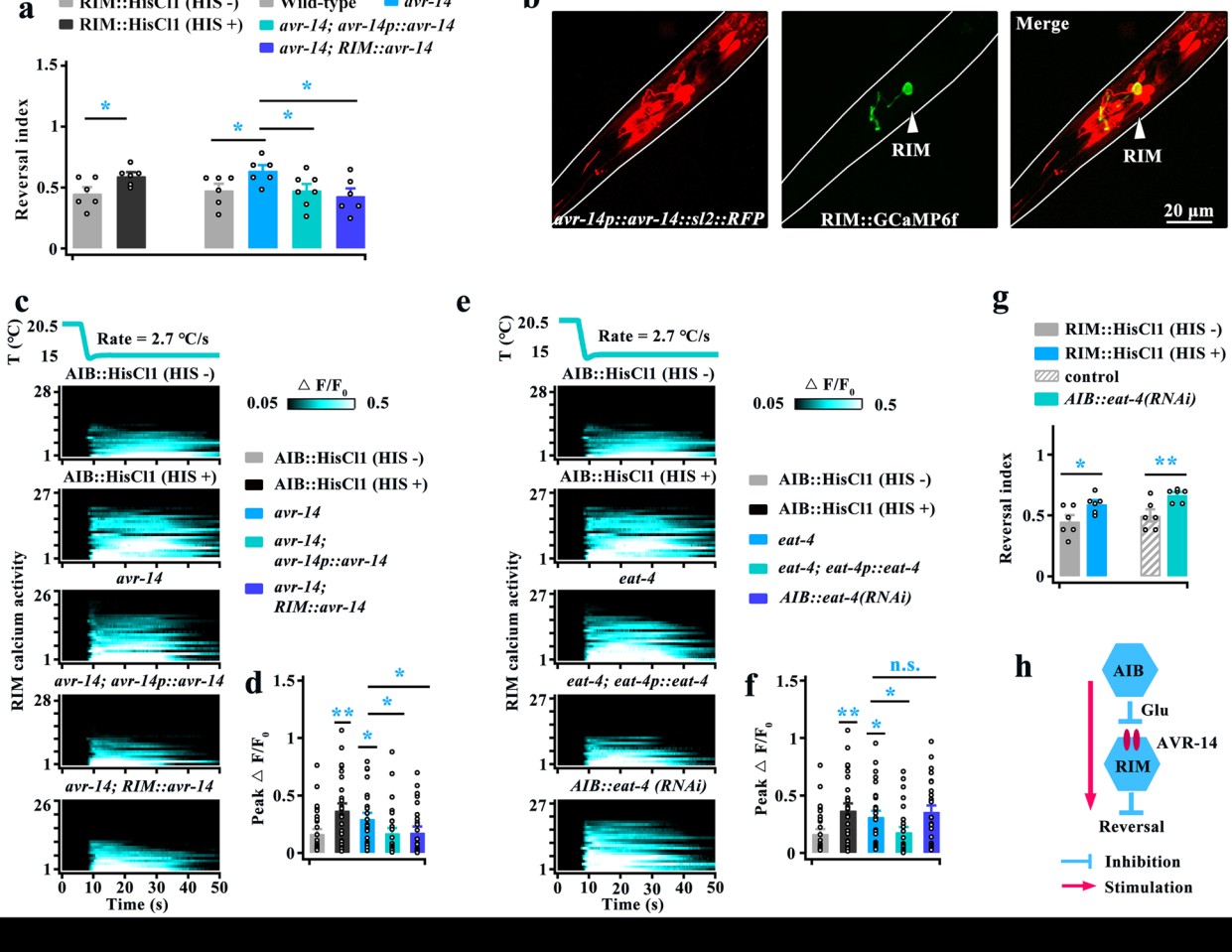

**Fig. 8 | The glutamate-gated Cl⁻ channel AVR-14 transduces the AIB-mediated inhibition of RIM. a** Comparison of the reversal index induced by rapid cooling in wild-type, RIM-silenced, *avr-14* mutant, *avr-14; avr-14p::avr-14* and *avr-14; RIM::avr-14* genetically rescued worms. $n = 6, 6, 6, 6, 7$ and 6 groups for each bar, ≥10 worms/group. **b** Expression pattern of *avr-14p::avr-14::sl2::RFP* and RIM::GCaMP6f in the *avr-14* mutant. AVR-14 is expressed in RIM. Scale bar, 20 μm. **c, d** Heatmaps depicting RIM calcium transients induced by rapid cooling in AIB-silenced, *avr-14* mutant, *avr-14; avr-14p::avr-14* and *avr-14; RIM::avr-14* genetically rescued worms (left panel), and comparison of calcium activity in the worms denoted (right panel). $n = 28, 27, 26, 28$ and 26 worms for each bar. **e, f** Heatmaps depicting the cooling-evoked calcium activities of RIM neurons in AIB-silenced, *eat-4* mutant, *eat-4; eat-*

*4p::eat-4* genetically rescued worms and AIB::*eat-4*(*RNAi*) worms (left panel), and comparison of calcium activity in the worms denoted in (**e**) (right panel). $n = 28, 27, 27, 27$ and 27 worms for each bar. **g** Comparison of the cooling-evoked reversal index between RIM-silenced and AIB::*eat-4*(RNAi) worms. $n = 6$ groups, and ≥10 worms/group. **h** Schematic model showing the AIB-mediated inhibition of RIM via AVR-14 during rapid cooling stimulation. Data are showed as mean ± SEM. Student's *t test* or Mann–Whitney rank sum tests (two-sided) in (**a, g**). One-way ANOVA test followed by Dunnett's multiple comparisons in (**a, d**). Kruskal–Wallis test with Dunnett's multiple comparisons in (**f**). *$p < 0.05$, **$p < 0.01$, $p > 0.05$ denotes not significant (n.s.).

We thus examined the avoidance behaviors of all these circuit-inhibited worms and found that (RIA + AIB + AVA)-silenced worms exhibited significantly decreased in rapid cooling-evoked avoidance behavior, similar to ASH-silenced worms (Supplementary Fig. 17c, d, Fig. 10c, d). These results suggest that these circuits synergistically respond to rapid cooling stimulation. Together, the results from our study show sophisticated circuits underlying cooling-evoked avoidance behavior, as displayed in Fig. 10e.

## Discussions

Ambient temperature is an unavoidable stimulus that animals are constantly confronted with. Thus, accurately sensing and responding to temperature changes is important for animal survival[2,63–66]. Ectotherms, particularly in *C. elegans*, employ multiple behavioral strategies and neural circuits to respond to temperature changes[11,12,17,25,26,67]. However, how worms respond to rapid temperature changes, especially rapid cooling stimulation, remains unknown.

In this study, we found that *C. elegans* responds to the cooling rate and avoids a rapid cooling stimulus (2.7 °C/s) within the physiological temperature range from 15 °C to 25 °C. ASH sensory neurons sense cooling rate and regulate rapid cooling-evoked avoidance behavior. These findings suggest that rapid cooling can be encoded as an aversive stimulus, even within the physiological temperature range. ASH neurons employ three parallel pathways, which can be divided into stimulatory and disinhibitory circuits, to synergistically respond to rapid cooling stimulation (Fig. 10e).

One of the stimulatory circuits is formed by ASH, AIZ and RIA. In this circuit, ASH activates AIZ via gap junctions to release glutamate, which acts on both GLR-3 and GLR-6 in RIA neurons to promote backward movement (Fig. 10e). The identified interneurons (AIZ, RIA) in this circuit were previously confirmed to act downstream of AFD and AWC sensory neurons to regulate cryophilic behavior[12,14,68]. These results indicate that worms also use the thermosensory circuit to respond to rapid cooling. However, there are some differences

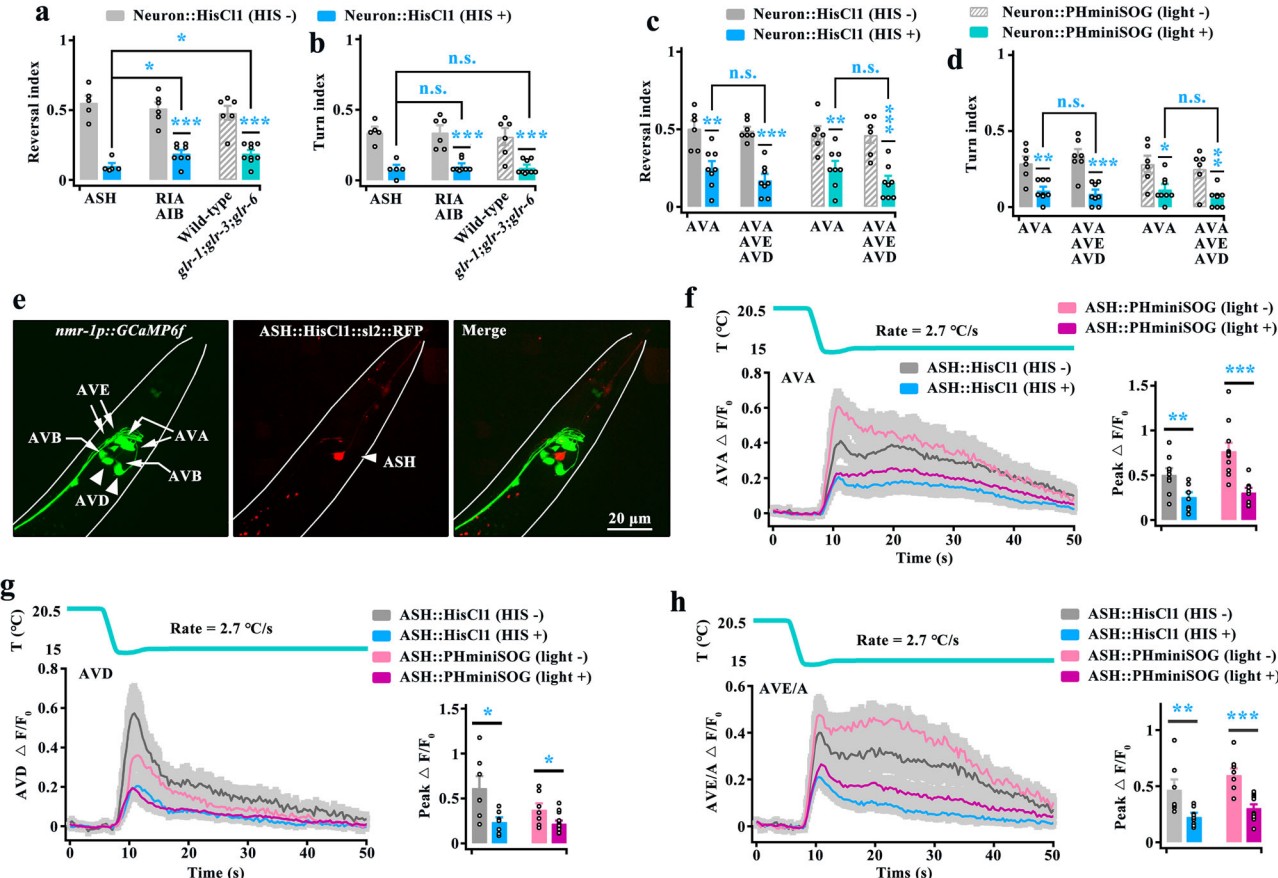

**Fig. 9 | Command interneurons regulate rapid cooling-evoked avoidance behavior. a, b** Comparison of cooling-evoked reversal and turn index among ASH-silenced, both RIA- and AIB-silenced, and *glr-1;glr-3;glr-6* mutant worms. *n* = 5, 5, 6, 8, 6 and 9 groups for each bar, ≥10 worms/group. **c, d** Comparison of cooling-evoked reversal and turn index between AVA-silenced and (AVA + AVE + AVD)-silenced worms. *n* = 6, 8, 7, 8, 6, 8, 6 and 8 groups for each bar, ≥10 worms/group. **e** Expression patterns of *nmr-1p*::GCaMP6f and ASH::HisCl1::*sl2::RFP* in wild-type worms. **f–h** Calcium transients of the command interneurons AVA (**f**, *n* = 9, 8, 11 and 8), AVD (**g**, *n* = 7, 7, 7 and 14) and AVE/A (**h**, *n* = 7, 9, 8 and 13) elicited by rapid cooling in ASH-inhibited worms. Data are showed as mean ± SEM. Student's *t* test or Mann–Whitney rank sum test (two-sided) in this figure. *$p < 0.05$, **$p < 0.01$, ***$p < 0.001$, $p > 0.05$ denotes not significant (n.s.).

between the two circuits: (1) In the cryophilic circuit, AIZ indirectly receives AFD or AWC sensory neuron signal inputs through AIY interneurons to regulate cryophilic behavior, whereas in our circuit, AIZ directly receives signals from ASH sensory neurons, which seemingly bypass AIY, to modulate cooling-evoked avoidance behavior. (2) In the cryophilic circuit, both AIZ and RIA neurons do not show positive calcium activities during cryophilic behavior. However, in our circuit, both AIZ and RIA neurons displayed obvious calcium transients in response to rapid cooling. We speculate that these differences occur because the intensity of rapid cooling is stronger than that of slow cooling in cryophilic behavior, which reaches the nociceptive stimulus threshold to trigger the avoidance response of worms.

The other stimulatory circuit is formed by ASH and a group of command interneurons (AVA, AVD and AVE) (Figs. 9 and 10e). In this circuit, the dominant AVA command interneurons receive inputs from ASH and from the first interneurons (AIZ and AIB) to respond to rapid cooling stimulation. Worms also use a disinhibitory circuit that consists of ASH, AIB and RIM to respond to rapid cooling, and this network can be depicted as ASH → AIB ⊣ RIM ⊣ reversal (Fig. 10e). According to the two inhibitions that exhibit stimulation logic, this circuit, which is equivalent to ASH stimulates AIB to promote the reversal. Notably, multiple escape circuits are used to respond to rapid cooling within the physiological temperature range, which further confirms that rapid cooling is encoded as a noxious signal. Worms live within the innocuous temperature range of ~13–26 °C, and the temperature below ~13 °C leads to worm paralysis, and are thus considered harmful

stimuli. In the natural world, when worms are exposed to rapid cooling stimulation, the temperature easily reaches a harmful temperature. We thus speculate that to survive, worm employs multiple escape circuits in response to rapid cooling.

Previous studies have confirmed that the interneurons of AIB, RIM and AVA have highly spontaneous calcium activity without odor stimulation in olfactory circuit[56]. However, in our study, these interneurons showed almost no spontaneous calcium activity without cooling stimulation (Supplementary Fig. 18a–c). Due to these interneurons receive the inputs from either AWC or ASH[9,10,40], and AWC neurons showed a highly spontaneous calcium activities without odor stimulation[56], whereas ASH displayed no calcium response without cooling stimulation (Fig. 1l, Supplementary Fig. 18d). We thus speculate that these differences occur because the different of spontaneous calcium activities between the sensory neuron ASH and AWC. RIA interneurons exhibit axonal calcium dynamics that integrate sensory input and motor feedback[47,48]. In our study, we found that the cell body of RIA has the same calcium activity as the axon in response to rapid cooling (Fig. 4d, Supplementary Movie 6). These results indicate that the cell body of RIA also modulates rapid cooling stimulation.

Temperature sensor is a hot research area. In this study, we found that ASH neurons sense rapid cooling directly. In addition, we have tested ASH calcium activity during rapid heating and found obvious heating-evoked calcium transients (Supplementary Fig. 18f), which is consistent with previous reports that ASH can respond to slow

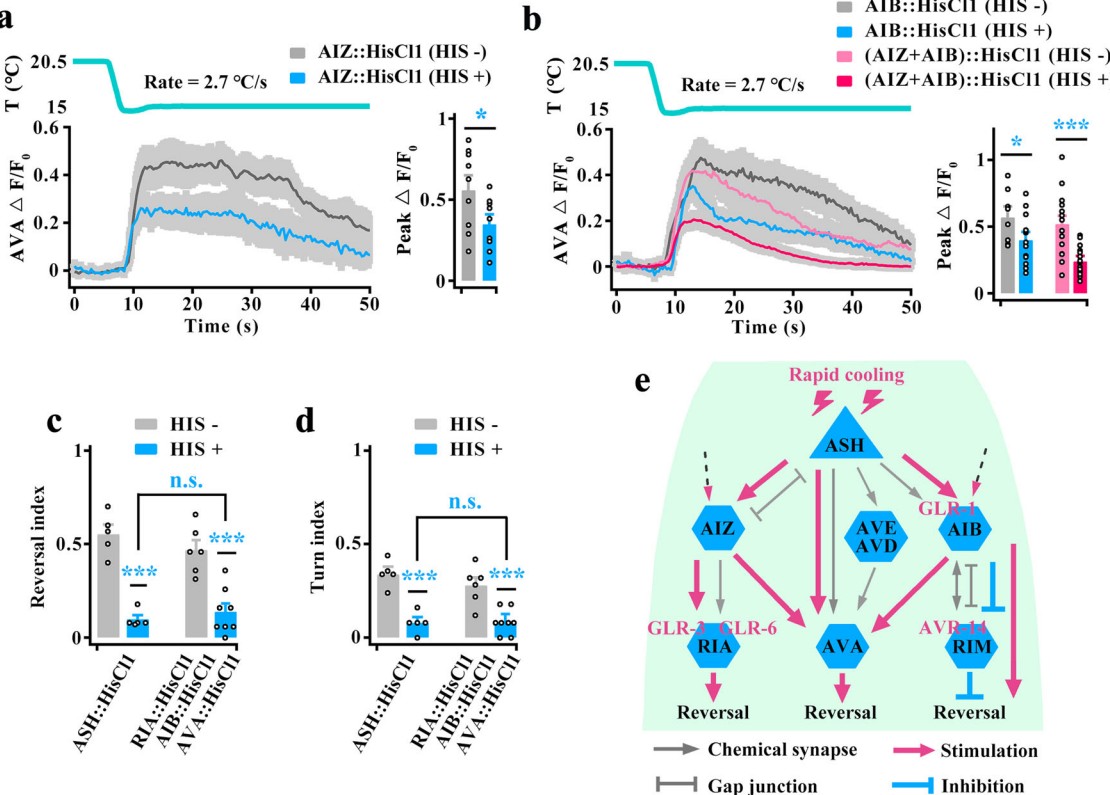

**Fig. 10 | Multiple parallel circuits mediate ASH responses to rapid cooling stimulation. a, b** AVA calcium transients induced by rapid cooling in AIZ-silenced (**a**, *n* = 9), AIB-silenced or (AIZ + AIB)-silenced (**b**, *n* = 8, 14, 15 and 17) worms. **c, d** Comparison of cooling-evoked reversal and turn index between ASH-silenced and (RIA + AIB + AVA)-silenced worms. *n* = 5, 5, 6 and 8 groups for each bar, ≥10 worms/group. **e** Schematic model showing the molecular and circuit mechanisms of the ASH response to rapid cooling stimuli. Data are showed as mean ± SEM. Student's *t* test or Mann–Whitney rank sum test (two-sided) in this figure. *$p < 0.05$, ***$p < 0.001$, $p > 0.05$ denotes not significant (n.s.).

heating[14,23], although the magnitude of calcium signal is less than that evoked by cooling. Rapid cooling- and heating-evoked calcium activity in ASH suggests that temperature sensors may mediate acute temperature stimulation. We thus attempted to explore potential sensors that mediate rapid cooling stimulation. These tested molecules included the TRPV channel proteins OSM-9 and OCR-2, the TRPA channel protein TRPA-1, the possible G protein-coupled receptor (GPCR) activation of Gα protein ODR-3 and the Gγ protein GPC-1. We found that OSM-9 and the unknown GPCRs may act as rapid cooling sensors because they exhibit obvious defects in rapid cooling-evoked calcium activity and avoidance behavior (Supplementary Fig. 19a–j), and this topic should be investigated in our next study.

In conclusion, the present study provides insights into avoidance behavior induced by rapid cooling stimuli and the molecular and neural mechanisms underlying this avoidance behavior. Many fundamental molecular and functional aspects of neural circuits are evolutionally conserved from *C. elegans* to mammals. Thus, the molecular and neural network identified in this study may help uncover the mechanisms in more complex organisms in response to rapid cooling stimulation.

## Methods
### Strains
*C. elegans* strains were maintained, transgenic strains were constructed by injection of plasmid into the germline, or by cross according to standard methods[69]. For the experiments involving transgenes, two or three independent transgenic lines were examined to confirm the results. The *C. elegans* strains used in this study are listed in Supplementary data 1.

### Neuron manipulation
For silencing neuron activity, neuron that expression of HisCl1 driven by neuro-specific promoter with application of exogenous histamine (Sigma-Aldrich), or without histamine as a control were used[28]. To ensure histamine efficiently acts on HisCl1 in the worms, a 200 μL of *E. coli* OP50 (shaking 16 h at 37 °C) with a final concentration of 30 mM histamine, were seeded on whole surface of NGM plate (60 mm diameter), which were dried at 37 °C for 48 h with lid on before spreading. These prepared plates with or without histamine were placed at room temperature and harvested the next day, stored in dark place at 4 °C, and run out within a week. The L4 worms were picked on these prepared NGM plates the day before the assay.

For blocking neurotransmission, TeTx was expressed in neuron driven by neuron-specific promoter[29]. For killing neuron, a highly efficient optogenetic membrane-targeted miniSOG, named PHmini-SOG, was used[30]. Briefly, more than 15 L1 stage worms with expression of both PHminiSOG and red fluorescence marker were first transferred to a 3 cm seeded OP50 NGM plate. Then the blue LED light (460 nm) generated by Prizmatix light source (Israel), in frequency 2 Hz (0.25 s on and 0.25 s off) controlled by digital function generator/Amplifier (RIGOL Co. Ltd, China), was used to illuminate the worms for 10 min. The intensity of blue light that received by the worms was measured at 2.12 mW/mm². After illumination, worms were kept in the dark. Killing of neurons were checked at the young adult stage by the disappearance of fluorescence from the PHminiSOG and red marker.

For activating neuron, two methods were used. One is chemogenetic method for stimulation of sensory neuron[34,46]. The sensory neuron with expression of rat TRPV1 driven by neuron-specific promoter can be excited by exogenous 100 μM capsaicin. The other is

optogenetic method for stimulation of interneuron that without cilia structure. All of optogenetic manipulations were performed on *lite-1(xu7)* genetic background to eliminate the intrinsic photophobic response[70]. The worms with the interneurons expressed of channelrhodopsin-2 (ChR2) were raised on NGM plates seeded with OP50 and All-Trans-Retinal (Sigma-Aldrich, final concentration of 5 μM), or without All-Trans-Retinal as a control[42]. To realize calcium imaging and optogenetic manipulation simultaneously in the worm, a separate blue light (460 nm, 0.5 mW/mm²) with 30 s pulse was generated by Prizmatix light source to stimulate the head of worm during the calcium imaging. The effective strains are all listed in Supplementary data 1.

### Behavioral assays

**Cooling-evoked avoidance behavior.** Behavioral assay was performed in an environmentally controlled room set at 20.5 °C and 30%–50% humidity range. To realize fast cooling, a Peltier element-based device was used for rapid cooling stimuli as previously described[21] with some modifications (Fig. 1a). Square (40 mm × 40 mm × 3.8 mm) Peltier elements with higher power density (24 V × 13 A) sandwiched between square copper sheet (60 mm × 60 mm × 0.1 mm) and a water-block that connecting liquid cooling system (Wafa, Shandong, China). Liquid coolant was maintained at 14 °C in water-block with 6 L/min circulation. Voltage and current of the Peltier element were controlled with a programmable power with a hundred-millisecond time scale (HSPY Co. Ltd, Beijing, China). Thermotaxis (Ttx) assay plates were used to perform the behavioral assay as previously described[71]. Briefly, 2 ml of Ttx medium (2% agar, 0.3% NaCl and 25 mM potassium phosphate buffer) was poured into a 6 cm plate, removed lids and dried the plates for 2 h at room temperature. Then 35 mm × 35 mm cut assay plate was placed on the copper sheet. The space between the bottom of the Ttx plate and copper sheet was filled with water. The temperature probes tap onto the Ttx plate to record temperature. For histamine-HisCl1 silencing neuron, the exogenous histamine (30 mM) was also added into the Ttx plate media before pouring.

For cooling range from 20.5 °C to 15 °C or from 20.5 °C to 17 °C, the starting temperature is holding at 20.5 °C (SEM = 0.005, $n = 50$ assays), for cooling range from 25.5 °C to 20 °C, the starting temperature is holding at 25.5 °C (SEM = 0.01, $n = 56$ assays). Before picking the worms, more than one independent cooling stimuli were performed, and the temperature onto the Ttx plate were simultaneously recorded. The cooling rate can be identified by fitting on Ttx plate surface temperature in the linear range from 20.5 °C to 15 °C, or from 20.5 °C to 17 °C, or from 25.5 °C to 20 °C, respectively. We realized the specific cooling rate and the cooling baseline temperature by finely modulating Ttx plate thickness and Peltier elements cooling power through a programmable power. Due to the slope of linear function (cooling rate) is depending on the fitting interval. We use the interval -3–18 s of temperature trace to fit cooling rate 0.4 °C/s (SEM = 0.002, $n = 50$ assays, Supplementary Fig. 1b), and use the interval -18–50 s to define the cooling baseline temperature 15 °C (SEM = 0.03, $n = 50$ assays). Using the interval -3.2–11.6 s of temperature trace to fit cooling rate 0.7 °C/s (SEM = 0.004, $n = 50$ assays, Supplementary Fig. 1c), and using -12–50 s to define the baseline temperature 14.9 °C (SEM = 0.02, $n = 50$ assays). Using the interval -3.6–7.6 s of temperature trace to fit cooling rate 1.4 °C/s (SEM = 0.007, $n = 50$ assays, Supplementary Fig. 1d), and using -8–50 s to define the baseline temperature 14.9 °C (SEM = 0.02, $n = 50$ assays). Using the interval -3.4–5.4 s of temperature trace to fit cooling rate 2.7 °C/s (SEM = 0.02, $n = 60$ assays, Supplementary Fig. 1e), and using -6–50 s to define the baseline temperature 15 °C (SEM = 0.03, $n = 60$ assays). Using the interval -3.6–4.8 s of temperature trace to fit cooling rate 2.7 °C/s (SEM = 0.03, $n = 50$ assays, Supplementary Fig. 1f), and using -5–50 s to define the baseline temperature at 17 °C (SEM = 0.03 °C,

$n = 50$ assays). Using the interval -3.6–5.6 s to fit cooling rate 2.7 °C/s (SEM = 0.02, $n = 56$ assays, Fig. 1d), and using -6–50 s to define the baseline temperature at 19.9 °C (SEM = 0.03 °C, $n = 56$ assays). These recording traces are in the Source data file.

Then 5 to 8 well-fed day1, cultivated at 20.5 °C worms were washed in M9 buffer with or without exogenous histamine (30 mM, for silencing worm), were picked onto the assay plate to rest at starting temperature 20.5 °C or 25.5 °C for at least 2 min respectively. Cooling stimulus was applicated to the animals which exhibiting forward sinusoidal locomotion at least 3 s. The locomotion behaviors of the worms were captured using a Zeiss discovery V16 stereomicroscope (Zeiss, Germany) with an EMCCD camera that controlled by Image-Pro Plus 6.0 (Media Cybernetics, Inc., Rockville, MD, USA). The instantaneous behavioral states consisting of discrete states: forward, reverse, turn, and pause in individual animal were analyzed by Image-Pro Plus 6.0. A cooling-evoked reversal or turn is defined by the worm stopping forward movement and initiating a reversal with at least half a head swing or initiating a turn, respectively, within 3–8 s after initiation of stimulation. An omega turn following the cooling-evoked reversal without interruption was also defined as a cooling-evoked turn. Worms were stimulated only once and the temperature curves during the stimulation were recorded. Similarly, the cooling rate and baseline temperature during stimulation were calculated. For example, the cooling rate at 0 °C/s and constant temperature at 20.5 °C (SEM = 0.006); cooling rate at 0.4 °C/s (SEM = 0.002) and cooling baseline temperature at 15.0 °C (SEM = 0.04); cooling rate at 0.7 °C/s (SEM = 0.003) and cooling baseline temperature at 15.0 °C (SEM = 0.04); cooling rate at 1.4 °C/s (SEM = 0.01) and cooling baseline temperature at 15.0 °C (SEM = 0.03); cooling rate at 2.7 °C/s (SEM = 0.02) and cooling baseline temperature at 15.0 °C (SEM = 0.03); cooling rate at 2.7 °C/s (SEM = 0.02) and cooling baseline temperature at 17.0 °C (SEM = 0.02); cooling rate at 2.7 °C/s (SEM = 0.02) and cooling baseline temperature at 20.0 °C (SEM = 0.02).

We then compared the differences of cooling rate and baseline temperature between worm pre-stimulation and during stimulation by using One-way ANOVA test. The results showed no significant differences in cooling rate 0 °C/s (F = 0.13, $P = 0.72$) and constant temperature at 20.5 °C (F = 0.00001, $P = 0.99$); in cooling rate 0.4 °C/s (F = 1.454, $P = 0.233$) and cooling baseline temperature at 15.0 °C (F = 2.62, $P = 0.111$); in cooling rate 0.7 °C/s (F = 0.137, $P = 0.713$) and cooling baseline temperature at 15.0 °C (F = 1.675, $P = 0.201$); in cooling rate at 1.4 °C/s (F = 1.239, $P = 0.271$) and cooling baseline temperature at 15.0 °C (F = 2.181, $P = 0.146$), in cooling rate at 2.7 °C/s (F = 2.671, $P = 0.108$) and cooling baseline temperature at 15.0 °C (F = 1.433, $P = 0.236$), in cooling rate at 2.7 °C/s (F = 2.054, $P = 0.158$) and cooling baseline temperature at 17.0 °C (F = 0.385, $P = 0.537$), in cooling rate at 2.7 °C/s (F = 3.45, $P = 0.07$) and cooling baseline temperature at 20.0 °C (F = 3.07, $P = 0.085$). The results suggest that the specific of cooling rate and baseline temperature are the same between pre-stimulation and during stimulation, and thus the genotype-associated differences reported in the Result section cannot be explained by any systematic variations elicited by cooling, but explained by the changed of neuron activity evoked by cooling.

**Spontaneous locomotion behavior.** For capsaicin-elicited spontaneous locomotion behavior without cooling stimulation (Fig. 6f), the exogenous capsaicin (100 μM) was added into the Ttx plate media before pouring. Worm locomotion behaviors without cooling stimulation were recorded in 10 min and the number of reversals were calculated[42]. A reversal is defined by the worm stopping forward movement and initiating a reversal with at least half a head swing.

### In vivo calcium imaging

**Gluing worm calcium imaging.** In vivo gluing worm calcium imaging of neuron was performed with the same environmental conditions as

behavioral assay, and essentially according to previous reports with some modifications[50,67]. Worms expressing GCaMP6f or R-GRCO1 were used for calcium imaging. A custom-built ring of Peltier-based temperature controller that could be mounted onto the stage of an Olympus IX-70 inverted microscope (Olympus, Japan) equipped with a × 40 objective lens. With the same programmable power as behavioral assay, this device also realized different cooling rate stimulation.

Employing the same temperature control protocol as described above for the behavioral assay. One independent cooling stimulation to a 2% (W/V) agar pad pre-gluing the worm and the temperature traces were recorded. We realized the specific cooling rate and baseline temperature also by fine-tuning Peltier elements power and agar thickness. Then a day 1 worm was glued onto the agar pad, immersed in M9 buffer (20 μL), the temperature probe was also immersed in M9 buffer and tapped on the agar pad and close to worm nose. Gluing the tip of worm's head should be avoided. Sample preparation was completed within 2 min. One worm only imaging once. For histamine-HisCl1 silencing the neuron activity, the exogenous histamine (30 mM) was added into the both agar pad before pouring and M9 buffer. For capsaicin-TRPV1 activating neuron during cooling stimulation, the exogenous capsaicin solution (100 μM) was added into the M9 buffer within 17 s before cooling stimulus because there is a about 17 s delay after delivering capsaicin according to microfluidic chip results[34].

Similarly, one-way ANOVA test was used to examine the worms suffered the same stimulation of specific cooling rate and the baseline temperature across experiments. The statistical results showed no significant differences in cooling rate and the baseline temperature between pre-stimulation and during stimulation. Such as rate of 0 °C/s (F = 0.08, $P = 0.77$) and constant temperature at 20.5 °C (F = 0.155, $P = 0.698$), rate of 0.4 °C/s (F = 0.62, $P = 0.44$) and baseline temperature of 15.0 °C (F = 2.69, $P = 0.118$), rate of 0.7 °C/s (F = 0.338, $P = 0.566$) and baseline temperature of 15.0 °C (F = 0.803, $P = 0.379$), rate of 1.4 °C/s (F = 2.384, $P = 0.136$) and baseline temperature of 15.0 °C (F = 2.4, $P = 0.134$), rate of 2.7 °C/s (F = 2.532, $P = 0.123$) and baseline temperature of 15.0 °C (F = 0.397, $P = 0.534$), rate of 2.7 °C/s (F = 1.158, $P = 0.292$) and baseline temperature of 17.0 °C (F = 3.798, $P = 0.062$), rate of 2.7 °C/s (F = 2.101, $P = 0.158$) and baseline temperature of 20.0 °C (F = 0.293, $P = 0.593$). The results suggest that the specific of cooling rate and baseline temperature are the same between pre-stimulation and during stimulation, and thus the calcium signal differences reported in the Result section cannot be explained by any systematic variations, but explained by the changed of neuron activity.

**Microfluidic calcium imaging.** For only capsaicin activating TRPV1 calcium imaging, a home-made microfluidic device was used as previously described[34,72]. Briefly, a worm was gently loaded into a microfluidic chip channel with its nose exposed to the flowing M9 buffer. M9 buffer with or without capsaicin solution (100 μM) were delivered using a programmable automatic drug feeding equipment (InBio life Science Instrument Co. Ltd, Wuhan, China).

To eliminate the light-evoked calcium activity, all the tested neurons were exposed under fluorescent excitation light for 2 min before recording. All fluorescence images were captured by an EMCCD camera with a 10 ms exposure time and 512 × 512 pixels at 0.33 s frame interval. The average fluorescence intensity of region of interest (ROI) of the soma and axon was captured and analyzed by using Image-Pro Plus 6.0 (Media Cybernetics, Inc., Rockville, MD, USA). An adjacent ROI in each frame was used to subtract the background. The average fluorescence intensity within the initial 5 s before cooling stimulation was taken as the basal signal, $F_O$. The change in fluorescence intensity relative to the initial intensity $F_O$, $\Delta F/F_O = (F - F_O) / F_O$, was plotted as a function of time for all curves by using IGOR Pro 6.12 (Wavemetrics, USA). The mean value of calcium signal and the SEM were plotted in various colors as indicated and in light grey, respectively. The maximum of $\Delta F/F_O$ during 5−50 s was defined as the peak $\Delta F/F_O$, and the

peak $\Delta F/F_O$ for each trace higher than 0.05 was defined as a calcium response. The peak $\Delta F/F_O$ was calculated for statistical analysis.

## Molecular biology

Three-fragment multisite gateway system (Invitrogen, Thermo Fisher Scientific, USA) and In-fusion (Clontech Laboratories, Inc., USA) methods were used to generate most plasmids as previously described[34,73,74]. Briefly, three entry clones containing three PCR products (promotor, gene of interest, sl2::RFP, sl2::GFP or 3'UTR, in name of slot1, slot2 and slot3, respectively) were recombined into the pDEST™ R4-R3 Vector II or custom-modified vectors using attL-attR (LR) recombination reactions to generate the expression plasmids. To specifically label AIZ neurons, a FLP/FRT site-specific recombination system and In-fusion methods were used[75]. Promoters labeling specific neurons are described as follows: AWA: odr-7[76] (5 kb), AWB: str-1[77] (4 kb), AWC^ON: str-2[78] (3.7 kb), AWC^OFF: srsx-3[79] (0.88 kb), AFD: gcy-23[80] (2.5 kb), ASER: gcy-5[81] (3.17 kb), ASEL: gcy-7[81] (1.28 kb), ADF: srh-142[82] (3.94 kb), ASG: gcy-15[80] (0.69 kb), ASH: sra-6[83] (3.8 kb), ASI: gpa-4[84] (2.4 kb), ASJ: trx-1[85] (1.0 kb), ASK: sra-9[83] (3 kb), ADL: ver-2[86] (2.72 kb), AIA: gcy-28d[87] (2 kb), AIB: npr-9[88] (2 kb), AIY: T19C4.5[89] (4 kb), AIZ: (ser-2(2) (4.72 kb) + odr-2b (2.64 kb))[42], RIB: sto-3[90] (0.98 kb), RIA: glr-3[55] (2.82 kb) or glr-6[55] (7 kb), RIM: cex-1[91] (1.02 kb), SMB: flp-12[92] (0.51 kb), AVA: rig-3[93] (3.94 kb), (AVA + AVB + AVE + AVD): nmr-1[94] (1.1 kb), RIC: tbh-1[52] (4.6 kb), RMG: ncs-1[95] (3.11 kb). Promoters for rescuing loss-of-function genes are eat-4[52] (5.5 kb), glr-1[55] (5.3 kb), avr-14[41] (1.85 kb), avr-15[96] (4.2 kb) and osm-6[39] (2.48 kb). All primers for cloning these promoters are listed in Supplementary data 2.

Gene of interest was used to generate the slot2 entry clone, including the genes of glr-1(4480 bp), glr-3(3738 bp), glr-6 (4511 bp), avr-14 (6074 bp), avr-15 (2227 bp), eat-4 (4941 bp), and osm-6 (1419 bp), which of them were PCR amplified from N2 wild-type cDNA library, and the genes of unc-7 (5753 bp) and unc-9 (3383 bp), which of them were PCR-amplified from N2 genomic DNA library, and the fragments of GCaMP6f, toxin light chain (TeTx), HisCl1, PHminiSOG, TRPV1, flp-sl2-flp, and ChR2. Neuron-specific RNAi was generated as previously described[54]. The eat-4 RNAi segment was amplified from N2 genomic DNA with primers containing attB sites, and recombined with attP sites in the pDONR221 vector to generate the entry clone slot2. All the primers for cloning genes of interest are listed in Supplementary Data 2. The fragments of sl2::RFP, sl2::GFP or 3'UTR were recombined to produce the slot3 entry clone.

Finally, the expression plasmids were generated by using LR reaction among the mixture of slot1, slot2, and slot3. The constructed plasmids are as followed: sra-6p::GCaMP6f, sra-6p::TeTx::sl2::RFP, sra-6p::HisCl1::sl2::RFP, sra-6p::PHminiSOG::sl2::RFP, sra-6p::TRPV1::sl2::RFP, sra-6p::eat-4::sl2::RFP, sra-6p::osm-6::sl2::RFP, sra-6p::unc-7::sl2::RFP, sra-6p::unc-9::sl2::RFP, odr-7p::HisCl1::sl2::GFP, str-1p::HisCl1::sl2::GFP, str-2p::HisCl1::sl2::GFP, str-2p::HisCl1::sl2::RFP, str-2p::PHmini-SOG::sl2::RFP, srsx-3p::HisCl1::sl2::GFP, srsx-3p::HisCl1::sl2::RFP, srsx-3p::PHminiSOG::sl2::RFP, gcy-23p::HisCl1::sl2::GFP, gcy-23p::HisCl1::sl2::RFP, gcy-23p::PHminiSOG::sl2::RFP, gcy-5p::HisCl1::sl2::GFP, gcy-5p::HisCl1::sl2::RFP, gcy-5p::PHminiSOG::sl2::RFP, gcy-7p::HisCl1::sl2::GFP, srh-142p::HisCl1::sl2::GFP, gcy-15p::HisCl1::sl2::GFP, gcy-15p::HisCl1::sl2::RFP, gcy-15p::PHminiSOG::sl2::RFP, sra-6p::HisCl1::sl2::GFP, gpa-4p::HisCl1::sl2::GFP, trx-1p::HisCl1::sl2::GFP, sra-9p::HisCl1::sl2::GFP, ver-2p::HisCl1::sl2::GFP, T19C4.5p::TeTx::sl2::RFP, T19C4.5p::HisCl1::sl2::RFP, T19C4.5p::PHminiSOG::sl2::RFP, odr-2bp::flp-sl2-flp, ser-2(2)p::frt-stop-frt-TeTx::sl2::RFP, ser-2(2)p::frt-stop-frt-HisCl1::sl2::RFP, ser-2(2)p::frt-stop-frt-PHminiSOG::sl2::RFP, ser-2(2)p::frt-stop-frt-GCaMP6f, ser-2(2)p::frt-stop-frt-TRPV1::sl2::RFP, ser-2(2)p::frt-stop-frt-eat-4(RNAi)::sl2::RFP, ser-2(2)p::frt-stop-frt-unc-7::sl2::RFP, ser-2(2)p::frt-stop-frt-unc-9::sl2::RFP, ser-2(2)p::frt-stop-frt-ChR2::sl2::RFP, npr-9p::TeTx::sl2::RFP, npr-9p::HisCl1::sl2::RFP, npr-9p::PHminiSOG::sl2::RFP, npr-9p::GCaMP6f, npr-9p::glr-1::sl2::RFP, npr-9p::eat-4(RNAi)::sl2::RFP, npr-9p::ChR2::sl2::RFP, gcy-28dp::TeTx::sl2::RFP, gcy-28dp::HisCl1::sl2::RFP, gcy-28dp::

*PHminiSOG::sl2::RFP, flp-12p::TeTx::sl2::RFP, flp-12p::HisCl1::sl2::RFP, flp-12p::PHminiSOG::sl2::RFP, glr-3p::TeTx::sl2::RFP, glr-3p::HisCl1::sl2::RFP, glr-3p::PHminiSOG::sl2::RFP, glr-3p::GCaMP6f, glr-3p::glr-1:: sl2::RFP, glr-3p::glr-3::sl2::RFP, cex-1p::TeTx::sl2::RFP, cex-1p::HisCl1::sl2::RFP, cex-1p::PHminiSOG::sl2::RFP, cex-1p::GCaMP6f, cex-1p::TRPV1::sl2::RFP, cex-1p::avr-14::sl2::RFP, cex-1p::avr-15::sl2::RFP, sto-3p::TeTx::sl2::RFP, sto-3p::HisCl1::sl2::RFP, sto-3p::PHminiSOG::sl2::RFP, rig-3p::HisCl1::sl2::RFP, rig-3p::PHminiSOG::sl2::RFP, rig-3p:: GCaMP6f, nmr-1p:: HisCl1::sl2::RFP, nmr-1p::PHminiSOG::sl2::RFP, nmr-1p::GCaMP6f, tbh-1p::GCaMP6f, ncs-1p::GCaMP6f, eat-4p::eat-4::sl2::RFP, glr-6p::glr-6::sl2::RFP, glr-1p::glr-1::sl2::RFP, avr-14p::avr-14::sl2::RFP, avr-15p::avr-15::sl2::RFP, osm-6p:: osm-6::sl2::RFP.*

### Statistical data analysis

Data are expressed as the mean ± SEM, and the statistical analysis was performed using SPSS software V19.02 (IBM, Inc). Comparison of two groups, student's *t test* was used when the data meets the Gaussian-distributed, and Mann–Whitney rank sum test was used when the data do not meet the normal distribution, respectively. Comparison of more than two groups, one-way ANOVA followed by Dunnett's multiple comparisons when the data meets the normal distribution, and Kruskal–Wallis test with Dunnett's multiple comparisons when the data do not meet the normal distribution. Asterisk denotes the statistical significance compared with the control: *$p < 0.05$, **$p < 0.01$, ***$p < 0.001$, n.s. denotes not significant.

### Reporting summary

Further information on research design is available in the Nature Portfolio Reporting Summary linked to this article.

## Data availability

All data in main Manuscript and Supplementary information are listed in the Source data file. Strains used in this study are shown in Supplementary data 1. All the primers for cloning promoter and genes of interest are listed in Supplementary data 2. The associated protocols used this study are available in the main manuscript. Materials used for this study will be available upon requests to the corresponding author. Source data are provided with this paper.

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

## Acknowledgements

We would like to thank the *Caenorhabditis* Genetic Center and National BioResource Project (NBRP) for strains. We are grateful for critical comments from Dr X.Z. Shawn Xu. We Thank Dr. E.M. Jorgensen for TeTx, Dr. C. Bargmann for the plasmids contenting of HisCl1, Dr. Yanxun V. Yu for the plasmids contenting of PHminiSOG, Dr. J. Yao for rat TRPV1 cDNA. Dr. zhaoyu Li for technical assistance and promoter *rig-3*. This work is supported by the Natural Science Foundation of China (grant 32171004) and Fundamental Research Funds for the Central Universities (grant 510319165).

## Author contributions

M.G., and Y.X.S, C.X.L., and Z.Y.W. conceived and designed the study. C.X.L., Y.X.S., and Z.Y.W. performed the majority behavioral and imaging experiments. C.X.L. and H.P. constructed and injected the majority plasmids. J.Y.H., W.W.S., R.L. and P.Z.W. participated in partial behavioral analyses, molecular cloning and injection. M.G., C.X.L., and Y.X.S. prepared all figures and movies. M.G., and Z.X.W. wrote the manuscript. All authors revised the manuscript.

## Competing interests

The authors declare no competing interests.
