## [Peer Review File · Nature Communications]

Molecular and circuit mechanisms underlying avoidance of rapid cooling stimuli in *C. elegans*REVIEWER COMMENTS

Reviewer #1 (Remarks to the Author):

In this article, Shan and collaborators provide an in-depth characterization of the neuronal circuit underlying cooling-evoked avoidance in crawling *C. elegans*. Cold-evoked behaviors are among the least characterized sensory-behaviors to date. Even in the powerful neurogenetic model *C. elegans*, we know very little. This article is timely and I am impressed by the huge number of technically-challenging experiments carried out to decipher the circuit underlying this response and the well-designed figures that efficiently present this complex dataset. I would like to congratulate the authors for this. The scientific outcome will be of interest to specialists and non-specialists in the field of neuroscience. Experiments are well-designed and most scientific claims are well supported by data. However, one important exception are the claims about ASH being a direct cooling-sensor, which either call for additional experiments or need to be rephrased as a hypothesis (in the latter case, without diminishing the impact of the work). In addition, several further controls are still needed in order to fully support the proposed models and there are major writing issues that need to be fully addressed before I can recommend publication of this article.

Major scientific concerns:

1. In order to validate the physiological relevance of calcium data analyses, all the calcium imaging lines should be assessed for normal cooling-evoked behavior.
2. *unc-31* and *unc-13* experiments are great, but the conclusion is an overstatement, as neither mutant affect gap junctions. Actually, it is totally possible that ASH is not a primary cool sensor and receives electrical inputs from one of several other cells. RIC, RMG, AIZ and ADA all connect to ASH by gap junctions. Furthermore considering the latest connectomic data by Witvliet et al. (2020, nemanode) together with more classic circuit description by White et al., one realizes that AIZ makes electrical connections with ASER, AWC and ASG (all three cool-responsive neurons), as well as AFD. Therefore, the ASER-AIZ-ASH or the AWC-AIZ-ASH connection could be at the base of the ASH cool-evoked response and ASH might not be directly the primary cooling sensor. Showing intact cooling-evoked ASH calcium activity in AWC-killed, ASE-killed, ASG-killed or (maybe even better) AIZ-killed backgrounds would be needed to ascertain the proposed model.
3. *str-199* promoter was used for targeting AWC: Is this promoter really expressed in AWC, and if yes, is it both AWCon and AWCoFF or only one of them? CENGEN data suggests the one cell the author see in the head could very well be ADL and not AWC. Providing further validation is essential, because I find premature to conclude that AWC is not mediating the cooling-evoked response based on this analysis with a poorly characterized promoter. Also, authors should test AWCon and AWCoFF separately.
4. The conclusion about *unc-7* and *unc-9* mutants are over-statements (Fig. 2e), because the authors have not done cell-specific rescue of either gene. It is not possible to confirm that the AIZ-ASH connection matters. It could well be the ASE-AIZ or the AWC-AIZ connections that are involved (in line with point 3 above) or any other electrical synapses elsewhere in the circuit, working indirectly.
5. "Taken together, the behavioral and calcium results confirm that both AIZ and AIB neurons directly receive signal flow from ASH neurons to mediate rapid cooling stimulation" The data are compatible with this model, but alternative models are possible. Therefore, this statement must be toned down.
6. Capsaicin data (Fig. 2f,h,i) would be more convincing if a non-transgenic control lacking the TRPV1 transgenes and comparing -CAP and +CAP was shown.
7. Same thing for the other CAP data in Fig 3 and 4.
8. The data for AIZ-specific *eat-4* RNAi, lack a proper control to demonstrate that the presence of a multi-copy transgene with AIZ-specific promoter is not per se impeding AIZ function.
9. It's quite surprising that the reversal and omega turn phenotypes are always similarly regulated in almost all the neuronal function impairing strains. Could authors further clarify if they always observed omega turn after reversal in this context? Alternatively, could they discuss how the same circuit controls both phenotypes?

10. AIB receive more synapse from AIZ compared to ASH (nemanode). Is it possible that ASH-AIZ-AIB pathway is more dominant compared to ASH-AIB pathway in this context? Could AIB still be activated (by cooling or chemogenetics) when AIZ is killed/inhibited?

11. How does RIA induce reversal in this context, as it lacks outgoing synapse to previously known reversal promoting neurons?

12. ASH has considerable outgoing synapse to reversal promoting AVA and AVD neurons (nemanode). AVA plays crucial role in control of reversals (several previous studies). Therefore it becomes very crucial to test the role of AVA in the present study. What is the role of AVA and AVD in the context of cold evoked reversals? Is it possible that in this context both AIZ and AIB circuit converge on AVA to control reversals? Could authors monitor AVA calcium activity in AIB, RIA double HisCl strain? Could they compare reversal and omega turn phenotype by blocking AVA activity and comparing it to AIB, RIA double HisCl strain?

13. Except for the result section which seems to have received more attention, the writing is poor overall and very poor in the Introduction and Method sections. There are many over-statements and sentences that make simply no sense. As a courtesy to the authors, despite the lack of line numbering, and because I was anyway getting over the manuscript in detail, I tried to list the most serious writing issues below, but additional improvements are needed everywhere along the manuscript. But honestly, this should have been done before submitting the manuscript.

Major writing concerns:

- Summary: "The ability to sense and respond to ambient temperature change is the most critical function of the nervous system." Overstatement with unjustified subjective judgment
- Introduction: "Temperature is the most important environmental cue that affects every biological and biochemical process." Overstatement with unjustified subjective judgment
- Introduction: "Thus, the ability to sense temperature and respond promptly is critical for the survival of all organisms." This might be true for animals but certainly not for all organisms.
- Introduction: 2nd paragraph about threshold temperature is confusing. There is a well-established notion in the field about the threshold temperature for AFD responsiveness, which is modulated by past experience. Then the responsiveness of the other neurons (AWC, ASI, FLP, PHC) is either not well characterized or totally unrelated to the threshold in AFD. As it reads now, it is totally unclear as to why authors talk about THE threshold. They should talk about "A threshold" or "different thresholds" and better define it/them.
- Introduction: "Unlike thermosensation (thermotaxis and heat avoidance), which has been widely studied previously, cold sensation has increasingly been the focus of worm acclimation to environmental temperature in recent years in *C. elegans*." This sentence makes no sense to me
- Introduction: "which means that neurons directly respond to cold stimuli, is still unknown" Syntax error
- Introduction: "to regulate body turning in buffer water" what does this mean?
- Overall: the verb 'adopt' seems misused most of the time.
- "To date, the locomotion behavior elicited by acute cold stimulation in a single worm is mainly using cold buffer water to trap the worm and evoke body turn locomotion" Syntax error
- "To test more behavioral states in a freely behaving worm underlying acute cold stimulation" what is a behavioral state for the author? Please define
- "was delivered to a forward sinusoidal locomotion worm" syntax error
- "We found that ASHs calcium activities showed a cooling rate-dependent way during the cooling range from 20.5 °C to 15 °C" What does 'way' mean?
- "As previous studies have shown that ASHs are not required for cooling sensation" Citation needed.
- "Because TeTx has a weak effect on only gap junctions between ASH and AIZ neurons" Not clear. Is it an observation? An assumption? the syntax looks wrong.
- "spontaneous locomotion": what does it mean? This must be defined.
- "RIM-inhibited worms displayed dramatically increased avoidance behavior" why using the word "dramatic", when the effect is obviously mild?
- "The glutamate-gated chloride channel AVR-14 encodes inhibition of RIM by AIB" the term 'encodes' seems strange.

- "In this study, we showed for the first time that *C. elegans* senses the cooling rate directly and avoids a rapid cooling stimulus (2.7 °C/s) within the physiological temperature range from 15 °C to 25 °C. Further studies demonstrated for the first time that polymodal ASH sensory neurons sense rapid cooling and regulate cooling-evoked avoidance behavior" All these 'first time' statements should be avoided. In addition, the part about ASH being a cooling sensor is, at best, totally premature.
- "Our data also showed probabilistic calcium activity in RIM during rapid cooling stimulation" I would not say that. I would say "Our data also showed probabilistic modulation of calcium activity in RIM during rapid cooling stimulation"
- "However, worms still showed a proportional avoidance behavior when the activities of both the stimulatory circuit and disinhibitory circuit were silenced (RIA::HisCl1 + AIB::HisCl1)" proportional to what? This is not clear.
- "It has been confirmed that AVA command interneurons that directly receive stimulation signals from ASHs to regulate various aversive stimuli that evoke avoidance behavior." This looks like a chopped sentence.
- "In this study, we find that ASH neurons can sense the cooling rate directly; thus, future work should also try to identify cooling rate sensors." Again, evidence for ASH being a primary cooling sensor is insufficient to claim this.
- "L4 worms with expression of HisCl1 were first picked on a thin OP50 lawn with 30 mM histamine" Please explain how a thin lawn is defined/produced.
- "By finely modulating Ttx plate thickness and the Peltier elements cooling power through a programmable power, we realized the cooling rate at 0 °C/s and constant temperature at (20.5 ± 0.008) °C, cooling rate at (0.4 ± 0.002) °C/s and baseline temperature at (15.0 ± 0.03) °C, cooling rate at (0.7 ± 0.004) °C/s and the baseline temperature at (15.0 ± 0.03) °C, cooling rate at (1.4 ± 0.007) °C/s and the baseline temperature at (15.0 ± 0.04) °C, cooling rate at (2.7 ± 0.02) °C/s and the baseline temperature at (15.0 ± 0.03) °C, cooling rate at (2.7 ± 0.03) °C/s and the baseline temperature at (20.0 ± 0.03) °C."
- What does the +/- values represent, how were they calculated, and were the recordings so precise that a precision of 0.002 °/s could be claimed. Hard to believe.
- "Usually five well-fed day1, cultivated at 20.5 °C worms were washed in M9 buffer with or without exogenous histamine (30 mM, for silencing worm), and then picked onto the assay plate to rest at starting temperature 20.5 °C or 25.5 °C for at least 2 min respectively" Why using the term 'Usually'? What was the unusual protocol and when was it applied?
- More details are needed to understand what was done with Image-Pro Plus 6.0?
- "To test the worms suffered the same stimulation in specific cooling rate and the baseline temperature across experiments" unclear
- "Therefore, these statistical results confirm the different phenotypes in this study are inherent characteristics of the worms, rather than effected by any systematic variations" wrong syntax
- Likewise, references for the choice of promoters are needed.

Reviewer #2 (Remarks to the Author):

In this manuscript, the authors identified new neural circuits and molecules that sense temperature rapid changes and regulate aversive behavior in *C. elegans*. *C. elegans* grow between 15 degrees to 25 degrees, and some sensory neurons such as AFD, AWC, OLL, PVC, and PHC that sense temperature stimulus have been known, and they are located all over the body, including the head, body, and tail. The most well-known AFD is known as a sensory neuron that responds very sensitively to temperature changes and can remember the experienced temperature to control behavioral plasticity. The authors showed that the ASH circuit, which is different from the AFD neural circuit, controls two kinds of downstream neural circuits, thereby facilitating the aversive behavior efficiently against rapid temperature changes. In this process, various techniques were used, such as quantitative behavioral

analysis, neural activity, neural manipulation, and genetic analysis. In addition, each experimental result had high reproducibility, and through neuronal manipulation of specific neurons, the function and information processing between neurons involved in aversive behavior were well understood. On the other hand, some ambiguous information needs to be clarified, and the current manuscript is missing scientifically intriguing points that should be brought up based on their experimental results.

1. The authors analyzed the behavioral change of worms in response to rapid cooling rates of 0, 0.4, 0.7, 1.4, and 2.7 degree/s, demonstrating that they exhibited reversal or turn behavior, particularly for temperature changes of 1.4 degree/s or more. As can be seen in Supplement Video 2, this aversive behavior occurred at the end of the temperature change. Do the temperature stimuli shown in each Figure represent the actual temperature the worm sense, or are they the temperature values set in the temperature control device they used? If it is the latter case, there is no problem, but if not, it is necessary to clarify whether ASH senses a temperature rapid change or the point at which temperature changes stop. The results also indicate similar neural activity in response to temperature stimuli, in addition to the observed behavioral patterns.

2. The author argues that the sensory information of temperature rapid change in ASH induces avoidance behavior through two circuits, stimulatory and disinhibitory circuits. In Fig. 8 they examine the relationship between these two circuits and demonstrate that they synergistically contribute to avoidance behavior. However, it is necessary to examine whether there is any change in the neural activity of the other pathway when the information processing of one pathway is blocked. For example, RIM, which represents a probability response of about 50% in the wild type, shows a further decrease in calcium response probability in AIZ-silenced worm or *glr-3*; *glr-6* mutant, as shown in Fig. 6q, or whether there is no change. Other cases are conceivable, of course.

3. The meaning of "control" in all graphs is ambiguous. For example, does "control" in Fig 1j-k refer to wild type? It is necessary to clarify and define what the control is for HisCl1, TeTx, and PHminiSOG in Fig. 2b-c. Their controls cannot be the same, and at least in experiments using Neuron::HisCl1, the control should be without the addition of histamine. The same applies to other graphs. The capsaicin experiment with neurons expressing TRPV1 makes sense.

4. The rapid cooling stimulus that the author focuses on, is performed under the temperature range (from 15 to 25 degrees) in which the worms can survive. If the ASH circuit is known to be a circuit that is promoted by harmful stimuli for survival, then it is necessary to discuss the relationship between the growth environment and why rapid cooling(2.7degree/s) should promote aversive behavior. So, in addition to the current paradigm, further discussion is needed on understanding animal behavior regulation, which includes internal and external states in response to temperature stimulus.

Minor;

1. Kainite in 12 lines on 4 pages is misspelled. Correct is kainate.
2. Clarify the wiring diagram information referenced in Figures 2A, 3A, and 6A.
3. ASH sensory neurons trigger reversals and omega turns in response to many aversive stimuli. Has there been any testing to see if the calcium transients and the sensory adaptation in ASH, *oms-9* and *gpc-1*, are involved in aversive behavior triggered by rapid cooling?
4. ASHs have been shown to sense rapid cooling and induce avoidance behavior. If rapid cooling similar to the detection of a noxious stimulus, is associated with aversive behavior mediated by ASH, consider the possibility of ASH circuit involvement in response to a rapid increase in temperature within the physiological temperature range.
5. In relation to Ca²⁺ imaging figures, please explain in materials and methods the peak delta F/F₀, which is used to compare neural activity between animals, is specifically calculated for what time interval.
6. The authors ablated the sensory or interneuron using various ways. According to Gray's study in 2005, these ablated worms, especially RIA or RIM-ablated worms, were found to have an effect on

reversal behavior. We are concerned whether the behavioral deficits due to neuron inhibition had no impact on the behavioral analysis during rapid cooling.

7. In relation to page 10, line 3, put the statistics difference between eat-4 and AIZ::HisCl1 in the graph of Fig 4C.

8. To aid in understanding, it would be helpful to move ASH::TRPV1 located at the top of the graph in Fig 5f, to the bottom of the graph, similar to the way it is presented in Supplementary Figure 6C.

Reviewer #3 (Remarks to the Author):

In this manuscript, the authors discovered a new sensory response in *C. elegans*, showing that the worms respond to fast cooling with avoidance behavior. To characterize how the avoidance of cooling is regulated, they identified that the sensory neuron ASH detects cooling stimulation, and two downstream circuits positively regulate the behavioral response together. Both of these circuits utilize glutamate to transmit signals with one engaging excitatory glr-3 and glr-6 receptors and the other using avr-14-mediated inhibitory signals. The study is valuable in identifying and characterizing a new behavioral response to a common sensory cue. The analysis on the regulatory circuit and molecules are comprehensive. However, there are concerns about data analysis and presentation, and the results are often not adequately discussed. These issues must be addressed.

1. The authors used "reversal index" and " Ω turn index" to quantify fast cooling-induced avoidance, but they did not explain how these indices are calculated. This information is critical because in many of their experiments where the activity or neurotransmission of specific neurons are altered, baseline reversals and Ω turns are likely altered as a result. For example, it was shown that activating AIB triggers reversals (Gordus et al Cell 2015) and AIZ is needed for initiating reversals (Liu et al Cell 2014). Inactivating AIB or AIZ likely changes reversal rate or turning rate even when the worms are not stimulated. Similarly, they disrupted both AIZ and AIB circuits in Fig8 using either HisCl1 or glr-1;glr-3;glr-6 triple mutants. These genetic changes will also alter baseline reversals and turns. These baseline effects need to be considered, for example, by measuring reversal rate without cooling and using it as a baseline to calculate cooling induced changes in reversals and turns.

2. They used chemical-genetic experiments to activate several neurons in the circuit with capsaicin in order to examine the effects on downstream neurons and behavior. These include interneurons such as AIB, AIZ and RIM that do not have cilia or opening to the outside. How does CAP access these neurons? Is the expression of TRPV1 in these interneurons specific and is it possible that some sensory neurons actually express TRPV1 in these transgenic lines?

3. The authors chose 4 downstream interneurons (AIA, AIB, AIY and AIZ) to characterize in this study. However, AIZ also send lots of synapses to AVE. ASH also strongly synapse onto AVA, AVB and form gap junctions with RIC, RMG, ADA. The rationale of their choices needs to be clarified. Additionally, the authors discussed that "Thus, future studies should clarify whether AVA neurons or other neurons modulate avoidance behavior during rapid cooling stimulation." Many previous studies have shown that AVA acts as an important downstream neuron of ASH to mediate worm behavior. The analysis of AVA in this fast-cooling response should be included in this study.

4. In their calcium imaging analysis, they indicated that "The percent of fluorescence intensity change relative to the initial intensity F_0 , $\Delta F/F_0 \times 100\%$, were plotted". How is F_0 defined? The intensity of the first image taken? If so, it is concerning to use a single image frame as the normalizing factor as it may result in large variations in their analyzed results. The authors should test alternative common methods for their analysis to see if the same conclusions hold.

5. In several cases, the authors did not discuss their calcium imaging results in the context of previous studies. They also did not seem to have considered important difference in experimental setups. For AIB and RIM, previous studies have shown that these neurons display spontaneous activity even when the worms are not stimulated with sensory cues (Gordus et al Cell 2015). However, these spontaneous activity patterns were not shown in this study. This difference needs to be explained. For AIZ, a previous study showed reversal-correlated activity of AIZ in the axon (Liu et al Cell 2014). But the

authors did not indicate whether they measured AIZ activity in the cell body or axon and this needs to be clarified. If they measured the cell body signals in AIZ, the rationale of doing so needs to be clarified. For RIA, previous studies showed that RIA axon displays compartmental head-bending correlated activity and global sensory activity in response to attractive odors (Hendricks et al Nature 2012; Jin et al Cell 2016). RIA activity in the cell body in response to sensory cues was not studied before (also, the authors did not indicate whether they measured RIA calcium activity in the cell body). If the authors measured RIA calcium response in the cell body, it is interesting that they examined this aspect of the RIA calcium activities, and the authors should include the contexts of previous studies when describing their results. In this study, all of the calcium imaging results from very different neurons (ASH, AIB, AIZ, RIA, RIM) showed a similar acute increase at cooling with a long and slow decrease of the signal. Perhaps, this is partly a result of gluing the worms for calcium imaging and starting the recording after a few minutes, which suppressed spontaneous activity or motor-associated activity in some of these neurons and made the cooling-induced sensory activity the primary response? In addition, the authors are analyzing the responses to a very fast cooling stimulus. Perhaps, this very strong stimulation triggers responses in these neurons that were not previously identified. These contexts and possibilities need to be included for readers to understand the results correctly.

6. The authors showed that activating RIM decreases reversal and Ω turn indices in response to fast-cooling and inhibiting RIM increases them. Meanwhile, previous studies showed that activating RIM with optogenetics increases reversals and turns. It is possible that the roles of RIM in regulating baseline reversals and turns vs sensory-cue triggered reversals and turns are opposite. The authors should test this possibility for a better understanding of their results by measuring the baseline level of reversals and turns in their transgenic lines that alter RIM activity or neurotransmission.

7. Figure 7 shows "calcium response probability". But the authors did not define this term.

8. It is interesting that the authors identified ASH as the sensing neuron for fast-cooling. However, they did not explore potential mechanisms for this sensory function of ASH. Does a worm sensing cooling using *ocr-2* or *osm-9* or any CNG channels?

9. The authors used mutants of *unc-7* and *unc-9* to test the involvement of gap junctions between ASH and AIZ in sensing fast cooling. However, these mutants disrupt gap junctions globally and provide weak evidence for the involvement of the gap junctions between ASH and AIZ. If the authors would like to conclude on this, they need to employ more selective methods, such as the one shown in Jang et al PNAS 2017, to test their hypothesis that "gap junctions mediate the stimulation of AIZ by ASH."

10. The comparison in some of the figures (or maybe the presentation. It is not clear because the authors did not describe it) is confusing. For example, in Fig4, the control and rescue should be compared with *glr-3* mutant, or with *eat-4* mutant. In Fig5, the control and rescues should be compared with *glr-1* mutant. In Fig7, the control and rescues should be compared with *avr-14* mutants. Similar analysis should be used for other figures involving similar type of statistical comparisons.

11. "To identify sensory neurons that are essential for this avoidance behavior, we employed the *Drosophila* histamine-gated chloride channel (HisCl1) plus 30 mM exogenous histamine to acutely silence amphid sensory neurons" This seems to be a very high concentration of histamine. Why was such high concentration of histamine needed?

Point-to-point response to the comments and suggestions of the reviewers

Responses to the reviewer #1:

In this article, Shan and collaborators provide an in-depth characterization of the neuronal circuit underlying cooling-evoked avoidance in crawling *C. elegans*. Cold-evoked behaviors are among the least characterized sensory-behaviors to date. Even in the powerful neurogenetic model *C. elegans*, we know very little. This article is timely and I am impressed by the huge number of technically-challenging experiments carried out to decipher the circuit underlying this response and the well-designed figures that efficiently present this complex dataset. I would like to congratulate the authors for this. The scientific outcome will be of interest to specialists and non-specialists in the field of neuroscience. Experiments are well-designed and most scientific claims are well supported by data. However, one important exception are the claims about ASH being a direct cooling-sensor, which either call for additional experiments or need to be rephrased as a hypothesis (in the latter case, without diminishing the impact of the work). In addition, several further controls are still needed in order to fully support the proposed models and there are major writing issues that need to be fully addressed before I can recommend publication of this article.

Response: We are very grateful to the reviewer for the positive comments on our work, the constructive suggestions and pointing out the errors in our manuscript.

Comment 1: In order to validate the physiological relevance of calcium data analyses, all the calcium imaging lines should be assessed for normal cooling-evoked behavior.

Response: We thank the reviewer for this constructive suggestion. To better understand and in accordance with the reviewer's suggestions, we have examined the cooling-evoked behaviors in all calcium imaging lines, including both the mutants and the transgene lines. These data are added in the figures in resubmitted manuscript, such as in Fig. 2, Fig. s15 and Fig. s19.

Comment 2: *unc-31* and *unc-13* experiments are great, but the conclusion is an overstatement, as neither mutant affect gap junctions. Actually, it is totally possible that ASH is not a primary cool sensor and receives electrical inputs from one of several other cells. RIC, RMG, AIZ and ADA all connect to ASH by gap junctions. Furthermore, considering the latest connectomic data by Witvliet et al. (2020, nemanode) together with more classic circuit description by White et al., one realizes that AIZ makes electrical connections with ASER, AWC and ASG (all three cool-responsive neurons), as well as AFD. Therefore, the ASER-AIZ-ASH or the AWC-AIZ-ASH connection could be at the base of the ASH cool-evoked response and ASH might not be directly the primary cooling sensor. Showing intact cooling-evoked ASH calcium activity in AWC-killed, ASE-killed, ASG-killed or (maybe even better) AIZ-killed backgrounds would be needed to ascertain the proposed model.

Response: We are extremely grateful to reviewer for pointing out this problem. According to the reviewer's comment, we have made additional experiments. We have tested ASH calcium transients induced by rapid cooling in AFD-inhibited, ASER-inhibited, ASG-inhibited (Fig. s4), AWC^{ON}-inhibited, AWC^{OFF}-inhibited and both AWC^{ON+OFF}-inhibited worms (Fig. s5), and AIZ-inhibited worms (Fig. s6). We found that ASH calcium activities in these neuron-inhibited worms were not significantly changed compared with the control worms. In addition, we examined ASH calcium activity in a lot of loss-of-function mutants, including *unc-13*, *unc-31*, *unc-7*, *unc-9*. The UNC-13 protein, encoded by the *unc-13* gene, is needed for the release of small and clear synaptic vesicles that carry small molecular neurotransmitters; the UNC-31 protein, encoded by the *unc-31* gene, is a critical component of the release machinery for neuropeptide-containing dense core vesicles; and both the UNC-7 and UNC-9 innexins, encoded by the *unc-7* and *unc-9* genes, respectively, are important structural components of gap junctions that affect locomotion. These null mutant worms are considered to exhibit weak information flow among neurons. The cooling-evoked calcium transients of ASH in these null mutant worms have no significant changes (Fig. 2a-c, e-f, Fig. s7). In addition, we checked ASH calcium activities in *osm-6* mutant worms, in which all the sensory neurons were silenced because of defects in functional

sensory cilia. We found the *osm-6* null mutant worms showed severe defects in cooling-evoked behaviors and calcium signals, and these defects could be partially rescued by expression of *osm-6* cDNA in ASH-specific neurons (Fig. 2d-f, Fig. s7). These results indicate that ASH can sense the cooling rate, at least rapid cooling. According your suggestion, we have toned down our conclusion and added this part in resubmitted manuscript.

Comment 3: *str-199* promoter was used for targeting AWC: Is this promoter really expressed in AWC, and if yes, is it both AWC^{ON} and AWC^{OFF} or only one of them? CENGEN data suggests the one cell the author see in the head could very well be ADL and not AWC. Providing further validation is essential, because I find premature to conclude that AWC is not mediating the cooling-evoked response based on this analysis with a poorly characterized promoter. Also, authors should test AWC^{ON} and AWC^{OFF} separately.

Response: Thank you very much for your suggestion. Definitely, AWC neurons are very important roles in temperature sensation¹⁻³. Thus, we rechecked the expression pattern of *str-199* promoter by using confocal microscopy, and indeed discovered a high probability of expression in other neurons besides AWC^{ON} or AWC^{OFF} neuron. We are very sorry for this carelessness.

To label AWC^{ON} and AWC^{OFF} specifically, we respectively use the promoter *str-2* to label AWC^{ON}, and use the promoter *srsx-3* to label AWC^{OFF}, and use both of them to label AWC^{ON+OFF} (Fig. s5a-f). We still found the promoter *srsx-3* have high expression in AWB beside AWC^{OFF} by confocal microscopy checking (data is not shown). We picked out the AWC^{OFF}-specific expression worms and used them to do experiments. We found that worms showed weak cooling-evoked reversal when both AWC^{ON} and AWC^{OFF} neurons were inhibited, whereas displayed no change in respective of AWC^{ON}-inhibited or AWC^{OFF}-inhibited worms (Fig. s2c-e, Fig. s3a-d). However, the rapid cooling-evoked calcium activities in ASH neurons have no change in both AWC^{ON+OFF}-inhibited worms (Fig. s5e-f). These results suggest that AWC neurons have a weak effect on regulation of cooling-evoked reversal but not regulate the ASH calcium

activity during rapid cooling. In addition, we found AWC neurons have obvious calcium transients in response to the rapid cooling stimulation (data not shown). These additional data were added in resubmitted manuscript.

Comment 4: The conclusion about *unc-7* and *unc-9* mutants are over-statements (Fig. 2e), because the authors have not done cell-specific rescue of either gene. It is not possible to confirm that the AIZ-ASH connection matters. It could well be the ASE-AIZ or the AWC-AIZ connections that are involved (in line with point 3 above) or any other electrical synapses elsewhere in the circuit, working indirectly.

Response: Thanks to your very constructive suggestions. To explore the gap junctions that function in stimulation of AIZ by ASH, we respectively PCR amplify *unc-7* and *unc-9* genomic DNA from wild-type worms, and express them in both ASH and AIZ neurons driven by ASH-specific promoter and AIZ-specific promoter, respectively (Fig. s9c). We examined the AIZ calcium transients induced by rapid cooling in these rescued worms, and found that defects of calcium activities in *unc-7* or *unc-9* mutant worms could be partially rescued by expression of *unc-7* or *unc-9* gDNA in both ASH and AIZ neurons (Fig. 3g). These rescued data suggest that AIZ, is at least partially stimulated by ASH neurons via gap junction during rapid cooling stimulation. We have added these data in resubmitted manuscript.

Comment 5: “Taken together, the behavioral and calcium results confirm that both AIZ and AIB neurons directly receive signal flow from ASH neurons to mediate rapid cooling stimulation” The data are compatible with this model, but alternative models are possible. Therefore, this statement must be toned down.

Response: Thanks very much for your constructive suggestions. Yes, it is a hasty conclusion. We have toned down this statement in the manuscript.

Comment 6: Capsaicin data (Fig. 2f, h, i) would be more convincing if a non-transgenic control lacking the TRPV1 transgenes and comparing -CAP and +CAP was shown.

Response: Thank you very much for pointing out our omitting control data. According

to your suggestion, we have tested calcium activities of interneurons AIZ or AIB in wild-type worms with or without application of CAP stimulation, and found that no significant changes between application of CAP and without application of CAP. We have added these control data in the Fig. 3f, 3i, Fig. 6e, and Fig. s10e in resubmitted manuscript. Due to the data of Fig. 2i in the previous version mainly tested the spontaneous locomotion behavior without cooling stimulation, we have removed this part in resubmitted manuscript.

Comment 7: Same thing for the other CAP data in Fig 3 and 4.

Response: Thanks for the suggestions. we carried out the control experiments according to your suggestion, and added the data in Fig. s10e and Fig. 6e in resubmitted manuscript.

Comment 8: The data for AIZ-specific *eat-4* RNAi, lack a proper control to demonstrate that the presence of a multi-copy transgene with AIZ-specific promoter is not per se impeding AIZ function.

Response: Thank you very much for pointing out the missing control experiments. To test the potential effect of the multi-copy transgene of AIZ on cooling-evoked avoidance behavior and RIA calcium signals, we used the expression of AIZ::GCaMP6f in wild-type worms as the control. Comparison of the control, the cooling-evoked avoidance behavior and RIA calcium signals in knocking down *eat-4* expression of AIZ-specific neurons in wild-type worms were significantly reduced (Fig. 5a-c). These results indicate that the glutamate, which released from AIZ, modulates the cooling-evoked avoidance behavior and RIA calcium signals. We have added these data in Fig. 5a-c of resubmitted manuscript.

Comment 9: It's quite surprising that the reversal and omega turn phenotypes are always similarly regulated in almost all the neuronal function impairing strains. Could authors further clarify if they always observed omega turn after reversal in this context? Alternatively, could they discuss how the same circuit controls both phenotypes?

Response: Yes, we indeed observed a high probability occurrence of omega turn that following the cooling-evoked reversal without interruption. This rapid cooling-evoked reversal coupled omega turn behaviors are the same as rapid heating-evoked avoidance behavior, where the worms also showed a high probability occurrence of omega turn following heating-evoked reversal³⁻⁶. In addition, the probability of occurrence of omega turn elicited by rapid cooling stimulation seems higher than that evoked by aversive stimuli, such as in Cu²⁺-evoked avoidance behavior in our previous work⁷⁶. We speculate this high probability of occurrence in omega turn that following reversal is because the constant low temperature stimulation after rapid cooling, to promote avoidance behavior.

Moreover, according to reviewer's suggestion, we defined a baseline reversal or turn occurred within 3-8 s in locomotion map without application of rapid cooling stimulation. After subtracted this baseline reversal or turn in cooling-evoked reversal or turn, we identified three parallel pathways which connected to each other during the rapid cooling. The pathways of ASH→AIZ→RIA and ASH→AVA seem to have more regulation in cooling-evoked reversal and turn (Fig. 4, Fig. 9), whereas the pathway of ASH→AIB| RIM more likely regulates the cooling-evoked reversal but not turn (Fig. 7). We have added these parts in Discussion in resubmitted manuscript.

Comment 10: AIB receives more synapse from AIZ compared to ASH (nemanode). Is it possible that ASH-AIZ-AIB pathway is more dominant compared to ASH-AIB pathway in this context? Could AIB still be activated (by cooling or chemogenetics) when AIZ is killed/inhibited?

Response: Thanks to your very constructive suggestions. AIB receives more synapses from AIZ compared to ASH. To examine whether AIB is stimulated by AIZ during the rapid cooling, we have examined AIB calcium transients in AIZ-inhibited worms (Fig. s3d in previous manuscript). We found that the magnitude of AIB calcium signals have no significant change during the rapid cooling. We speculate this probably the intensity of rapid cooling is strong enough to directly stimulate AIB to trigger the avoidance behavior, just like the avoidance behaviors induced by other noxious stimuli (e.g., nose

touch and osmotic shock⁸, 1-octanol⁹ and quinine stimulation¹⁰). We have added this data in Fig. s9e in resubmitted manuscript.

Comment 11: How does RIA induce reversal in this context, as it lacks outgoing synapse to previously known reversal promoting neurons?

Response: We sincerely appreciate the valuable comment. Indeed, according to the latest connectomic data¹¹⁻¹³ (nemanode) and previous works^{14,15}, RIA lacks outgoing synapse to activate reversal-correlated neurons. In our study, we found RIA mediates the rapid cooling-evoked reversal. Due to the widely presence of neuroendocrine modulation among the neurons during aversive stimuli^{7,16}, we speculate RIA neurons may stimulate reversal-correlated neurons during rapid cooling through neuroendocrine signals. The axon of RIA has rich calcium dynamics in response to the sensory input^{14,15}. To test whether the axon of RIA regulates the cooling-evoked avoidance behavior, we simultaneously examined the axon and cell body of RIA calcium activities when these are in the same focal plane. We found both of them could be stimulated by rapid cooling, and both of them showed the same calcium dynamics (Fig. 4d, Movie s6). These results suggest that the axon and cell body of RIA have similar function in regulation of cooling-evoked avoidance behavior. To explore which reversal-correlated neurons functionally act downstream of RIA to promote the cooling-evoked reversal, we tested AVA calcium activity in RIA-silenced worms. However, we did not observe significant change in AVA calcium activity after silencing RIA neurons during rapid cooling stimulation (Fig. s17a). These results suggest that RIA mediate the rapid cooling may through other neurons but not through AVA. We will do more studies about this question in the future. We have added these data in resubmitted manuscript.

Comment 12: ASH has considerable outgoing synapse to reversal promoting AVA and AVD neurons (nemanode). AVA plays crucial role in control of reversals (several previous studies). Therefore, it becomes very crucial to test the role of AVA in the present study. What is the role of AVA and AVD in the context of cold evoked reversals? Is it possible that in this context both AIZ and AIB circuit converge on AVA to control

reversals? Could authors monitor AVA calcium activity in AIB, RIA double HisCl strain? Could they compare reversal and omega turn phenotype by blocking AVA activity and comparing it to AIB, RIA double HisCl strain?

Response: Thanks for your suggestions. In our previous manuscript, we compared the cooling-evoked reversal between ASH-silenced and (RIA+AIB)-silenced worms (Fig. 9a-b in resubmitted manuscript). The results imply that there may exist other pathways mediate the cooling-evoked avoidance behaviors.

Previous works reported that command interneurons play an important role in control of reversal^{8,9,17,18}. Thus, according to your suggestions, we have made extensive experiments about the role of reversal-correlated command interneurons (AVA, AVD, and AVE) in response to rapid cooling. We tested the cooling-evoked avoidance behavior in these command interneuron-inhibited worms, and found that worms after inhibition of AVA or (AVA+AVD+AVA/AVE) neurons activities, showed serious defects in cooling-evoked reversal and turn (Fig. 9c-d). We also examined the cooling-evoked calcium activities of these command interneurons in ASH-inhibited worms, and found that command interneurons AVA, AVD, and AVA/AVE all could be stimulated by ASH (Fig. 9f-h). These behavioral and calcium results suggest that these command interneurons are involved in cooling-evoked avoidance behavior. Moreover, we compared the avoidance behavior between AVA-inhibited worms and (AVA+AVD+AVE/A)-inhibited worms, and found that no significant change between them (Fig. 9c-d). these results indicate that AVA is the primary command interneuron in response to rapid cooling stimulation.

To test whether AVA converges the stimulation signals from AIZ or AIB neurons during the rapid cooling stimulation, we tested AVA calcium transients in AIZ-silenced or AIB-silenced, or both (AIZ+AIB)-silenced worms. We found that AVA calcium activity in these interneurons-inhibited worms showed decreased, but still displayed obvious calcium response (Fig. 10a-b). Together the data in Fig. 9f, the command interneurons AVA could receive excitatory signals from interneurons of AIZ and AIB besides ASH during the rapid cooling. In addition, we tested AVA calcium activities in RIA-silenced or RIM-silenced worms, but observed no significant change in them (Fig.

s17a-b). These data suggest that RIA-mediated pathway and RIM-mediated pathway are not through AVA to modulate rapid cooling stimulation.

We then compared the avoidance behavior between ASH-silenced and (RIA+AIB+AVA)-silenced worms, and found that cooling-evoked reversal and turn between ASH-silenced and (RIA+AIB+AVA)-silenced worms have no significant differences (Fig. 10c-d), indicating that these pathways act downstream of ASH to mediate the rapid cooling. All these additional experiments were added in resubmitted manuscript.

Comment 13: Except for the results section which seems to have received more attention, the writing is poor overall and very poor in the Introduction and Method sections. There are many over-statements and sentences that make simply no sense. As a courtesy to the authors, despite the lack of line numbering, and because I was anyway getting over the manuscript in detail, I tried to list the most serious writing issues below, but additional improvements are needed everywhere along the manuscript. But honestly, this should have been done before submitting the manuscript.

Response: We are very sorry for our poor writing in our manuscript. We have made a thorough revision, toned down the conclusion and rewritten many paragraphs. We also invited a friend of us who is a native English speaker to help polish our resubmitted manuscript. We hope that the resubmitted manuscript with much more improvements meets the quality for publication.

Major writing concerns:

Concern 1: Summary: “The ability to sense and respond to ambient temperature change is the most critical function of the nervous system.” Overstatement with unjustified subjective judgment.

Response: We are very sorry for this overstatement writing. We have rephrased in the Summary in resubmitted manuscript.

Concern 2: Introduction: “Temperature is the most important environmental cue that

affects every biological and biochemical process.” Overstatement with unjustified subjective judgment.

Response: Sorry for this careless writing. We have rephrased.

Concern 3: Introduction: “Thus, the ability to sense temperature and respond promptly is critical for the survival of all organisms.” This might be true for animals but certainly not for all organisms.

Response: Sorry for this careless writing. We have rephrased.

Concern 4: Introduction: 2nd paragraph about threshold temperature is confusing. There is a well-established notion in the field about the threshold temperature for AFD responsiveness, which is modulated by past experience. Then the responsiveness of the other neurons (AWC, ASI, FLP, PHC) is either not well characterized or totally unrelated to the threshold in AFD. As it reads now, it is totally unclear as to why authors talk about THE threshold. They should talk about “A threshold” or “different thresholds” and better define it/them.

Response: We sincerely thanks for your professional suggestion. We have rewritten this part according to your suggestion. We try to use the value of absolute temperature, or temperature range, or temperature rate to describe the temperature stimulation in resubmitted manuscript.

Concern 5: Introduction: “Unlike thermosensation (thermotaxis and heat avoidance), which has been widely studied previously, cold sensation has increasingly been the focus of worm acclimation to environmental temperature in recent years in *C. elegans*.” This sentence makes no sense to me.

Response: Sorry for this careless writing. We have rephrased.

Concern 6: Introduction: “which means that neurons directly respond to cold stimuli, is still unknown” Syntax error

Response: We feel sorry for this careless writing. We have rephrased.

Concern 7: Introduction: “to regulate body turning in buffer water” what does this mean?

Response: We have made an extensive revision of the “Introduction” section, and rephrased this cold stimulation pattern.

Concern 8: Overall: the verb ‘adopt’ seems misused most of the time.

Response: Thank you for your reminder, we have rephrased in resubmitted manuscript.

Concern 9: To date, the locomotion behavior elicited by acute cold stimulation in a single worm is mainly using cold buffer water to trap the worm and evoke body turn locomotion” Syntax error

Response: Sorry for this mistake, we have rephrased.

Concern 10: “To test more behavioral states in a freely behaving worm underlying acute cold stimulation” what is a behavioral state for the author? Please define.

Response: Thanks for your suggestion, we have defined the behavioral state in resubmitted manuscript.

Concern 11: “was delivered to a forward sinusoidal locomotion worm” syntax error

Response: Sorry for this mistake, we have rephrased.

Concern 12: “We found that ASHs calcium activities showed a cooling rate-dependent way during the cooling range from 20.5 °C to 15 °C” What does ‘way’ mean?

Response: We are sorry for this mistake. We have rephrased in resubmitted manuscript.

Concern 13: “As previous studies have shown that ASHs are not required for cooling sensation” Citation needed.

Response: We are very sorry for this inaccurate description, and we have rephrased in resubmitted manuscript. Actually, there are only two references reporting that ASH

responds to slow heating^{2,3}, but we do not find the references about ASH responds to cooling or cold.

Concern 14: “Because TeTx has a weak effect on only gap junctions between ASH and AIZ neurons” Not clear. Is it an observation? An assumption? the syntax looks wrong.

Response: Tetanus (TeTx) has been reported to block neurotransmitter release by cleaving the synaptic vesicle protein synaptobrevin^{19,20}. It mainly affects the function of chemical synapse. In addition, we have found that TeTx has a weak effect on the gap junction between ASH and RIC during Cu²⁺ stimulation⁷. We are very sorry for this wrong syntax, and we have rephrased in resubmitted manuscript.

Concern 15: “spontaneous locomotion”: what does it mean? This must be defined.

Response: We are sorry for this inaccurate description. Actually, “Spontaneous locomotion” is the worm in a freely moving on the Thermotaxis assay plate without cooling stimulation. We have defined this in Supplementary materials and methods.

Concern 16: “RIM-inhibited worms displayed dramatically increased avoidance behavior” why using the word “dramatic”, when the effect is obviously mild?

Response: We are sorry for this misused. We have rephrased in resubmitted manuscript.

Concern 17: “The glutamate-gated chloride channel AVR-14 encodes inhibition of RIM by AIB” the term ‘encodes’ seems strange.

Response: We are sorry for this misused. We have rephrased in resubmitted manuscript.

Concern 18: In this study, we showed for the first time that *C. elegans* senses the cooling rate directly and avoids a rapid cooling stimulus (2.7 °C/s) within the physiological temperature range from 15 °C to 25 °C. Further studies demonstrated for the first time that polymodal ASH sensory neurons sense rapid cooling and regulate cooling-evoked avoidance behavior” All these ‘first time’ statements should be avoided.

In addition, the part about ASH being a cooling sensor is, at best, totally premature.

Response: Thanks to your constructive suggestions. This concern is similar to the comment 2. According to your suggestion, we have made additional experiments, including testing ASH neurons calcium transients in AFD-, ASER-, ASG-, AWC^{ON}-, AWC^{OFF}-, AWC^{ON+OFF}- and AIZ-inhibited worms (Fig. s4, Fig. s5, Fig. s6), and in loss-of-function mutants of *unc-13*, *unc-31*, *unc-7*, *unc-9*, *osm-6*, and in *osm-6*; ASH::*osm-6* genetically rescued worms (Fig. 2a-f, Fig. s7). All of these results indicate that ASH neurons can sense the cooling, at least rapid cooling. We also thank you very much for your suggestion about writing. We have toned down this conclusion, removed all the “first time”, and re-written this part in resubmitted manuscript.

Concern 19: “Our data also showed probabilistic calcium activity in RIM during rapid cooling stimulation” I would not say that. I would say “Our data also showed probabilistic modulation of calcium activity in RIM during rapid cooling stimulation”

Response: Thanks for your suggestion. We agree with you, and have re-written this part in the resubmitted manuscript.

Concern 20: “However, worms still showed a proportional avoidance behavior when the activities of both the stimulatory circuit and disinhibitory circuit were silenced (RIA::HisC11 + AIB::HisC11)” proportional to what? This is not clear.

Response: We are sorry for this careless. To avoid misleading, we have removed “proportional avoidance behavior” in resubmitted manuscript.

Concern 21: “It has been confirmed that AVA command interneurons that directly receive stimulation signals from ASHs to regulate various aversive stimuli that evoke avoidance behavior.” This looks like a chopped sentence.

Response: Sorry for this writing. We have rephased in resubmitted manuscript.

Concern 22: “In this study, we find that ASH neurons can sense the cooling rate directly; thus, future work should also try to identify cooling rate sensors.” Again,

evidence for ASH being a primary cooling sensor is insufficient to claim this.

Response: This concern is the same as comment 2 and concern 18. We have made additional experiments to confirm this conclusion. These experiments including testing ASH neurons calcium transients in AFD-, ASER-, ASG-, AWC^{ON-}, AWC^{OFF-}, AWC^{ON+OFF-} and AIZ-inhibited (Fig. s4, Fig. s5, Fig. s6), and in loss-of-function mutants of *unc-13*, *unc-31*, *unc-7*, *unc-9*, *osm-6*, and in *osm-6*; ASH::*osm-6* genetically rescued worms (Fig. 2). All these data suggest that sensory neurons ASH can sense rapid cooling. These data were added in resubmitted manuscript.

Moreover, we have made additional experiments to explore potential molecules that mediate ASH response to rapid cooling stimulation. These tested molecules included the TRPV channel protein OSM-9 and OCR-2, the TRPA channel protein TRPA-1, and the possible G protein-coupled receptor (GPCR) activation of G α protein ODR-3 and the G γ protein GPC-1. We found that rapid cooling-evoked calcium transients and avoidance behaviors in *osm-9* and *odr-3* were significantly reduced. These results suggest that OSM-9 and the unknown GPCR may mediate the rapid cooling in ASH neurons. These results were added in Fig. s19a-j in resubmitted manuscript.

Concern 23: “L4 worms with expression of HisC11 were first picked on a thin OP50 lawn with 30 mM histamine” Please explain how a thin lawn is defined/produced.

Response: We are sorry for this inaccurate description. We used a 200 μ L of *E. coli* OP50 (shaking 16 h at 37 °C) with a final concentration 30 mM histamine, to seed on whole surface of NGM plate (60 mm diameter). These NGM plate were dried at 37 °C for 48 h with lid on before spreading. These prepared histamine plates were harvested the next day, and stored in dark place at 4 °C. The OP50 lawn generated by this method is thinner compared to the plate of maintaining worms, because the growth of OP50 at the room temperature were stopped. We have re-written this part in resubmitted Supplementary materials and methods.

Concern 24: “By finely modulating Ttx plate thickness and the Peltier elements

cooling power through a programmable power, we realized the cooling rate at 0 °C/s and constant temperature at (20.5 ± 0.008) °C, cooling rate at (0.4 ± 0.002) °C/s and baseline temperature at (15.0 ± 0.03) °C, cooling rate at (0.7 ± 0.004) °C/s and the baseline temperature at (15 ± 0.03) °C, cooling rate at (1.4 ± 0.007) °C/s and the baseline temperature at (15.0 ± 0.04) °C, cooling rate at (2.7 ± 0.02) °C/s and the baseline temperature at (15.0 ± 0.03) °C, cooling rate at (2.7 ± 0.03) °C/s and the baseline temperature at (17.0 ± 0.03) °C, cooling rate at (2.7 ± 0.02) °C/s and the baseline temperature at (20.0 ± 0.03) °C.” What does the +/- values represent, how were they calculated, and were the recordings so precise that a precision of 0.002 °C/s could be claimed. Hard to believe.

Response: We are very sorry for this inaccurate description. The “±” represents standard error range. Actually, the range of variation at cooling rate 0.4 °C/s is ~0.36-0.44 °C/s. Due to the sample size is large ($n \geq 50$), the standard error looks like very small. To avoid misunderstanding, we have rephased this part in resubmitted Supplementary materials and methods. The temperature traces were also added in raw data file.

Concern 24: “Usually five well-fed day1, cultivated at 20.5 °C worms were washed in M9 buffer with or without exogenous histamine (30 mM, for silencing worm), and then picked onto the assay plate to rest at starting temperature 20.5 °C or 25.5 °C for at least 2 min respectively” Why using the term ‘Usually’? What was the unusual protocol and when was it applied?

Response: We are really sorry for this inaccurate description. In most cases, we picked five worms to assay, but to some neuron-manipulated worms, which have higher motor activities that easily crawl out examination area, we would pick more than five worms (about 7 or 8 worms) to assay. We have rephased this part in resubmitted Supplementary materials and methods.

Concern 25: More details are needed to understand what was done with Image-Pro Plus 6.0?

Response: Thanks for your suggestion. Image-Pro Plus 6.0 has powerful analytical functions in tracking moving object besides image processing, including analyzing velocity and angle, body length, fluorescence intensity of range of interest (ROI) etc. These behavioral analytical functions are the same in wormLab software. Actually, in our previous work^{7,21}, we use MATLAB (Mathworks, Inc., Natick, MA) to assay the locomotion behaviors (reversal, turn, the magnitude of body bend etc.), and the calcium imaging. But to some students who have no basic of programming, the Image-Pro Plus 6.0 is suitable. We have added this part in the Supplementary materials and methods.

Concern 26: “To test the worms suffered the same stimulation in specific cooling rate and the baseline temperature across experiments” unclear.

Response: We are sorry for this writing, and rephased in resubmitted Supplementary materials and methods.

Concern 27: “Therefore, these statistical results confirm the different phenotypes in this study are inherent characteristics of the worms, rather than effected by any systematic variations” wrong syntax

Response: We are sorry for this writing, and rephased in resubmitted Supplementary materials and methods.

Concern 28: Likewise, references for the choice of promoters are needed.

Response: Thank you very much for your suggestion, we have added the references for the choice of promoters in Supplementary materials and methods.

Lastly, Special thanks to you for your good comments again.

Responses to the reviewer #2:

In this manuscript, the authors identified new neural circuits and molecules that sense

temperature rapid changes and regulate aversive behavior in *C. elegans*. *C. elegans* grow between 15 degrees to 25 degrees, and some sensory neurons such as AFD, AWC, OLL, PVC, and PHC that sense temperature stimulus have been known, and they are located all over the body, including the head, body, and tail. The most well-known AFD is known as a sensory neuron that responds very sensitively to temperature changes and can remember the experienced temperature to control behavioral plasticity.

The authors showed that the ASH circuit, which is different from the AFD neural circuit, controls two kinds of downstream neural circuits, thereby facilitating the aversive behavior efficiently against rapid temperature changes. In this process, various techniques were used, such as quantitative behavioral analysis, neural activity, neural manipulation, and genetic analysis. In addition, each experimental result had high reproducibility, and through neuronal manipulation of specific neurons, the function and information processing between neurons involved in aversive behavior were well understood. On the other hand, some ambiguous information needs to be clarified, and the current manuscript is missing scientifically intriguing points that should be brought up based on their experimental results.

Response: We thank the reviewer #2 sincerely for the nice and the positive comments on our study, the constructive suggestions and pointing out the errors and deficiencies in our manuscript.

Comment 1: The authors analyzed the behavioral change of worms in response to rapid cooling rates of 0, 0.4, 0.7, 1.4, and 2.7 degree/s, demonstrating that they exhibited reversal or turn behavior, particularly for temperature changes of 1.4 degree/s or more. As can be seen in Supplement Video 2, this aversive behavior occurred at the end of the temperature change. Do the temperature stimuli shown in each Figure represent the actual temperature the worm sense, or are they the temperature values set in the temperature control device they used? If it is the latter case, there is no problem, but if not, it is necessary to clarify whether ASH senses a temperature rapid change or the point at which temperature changes stop. The results also indicate similar neural activity

in response to temperature stimuli, in addition to the observed behavioral patterns.

Response: We sincerely appreciate the valuable comments. The temperature stimuli shown in each Figure represent the temperature values that set by the programmable temperature control device. Due to the temperature stimulation is not like the aversive stimuli (Cu^{2+} , quinine, SDS etc.), that can accurately deliver to the worms' nose by a dry drop test when they were crawling on the plate⁷, the rapid cooling-evoked behavior in Supplement Video 2 occurred at the end of the temperature change. Actually, there are a lot of reversal occurred during cooling (~3-7 s) when stimulated with relatively slow rate of 1.4 °C/s (Fig. s1d). Similarly, there are many ASH calcium transients occurred during the time ~5-8 s elicited by the cooling rate 1.4 °C/s (Fig. s1k). These results suggest that worms respond to the rapid temperature change.

Comment 2: The author argues that the sensory information of temperature rapid change in ASH induces avoidance behavior through two circuits, stimulatory and disinhibitory circuits. In Fig. 8 they examine the relationship between these two circuits and demonstrate that they synergistically contribute to avoidance behavior. However, it is necessary to examine whether there is any change in the neural activity of the other pathway when the information processing of one pathway is blocked. For example, RIM, which represents a probability response of about 50% in the wild type, shows a further decrease in calcium response probability in AIZ-silenced worm or *glr-3; glr-6* mutant, as shown in Fig. 6q, or whether there is no change. Other cases are conceivable, of course.

Response: Thanks to your constructive suggestions. Actually, in previous manuscript of Fig. s4b-f, we have tested RIM calcium activity in AIZ-inhibited worms during the rapid cooling, and found that RIM calcium activity has no change during the rapid cooling. We have added these data in Fig. s10d in resubmitted manuscript. Based on your and other two reviewer's suggestions, we have made additional experiments: (1) We identified that AVA is the primary command interneuron in response to rapid cooling stimulation among command interneurons AVA, AVD, and AVE/AVA (Fig. 9). (2) AVA converges the stimulation signals from AIZ or AIB neurons, but not RIA or RIM during

the rapid cooling stimulation (Fig. 10a-b, Fig. s17a-b). (3) (RIA+AIB+AVA)-mediated pathways act downstream of ASH to regulate the rapid cooling (Fig. 10c-e). All these additional experiments were added in resubmitted manuscript.

Comment 3: The meaning of “control” in all graphs is ambiguous. For example, does “control” in Fig 1j-k refer to wild type? It is necessary to clarify and define what the control is for HisCl1, TeTx, and PHminiSOG in Fig. 2b-c. Their controls cannot be the same, and at least in experiments using Neuron::HisCl1, the control should be without the addition of histamine. The same applies to other graphs. The capsaicin experiment with neurons expressing TRPV1 makes sense.

Response: Thank you very much for your careful checks. Indeed, the control in our manuscript is ambiguous. We are very sorry for this. The control of HisCl1, TeTx, and PHminiSOG manipulation are the worms with expression of HisCl1 in neuron without application of histamine, the wild-type worms, and the worms with expression of PHminiSOG in neuron without periodic blue light illumination, respectively. We have added the information of control in the Figures in resubmitted manuscript.

Comment 4: The rapid cooling stimulus that the author focuses on, is performed under the temperature range (from 15 to 25 degrees) in which the worms can survive. If the ASH circuit is known to be a circuit that is promoted by harmful stimuli for survival, then it is necessary to discuss the relationship between the growth environment and why rapid cooling (2.7degree/s) should promote aversive behavior. So, in addition to the current paradigm, further discussion is needed on understanding animal behavior regulation, which includes internal and external states in response to temperature stimulus.

Response: We thank the reviewer very much for this very nice comment. Worm is ectotherms, and temperature is ubiquitous stimulus that regulates the body temperature and the internal rate of biochemical reaction. It is thus imperative for animals to sense temperature changes. Worms live within innocuous temperature range of ~13-26 °C, and temperature below ~13 °C leads to worm paralysis, and are thus considerer harmful

stimuli. In natural world, when worms exposed the rapid cooling stimulation, it means temperature is easily reaches the harmful temperature (~13 °C). Thus, accurately sensing and responding to temperature change is important for animal survival. To survival, worms employ avoidance behavior to respond to rapid cooling, including using the reversal to avoid and the turn to change the direction of motion. We added this part in Discussion in revised manuscript.

Minor concerns:

Concern 1: Kainite in 12 lines on 4 pages is misspelled. Correct is kainate.

Response: We feel sorry for our carelessness. In our resubmitted manuscript, the typo is revised. Thanks for your correction.

Concern 2: Clarify the wiring diagram information referenced in Figures 2A, 3A, and 6A.

Response: Thanks for your suggestion. We added the diagram information in the Figure legend in resubmitted manuscript.

Concern 3: ASH sensory neurons trigger reversals and omega turns in response to many aversive stimuli. Has there been any testing to see if the calcium transients and the sensory adaptation in ASH, *osm-9* and *gpc-1*, are involved in aversive behavior triggered by rapid cooling?

Response: Thanks to your constructive suggestions. We have made additional experiments to explore potential molecules that mediate ASH response to rapid cooling stimulation. These tested molecules included the TRPV channel protein OSM-9 and OCR-2, the TRPA channel protein TRPA-1, and the possible G protein-coupled receptor (GPCR) activation of G α protein ODR-3 and the G γ protein GPC-1. We found that the rapid cooling-evoked calcium transients and avoidance behaviors in *osm-9* and *odr-3* mutant worms were significantly reduced. These results suggest that OSM-9 and the unknown GPCR may mediate the rapid cooling in ASH neurons. These results were added in Fig. s19a-j in resubmitted manuscript.

Concern 4: ASHs have been shown to sense rapid cooling and induce avoidance behavior. If rapid cooling similar to the detection of a noxious stimulus, is associated with aversive behavior mediated by ASH, consider the possibility of ASH circuit involvement in response to a rapid increase in temperature within the physiological temperature range.

Response: Thanks for your suggestion. To test whether ASH responds to the rapid heating, we changed our semiconductor cooling surface to heating surface. We realized an approximate linear heating rate 2.5 °C/s in range of 20 °C to 25.5 °C. By using this rapid heating stimulation, we found that ASH showed obvious calcium activity (Fig. s18a), which is consistent with previous reports that ASH can be stimulated during slow increasing of temperature^{2,3}. These results suggest that ASH can respond to the rapid heating. However, the magnitude of calcium activity in ASH induced by rapid heating is lower than rapid cooling (Fig. s18a). We speculate this difference between them may due to the ASH calcium activity was modulated by other temperature sensory neurons, such as AFD, AWC, ASI etc., because these temperature sensory neurons could be positively stimulated by temperature increasing, but not be activated by temperature decreasing. Future study should clarify this difference of calcium signal between cooling-evoked and heating-evoked. We added this part in resubmitted manuscript.

Concern 5: In relation to Ca²⁺ imaging figures, please explain in materials and methods the peak delta F/F₀, which is used to compare neural activity between animals, is specifically calculated for what time interval.

Response: We sincerely thank the reviewer for careful reading. We have re-written this part in Materials and methods. We used the following method to analyze the calcium imaging. The average fluorescence intensity of region of interest (ROI) of the soma and axon was captured and analyzed, and the adjacent ROI in each frame was used to subtract the background. The average fluorescence intensity within the initial 5 s before cooling stimulation was taken as the basal signal, F_0 . The change in fluorescence intensity relative to the initial intensity F_0 , $\Delta F/F_0 = (F - F_0) / F_0$, was plotted as a

function of time for all curves by using IGOR Pro 6.12 (Wavemetrics, USA). The mean value of calcium signal and the SEM were plotted in various colors as indicated and in light grey, respectively. The maximum of $\Delta F/F_0$ during 5 - 50 s was defined as the peak $\Delta F/F_0$, and the peak $\Delta F/F_0$ for each trace higher than 0.05 was defined as a calcium response. The peak $\Delta F/F_0$ was used to statistical analysis.

Concern 6: The authors ablated the sensory or interneuron using various ways. According to Gray's study in 2005, these ablated worms, especially RIA- or RIM-ablated worms, were found to have an effect on reversal behavior. We are concerned whether the behavioral deficits due to neuron inhibition had no impact on the behavioral analysis during rapid cooling.

Response: We sincerely appreciate your valuable comment. Indeed, the neuron-ablated worms have a baseline reversal and turn change in exploratory behaviors²²⁻²⁴. It is thus worth noting that, the behavioral deficits of neuron-inhibited worms elicited by rapid cooling are whether due to the baseline behaviors changed, or the modulation of cooling signal changed, or both of them effect. To reduce the impact of baseline behavior, we have made some efforts. (1) In selection of assay worms, we select the worm which exhibiting an obvious tendency of forward movement. After 3 seconds of forward movement, we applicate the cooling stimulus to the worm. (2) In definition of cooling-evoked reversal or turn, we define a cooling-evoked reversal as the worm stopping forward movement and initiating a reversal with at least half a head swing within 5 seconds of stimulation initiation (duration of 3-8 seconds in the locomotion map), whereas a cooling-evoked turn is defined as any of the following: One is stopping forward movement and initiating a turn within 5 s period, and the second is an omega turn following the cooling-evoked reversal without interruption. The method of selecting assay worms and the definition of cooling-evoked behavior significantly reduced the effect of baseline behaviors. These parts have been added in resubmitted manuscript.

Even so, there are still some baseline reversal and turn were scored in our previous manuscript, such as in Fig. s1a (without cooling), there are some of reversal (~10%

ratio) or turn (~8% ratio) occur within 3-8 s period. To eliminate this baseline reversal and turn, and according to the reviewers' suggestion, we have made additional experiments. We thawed the strains that are related to this study, and tested the baseline reversal and turn of these worms under the same assay condition as cooling but without application of cooling stimulation. We first recorded locomotion behavior of an initiating forward movement worm at least 8 seconds without application of cooling stimulation, and a reversal or turn that occurred within 3-8 s is defined a baseline reversal or baseline turn. In addition, an omega turn following the baseline reversal without interruption is also defined a baseline turn. We then calculated the reversal or turn index by the ratio of reversal or turn that elicited by cooling minus ratio of baseline reversal or turn without cooling, respectively. By using this method, we eliminate the effects of baseline reversal and turn in our study. The data of baseline reversal and turn were added in Supplementary figures of resubmitted manuscript.

Concern 7: In relation to page 10, line 3, put the statistics difference between *eat-4* and AIZ::HisC11 in the graph of Fig 4C.

Response: Thank you very much for pointing out this problem. We have added the results of statistical analysis in the Fig. 5a-c in the resubmitted manuscript.

Concern 8: To aid in understanding, it would be helpful to move ASH::TRPV1 located at the top of the graph in Fig 5f, to the bottom of the graph, similar to the way it is presented in Supplementary Figure 6C.

Response: Thanks for suggestion. We have revised this in Fig. 6f in resubmitted manuscript.

Thank you again for your comments and suggestions.

Responses to the reviewer #3:

In this manuscript, the authors discovered a new sensory response in *C. elegans*,

showing that the worms respond to fast cooling with avoidance behavior. To characterize how the avoidance of cooling is regulated, they identified that the sensory neuron ASH detects cooling stimulation, and two downstream circuits positively regulate the behavioral response together. Both of these circuits utilize glutamate to transmit signals with one engaging excitatory *glr-3* and *glr-6* receptors and the other using *avr-14*-mediated inhibitory signals. The study is valuable in identifying and characterizing a new behavioral response to a common sensory cue. The analysis on the regulatory circuit and molecules are comprehensive. However, there are concerns about data analysis and presentation, and the results are often not adequately discussed. These issues must be addressed.

Response: We are grateful to the reviewer for the positive comments for our work, very hortative and constructive suggestions and pointing out the errors and deficiencies in our manuscript.

Comment 1: The authors used “reversal index” and “ Ω turn index” to quantify fast cooling-induced avoidance, but they did not explain how these indices are calculated. This information is critical because in many of their experiments where the activity or neurotransmission of specific neurons are altered, baseline reversals and Ω turns are likely altered as a result. For example, it was shown that activating AIB triggers reversals (Gordus et al Cell 2015) and AIZ is needed for initiating reversals (Liu et al Cell 2014). Inactivating AIB or AIZ likely changes reversal rate or turning rate even when the worms are not stimulated. Similarly, they disrupted both AIZ and AIB circuits in Fig. 8 using either HisCl1 or *glr-1;glr-3;glr-6* triple mutants. These genetic changes will also alter baseline reversals and turns. These baseline effects need to be considered, for example, by measuring reversal rate without cooling and using it as a baseline to calculate cooling induced changes in reversals and turns.

Response: We are extremely grateful to reviewer for pointing out this problem. This comment is similar to the reviewer #2 concern 6. Indeed, the neuron-ablated worms have a baseline reversal and turn change in exploratory behaviors²²⁻²⁴. It is thus worth noting that, the behavioral deficits of neuron-inhibited worms elicited by rapid cooling

are whether due to the baseline behaviors changed, or the modulation of cooling signal changed, or both of them effect. To reduce the impact of baseline behavior, we have made some efforts: (1) In selection of assay worms, we selected the worm which exhibiting an obvious tendency of forward movement. After 3 seconds of forward movement, we applicated the cooling stimulus to the worm. (2) In definition of cooling-evoked reversal or turn, we defined a cooling-evoked reversal as the worm stopping forward movement and initiating a reversal with at least half a head swing within 5 seconds of stimulation initiation (duration of 3-8 seconds in the locomotion map), whereas a cooling-evoked turn was defined as any of the following: One is stopping forward movement and initiating a turn within 5 s period, and the second is an omega turn following the cooling-evoked reversal without interruption. The method of selecting assay worms and the defining of cooling-evoked behavior significantly reduced the effect of baseline behaviors. These parts have been added in Supplementary materials and methods and resubmitted manuscript.

Even so, there are still some baseline reversal and turn were scored in our previous manuscript, such as in Fig. s1a (without cooling), there are some of reversal (~10% ratio) or turn (~8% ratio) occurs within 3-8 s period. To eliminate this baseline reversal and turn, and according to the reviewers' suggestion, we have made additional experiments. We thawed the strains that are related to this study, and tested the baseline reversal and turn of these worms under the same assay condition as cooling but without application of cooling stimulation. We first recorded locomotion behavior of an initiating forward movement worm at least 8 seconds without cooling stimulation, and a reversal or turn that occurred within 3-8 s was defined a baseline reversal or baseline turn. In addition, an omega turn following the baseline reversal without interruption was also defined a baseline turn. We then calculated the reversal or turn index by the ratio of reversal or turn that elicited by cooling minus ratio of baseline reversal or turn without cooling, respectively. By using this method, we eliminate the effects of baseline reversal and turn in this study. The data of baseline reversal and turn were added in Supplementary figures of resubmitted manuscript.

Comment 2: They used chemical-genetic experiments to activate several neurons in the circuit with capsaicin in order to examine the effects on downstream neurons and behavior. These include interneurons such as AIB, AIZ and RIM that do not have cilia or opening to the outside. How does CAP access these neurons? Is the expression of TRPV1 in these interneurons specific and is it possible that some sensory neurons actually express TRPV1 in these transgenic lines?

Response: Thanks for your very constructive suggestions. Indeed, the capsaicin is liposoluble that do not act on the non-cilia interneurons directly, that's why we often found no calcium response in interneurons during the experiments. In addition, we inspected the expression of TRPV1 in AIZ-specific or AIB-specific transgenic worms by using confocal microscopy, and found very weak expression in some sensory neurons. We are very sorry for this inaccuracy. Thus, we used optogenetics method to stimulate those interneurons calcium activity by the help of Dr Li, who is skilled in this area²³. We added these optogenetics results in Fig. 4f, Fig. 7h in resubmitted manuscript.

Comment 3: The authors chose 4 downstream interneurons (AIA, AIB, AIY and AIZ) to characterize in this study. However, AIZ also send lots of synapses to AVE. ASH also strongly synapse onto AVA, AVB and form gap junctions with RIC, RMG, ADA. The rationale of their choices needs to be clarified. Additionally, the authors discussed that “Thus, future studies should clarify whether AVA neurons or other neurons modulate avoidance behavior during rapid cooling stimulation.” Many previous studies have shown that AVA acts as an important downstream neuron of ASH to mediate worm behavior. The analysis of AVA in this fast-cooling response should be included in this study.

Response: We thank the reviewer for this constructive suggestion. This comment is similar to the reviewer #1 comment 12. In our previous manuscript, we compared the cooling-evoked reversal between ASH-silenced and (RIA+AIB)-silenced worms (Fig. 9a-b in resubmitted manuscript). The results imply that there may exist other pathways mediate the cooling-evoked avoidance behaviors.

Previous works reported the command interneurons play important role in control

of reversal^{8,9,17,18}. Thus, according to your and the reviewer #1 suggestions, we have made extensive experiments about the role of reversal-correlated command interneurons (AVA, AVD, and AVE) in response to rapid cooling. We tested the cooling-evoked avoidance behaviors in these command interneurons-inhibited worms, and found the worms which inhibition of AVA or (AVA+AVD+AVA/AVE) neurons activities, showed serious defects in cooling-evoked reversal and turn (Fig. 9c-d). we also examined the cooling-evoked calcium activities in these command interneurons, and found the AVA, AVD, and AVA/AVE could be stimulated by ASH during the cooling stimulation (Fig. 9f-h). These behavioral and calcium results suggest these command interneurons are involved in cooling-evoked avoidance behavior. Moreover, we compared the avoidance behaviors between AVA-inhibited and (AVA+AVD+AVE)-inhibited worms, and found that there were no significant differences between them (Fig. 9c-d). These results indicate that AVA is the primary command interneuron in response to rapid cooling stimulation.

To test whether AVA converges the stimulation signals from AIZ or AIB neurons during the rapid cooling stimulation, we tested AVA calcium transients in AIZ-silenced or AIB-silenced, or both (AIZ+AIB)-silenced worms. We found that AVA showed a decrease of cooling-evoked calcium activity in these interneurons-inhibited worms, but still displayed obvious calcium signal (Fig. 10a-b). Together the data in Fig. 9f, the command interneurons AVA could receive excitatory signals from interneurons of AIZ and AIB besides ASH during the rapid cooling. In addition, we tested AVA calcium activities in RIA-silenced or RIM-silenced worms, but observed no significant change in them (Fig. s17a-b). These data suggest that RIA-mediated pathway and RIM-mediated pathway are not through AVA to modulate rapid cooling stimulation.

We then compared the avoidance behavior between ASH-silenced and (RIA+AIB+AVA)-silenced worms, and found that cooling-evoked reversal and turn between ASH-silenced and (RIA+AIB+AVA)-silenced worms have no significant differences (Fig. 10c-d), indicating that these pathways act downstream of ASH to mediate the rapid cooling.

Moreover, we tested the other interneuron calcium activities, including the RIC

and RMG, which have direct synaptic connections with ASH, and found that the calcium activities of these interneurons were not changed in ASH-silenced worms. These results indicating that these interneurons are not involved in rapid cooling-evoked avoidance behavior (Fig. s16c-f). All these additional experiments were added in resubmitted manuscript.

Comment 4: In their calcium imaging analysis, they indicated that “The percent of fluorescence intensity change relative to the initial intensity F_0 , $\Delta F/F_0 \times 100\%$, were plotted”. How is F_0 defined? The intensity of the first image taken? If so, it is concerning to use a single image frame as the normalizing factor as it may result in large variations in their analyzed results. The authors should test alternative common methods for their analysis to see if the same conclusions hold.

Response: We are sorry to this unclear expression, and this comment is similar to reviewer #2 concern 5. Actually, in this study together with our previous work^{7,21,25}, we used the following assay method to analyze the calcium imaging. The average fluorescence intensity of region of interest (ROI) of the soma and axon was captured and analyzed, and the adjacent ROI in each frame was used to subtract the background. The average fluorescence intensity within the initial 5 s before cooling stimulation was taken as the basal signal, F_0 . The change in fluorescence intensity relative to the initial intensity F_0 , $\Delta F/F_0 = (F - F_0) / F_0$, was plotted as a function of time for all curves by using IGOR Pro 6.12 (Wavemetrics, USA). The mean value of calcium signal and the SEM were plotted in various colors as indicated and in light grey, respectively. The maximum of $\Delta F/F_0$ during 5-50 s was defined as the peak $\Delta F/F_0$, and the peak $\Delta F/F_0$ for each trace higher than 0.05 was defined as a calcium response. The peak $\Delta F/F_0$ was used for statistical analysis. We have re-written this part in Materials and methods in resubmitted manuscript.

Comment 5: In several cases, the authors did not discuss their calcium imaging results in the context of previous studies. They also did not seem to have considered important difference in experimental setups. For AIB and RIM, previous studies have shown that

these neurons display spontaneous activity even when the worms are not stimulated with sensory cues (Gordus et al Cell 2015). However, these spontaneous activity patterns were not shown in this study. This difference needs to be explained (1). For AIZ, a previous study showed reversal-correlated activity of AIZ in the axon (Li et al Cell 2014). But the authors did not indicate whether they measured AIZ activity in the cell body or axon and this needs to be clarified. If they measured the cell body signals in AIZ, the rationale of doing so needs to be clarified (2). For RIA, previous studies showed that RIA axon displays compartmental head-bending correlated activity and global sensory activity in response to attractive odors (Hendricks et al Nature 2012; Jin et al Cell 2016). RIA activity in the cell body in response to sensory cues was not studied before (also, the authors did not indicate whether they measured RIA calcium activity in the cell body). If the authors measured RIA calcium response in the cell body, it is interesting that they examined this aspect of the RIA calcium activities, and the authors should include the contexts of previous studies when describing their results (3). In this study, all of the calcium imaging results from very different neurons (ASH, AIB, AIZ, RIA, RIM) showed a similar acute increase at cooling with a long and slow decrease of the signal. Perhaps, this is partly a result of gluing the worms for calcium imaging and starting the recording after a few minutes, which suppressed spontaneous activity or motor-associated activity in some of these neurons and made the cooling-induced sensory activity the primary response? In addition, the authors are analyzing the responses to a very fast cooling stimulus. Perhaps, this very strong stimulation triggers responses in these neurons that were not previously identified (4). These contexts and possibilities need to be included for readers to understand the results correctly.

Response: we feel great thanks for your professional review work on our article. We have numbered your questions in above comment.

Answer question 1: This comment is similar to comment 1. According to your and reviewer #2 suggestions, we have made additional experiments. We thawed the strains that are related to this study, and recorded locomotion behavior of an initiating forward movement worm at least 8 seconds under the same assay condition as cooling but without application of cooling stimulation. A reversal or turn that occurred within 3-8 s

was defined a baseline reversal or baseline turn. In addition, an omega turn following the baseline reversal without interruption was also defined a baseline turn. Indeed, by using this method to test baseline behavior, some neuron-inhibited worms showed the spontaneous reversal or turn decrease, such as the manipulation of worms in Fig. s3a-b, Fig. s8a-b, Fig. s10a-b, Fig. s16a-b, Fig. s17c-d. Meanwhile, some neuron-inhibited worms showed spontaneous reversal or turn increase, such as in RIM-inhibited worms in Fig. s14a-b.

Thus, to eliminate the effects of baseline reversal and turn in this study, the spontaneous activities (baseline reversal or baseline turn) were used to subtract the background. Then the results after subtracting of background were used to inspect. The data of baseline reversal and turn were added in Supplementary figures of resubmitted manuscript.

Answer question 2: Indeed, the axon of AIZ has a reversal-correlated calcium activity in previous reports^{23,26}. We thus tested the axon of AIZ calcium transients during the rapid cooling and found that the axon showed obvious calcium activity (Fig. 3d, Movie s5). These results suggest that axon of AIZ could be stimulated by rapid cooling. Moreover, we compared the calcium activities between axon and cell body when these are in the same focal plane, and found that the magnitude of calcium signals in cell body is stronger than in axon during the rapid cooling (Fig. 3d, Movie s5). We thus use the soma of AIZ calcium activity for the following research. We added this part in the resubmitted manuscript.

Answer question 3: RIA interneurons have compartmentalized axonal calcium dynamics that integrate sensory input and motor feedback^{14,15}. To explore whether axon of RIA responds to rapid cooling, we examined the calcium activity in axon during the rapid cooling, and found robust calcium transients in axon of RIA. Moreover, we compared the cooling-evoked calcium activities between axon and cell body of RIA when they were in the same focal plane, and found that both showed similar calcium activity during rapid cooling (Fig. 4d, Movie s6). These results suggest that the soma

and axon of RIA have same modulation in response to rapid cooling, which are different from previous reports¹⁴. These differences imply that RIA may exist new response pattern to rapid cooling. We have added these data and discussion in resubmitted manuscript.

Answer question 4: Thank you once again for your suggestion. The patterns of calcium signals in tested neurons are really simple during the rapid cooling stimulation, which displayed acute increase at cooling and followed with a long and slow decrease. This cooling-evoked calcium response pattern of ASH is similar to the aversive stimuli-elicited calcium response pattern that observed in gluing worm in vivo calcium imaging²⁷. However, this cooling-elicited calcium response pattern is very different from our previous observation in microfluidic chip, where ASH showed robust calcium transient at initiation of Cu^{2+} stimulation and sometimes continued increasing during the Cu^{2+} stimulation^{7,21}, or where the interneurons usually showed non-uniform calcium response pattern^{7,16}. Moreover, this rapid cooling-evoked calcium response pattern is also different from observation in gluing worms calcium imaging with slow rising temperature stimulation, where the sensory neurons AWC, ASH, ASI, and the interneurons RIB, AVA, SMD showed stochastic calcium events^{1,3,28}. We speculate these different response patterns among them may due to the intensity of the rapid cooling is stronger than slow increasing in temperature that activates the unknown molecules and circuits, or the different of stimulation duration, or the method of gluing worm calcium imaging. We have added this part in **Discussion** in resubmitted manuscript.

Comment 6: The authors showed that activating RIM decreases reversal and Ω turn indices in response to fast-cooling and inhibiting RIM increases them. Meanwhile, previous studies showed that activating RIM with optogenetics increases reversals and turns. It is possible that the roles of RIM in regulating baseline reversals and turns vs sensory-cue triggered reversals and turns are opposite. The authors should test this possibility for a better understanding of their results by measuring the baseline level of

reversals and turns in their transgenic lines that alter RIM activity or neurotransmission.

Response: Thank you for your suggestion. This comment is similar to the comment 1. According to your suggestion, we have made additional experiments to test the effect of baseline reversal and turn on cooling-evoked avoidance behavior. Indeed, we found the baseline reversal in RIM-inhibited worms was increased without rapid cooling stimulation (Fig. s14a). However, after subtraction of this baseline reversal, the cooling-evoked reversal were still increased after inhibition of RIM activity (Fig. 7b). These data suggest that RIM inhibition of reversal during the rapid cooling. This difference between our study and previous studies may due to RIM has rich structural connections with other neurons. We have added these additional data in resubmitted manuscript.

Comment 7: Figure 7 shows “calcium response probability”. But the authors did not define this term.

Response: Thank you for your suggestion. In testing RIM calcium activity induced by rapid cooling, we found RIM showed the probabilistic calcium response events, that was different from ASH displaying stable calcium activity during the rapid cooling. Thus, we defined a calcium event of RIM based on the activation of ASH neurons during rapid cooling (Fig. 7d). To avoid misunderstanding, we only use the peak $\Delta F/F_0$ to assay results. We have added Fig. 7d in resubmitted manuscript.

Comment 8: It is interesting that the authors identified ASH as the sensing neuron for fast-cooling. However, they did not explore potential mechanisms for this sensory function of ASH. Does a worm sensing cooling by using *ocr-2* or *osm-9* or any CNG channels?

Response: We thank the reviewer for this constructive suggestion. According your and reviewer #2 suggestions, we have made additional experiments to tentatively explore the rapid cooling sensor in ASH. We tested some molecules including the TRPV channel protein OSM-9, OCR-2, the TRPA channel protein TRPA-1, and the possible G protein coupled receptor (GPCR) activation of G α protein ODR-3 and G γ protein GPC-1. We found the OSM-9 and the unknown GPCR may act as rapid cooling sensors,

because they all showed obvious defects in rapid cooling-evoked calcium activity and avoidance behavior (Fig. s19a-j). This topic should be investigated in our next study. We have added these data in Fig. s19a-i in resubmitted manuscript.

Comment 9: The authors used mutants of *unc-7* and *unc-9* to test the involvement of gap junctions between ASH and AIZ in sensing fast cooling. However, these mutants disrupt gap junctions globally and provide weak evidence for the involvement of the gap junctions between ASH and AIZ. If the authors would like to conclude on this, they need to employ more selective methods, such as the one shown in Jang et al PNAS 2017, to test their hypothesis that "gap junctions mediate the stimulation of AIZ by ASH."

Response: We thank the reviewer for pointing this out, and thanks for providing the improvement. According to your suggestion, we have made additional experiments to confirm gap junction mediates the stimulation of AIZ by ASH. We first amplified the genes of *unc-7* and *unc-9*, and inserted into the expression vector with ASH or AIZ promoter, respectively. We found the defects of calcium activity in *unc-7* or *unc-9* mutant worms could be partially rescued by specific expression of *unc-7* or *unc-9* genomic DNA in both ASH and AIZ neurons (Fig. s9c, Fig. 3g). These results suggest gap junction, at least partially mediate the activation of AIZ by ASH during rapid cooling stimulation. We have added these data in Fig. s9c and Fig. 3g in resubmitted manuscript.

Comment 10: The comparison in some of the figures (or maybe the presentation. It is not clear because the authors did not describe it) is confusing. For example, in Fig4, the control and rescue should be compared with *glr-3* mutant, or with *eat-4* mutant. In Fig5, the control and rescues should be compared with *glr-1* mutant. In Fig7, the control and rescues should be compared with *avr-14* mutants. Similar analysis should be used for other figures involving similar type of statistical comparisons.

Response: We thank you very much for pointing out these problems. The control that used to statistical analysis is really wrong. We are sorry for this. We have re-marked the

statistical results in the Fig. 5, Fig. 6, Fig. 8 in resubmitted manuscript.

Comment 11: “To identify sensory neurons that are essential for this avoidance behavior, we employed the *Drosophila* histamine-gated chloride channel (HisC11) plus 30 mM exogenous histamine to acutely silence amphid sensory neurons” This seems to be a very high concentration of histamine. Why was such high concentration of histamine needed?

Response: Thank you for your question. Indeed, the normal concentration of histamine that used to silence neuron activity is 10 mM²⁹. Based on the results in published paper²⁹, worms need at least 5 min to reach 100% paralysis. In our study, we mainly used histamine plus HisC11 receptor to silence neuron, and used the gluing worm method to test the calcium signals. To avoid osmotic stimulation induced by water evaporates in gluing worm method, we must finish the experiment within 4 min (quiescence 2 min, recording 1 min). Thus, to paralyze the worm as soon as possible, we raised concentration of histamine. In addition, in our study, we used control that without adding histamine to test the experiment results. We added this part in Materials and methods.

Lastly, we thank reviewers once again for their good comments, constructive suggestions, and pointing out the errors and deficiencies. These comments and concerns are really valuable in improving the quality of our manuscript.

References

- 1 Biron, D., Wasserman, S., Thomas, J. H., Samuel, A. D. & Sengupta, P. An olfactory neuron responds stochastically to temperature and modulates *Caenorhabditis elegans* thermotactic behavior. *Proc Natl Acad Sci U S A* **105**, 11002-11007, doi:10.1073/pnas.0805004105 (2008).
- 2 Kuhara, A. *et al.* Temperature sensing by an olfactory neuron in a circuit controlling behavior of *C. elegans*. *Science* **320**, 803-807, doi:10.1126/science.1148922 (2008).
- 3 Kotera, I. *et al.* Pan-neuronal screening in *Caenorhabditis elegans* reveals asymmetric dynamics of AWC neurons is critical for thermal avoidance behavior. *Elife* **5**, doi:ARTN e1902110.7554/eLife.19021 (2016).
- 4 Ghosh, R., Mohammadi, A., Kruglyak, L. & Ryu, W. S. Multiparameter behavioral

- profiling reveals distinct thermal response regimes in *Caenorhabditis elegans*. *BMC Biol* **10**, 85, doi:10.1186/1741-7007-10-85 (2012).
- 5 Mohammadi, A., Rodgers, J. B., Kotera, I. & Ryu, W. S. Behavioral response of *Caenorhabditis elegans* to localized thermal stimuli. *Bmc Neurosci* **14**, doi:Artn 6610.1186/1471-2202-14-66 (2013).
- 6 Byrne Rodgers, J. & Ryu, W. S. Targeted thermal stimulation and high-content phenotyping reveal that the *C. elegans* escape response integrates current behavioral state and past experience. *PLoS One* **15**, e0229399, doi:10.1371/journal.pone.0229399 (2020).
- 7 Guo, M. *et al.* Reciprocal inhibition between sensory ASH and ASI neurons modulates nociception and avoidance in *Caenorhabditis elegans*. *Nat Commun* **6**, doi:ARTN 565510.1038/ncomms6655 (2015).
- 8 Piggott, B. J., Liu, J., Feng, Z. Y., Wescott, S. A. & Xu, X. Z. S. The neural circuits and synaptic mechanisms underlying motor initiation in *C. elegans*. *Cell* **147**, 922-933, doi:10.1016/j.cell.2011.08.053 (2011).
- 9 Summers, P. J. *et al.* Multiple sensory inputs are extensively integrated to modulate nociception in *C. elegans*. *J Neurosci* **35**, 10331-10342, doi:10.1523/JNEUROSCI.0225-15.2015 (2015).
- 10 Zou, W. J. *et al.* Decoding the intensity of sensory input by two glutamate receptors in one *C. elegans* interneuron. *Nat Commun* **9**, doi:ARTN 431110.1038/s41467-018-06819-5 (2018).
- 11 White, J. G., Southgate, E., Thomson, J. N. & Brenner, S. The structure of the nervous system of the nematode *Caenorhabditis elegans*. *Philos Trans R Soc Lond B Biol Sci* **314**, 1-340, doi:10.1098/rstb.1986.0056 (1986).
- 12 Cook, S. J. *et al.* Whole-animal connectomes of both *Caenorhabditis elegans* sexes. *Nature* **571**, 63, doi:10.1038/s41586-019-1352-7 (2019).
- 13 Witvliet, D. *et al.* Connectomes across development reveal principles of brain maturation. *Nature* **596**, 257, doi:10.1038/s41586-021-03778-8 (2021).
- 14 Hendricks, M., Ha, H., Maffey, N. & Zhang, Y. Compartmentalized calcium dynamics in a *C. elegans* interneuron encode head movement. *Nature* **487**, 99-103, doi:10.1038/nature11081 (2012).
- 15 Jin, X., Pokala, N. & Bargmann, C. I. Distinct circuits for the formation and retrieval of an imprinted olfactory memory. *Cell* **164**, 632-643, doi:10.1016/j.cell.2016.01.007 (2016).
- 16 Li, Z. Y. *et al.* Dissecting a central flip-flop circuit that integrates contradictory sensory cues in *C. elegans* feeding regulation. *Nat Commun* **3**, doi:ARTN 77610.1038/ncomms1780 (2012).
- 17 Chalfie, M. *et al.* The neural circuit for touch sensitivity in *Caenorhabditis elegans*. *J Neurosci* **5**, 956-964, doi:10.1523/JNEUROSCI.05-04-00956.1985 (1985).
- 18 Guo, Z. V., Hart, A. C. & Ramanathan, S. Optical interrogation of neural circuits in *Caenorhabditis elegans*. *Nat Methods* **6**, 891-896, doi:10.1038/nmeth.1397 (2009).
- 19 Schiavo, G. *et al.* Tetanus and botulinum-B neurotoxins block neurotransmitter release by proteolytic cleavage of synaptobrevin. *Nature* **359**, 832-835, doi:10.1038/359832a0 (1992).
- 20 Macosko, E. Z. *et al.* A hub-and-spoke circuit drives pheromone attraction and social behaviour in *C. elegans*. *Nature* **458**, 1171-U1110, doi:10.1038/nature07886 (2009).
- 21 Guo, M. *et al.* Dissecting molecular and circuit mechanisms for inhibition and delayed

- response of ASI neurons during nociceptive stimulus. *Cell Rep* **25**, 1885-1897 e1889, doi:10.1016/j.celrep.2018.10.065 (2018).
- 22 Gray, J. M., Hill, J. J. & Bargmann, C. I. A circuit for navigation in *Caenorhabditis elegans*. *Proc Natl Acad Sci USA* **102**, 3184-3191, doi:10.1073/pnas.0409009101 (2005).
- 23 Li, Z. Y., Liu, J., Zheng, M. H. & Xu, X. Z. S. Encoding of both analog- and digital-like behavioral outputs by one *C. elegans* interneuron. *Cell* **159**, 751-765, doi:10.1016/j.cell.2014.09.056 (2014).
- 24 Gordus, A., Pokala, N., Levy, S., Flavell, S. W. & Bargmann, C. I. Feedback from network states generates variability in a probabilistic olfactory circuit. *Cell* **161**, 215-227, doi:10.1016/j.cell.2015.02.018 (2015).
- 25 Liu, H. *et al.* Reciprocal modulation of 5-HT and octopamine regulates pumping via feedforward and feedback circuits in *C. elegans*. *Proc Natl Acad Sci U S A* **116**, 7107-7112, doi:10.1073/pnas.1819261116 (2019).
- 26 Chalasani, S. H. *et al.* Dissecting a circuit for olfactory behaviour in *Caenorhabditis elegans*. *Nature* **450**, 63-70, doi:10.1038/nature06292 (2007).
- 27 Hilliard, M. A. *et al.* In vivo imaging of *C. elegans* ASH neurons: cellular response and adaptation to chemical repellents. *EMBO J* **24**, 63-72, doi:10.1038/sj.emboj.7600493 (2005).
- 28 Beverly, M., Anbil, S. & Sengupta, P. Degeneracy and neuromodulation among thermosensory neurons contribute to robust thermosensory behaviors in *Caenorhabditis elegans*. *Journal of Neuroscience* **31**, 11718-11727, doi:10.1523/Jneurosci.1098-11.2011 (2011).
- 29 Pokala, N., Liu, Q., Gordus, A. & Bargmann, C. I. Inducible and titratable silencing of *Caenorhabditis elegans* neurons in vivo with histamine-gated chloride channels. *Proc Natl Acad Sci U S A* **111**, 2770-2775, doi:10.1073/pnas.1400615111 (2014).

REVIEWER COMMENTS

Reviewer #1 (Remarks to the Author):

The authors have satisfactorily addressed all my concerns, with an extensively revised manuscript including a lot of new supporting data. I congratulate the authors for this terrific study on cool-evoked avoidance in the worm that will be highly influential in the field. The level of mechanistic detail gained in a single study on a so far largely under-studied sensory modality is impressive.

Reviewer #2 (Remarks to the Author):

The authors provided sufficient motivation for this research on rapid cooling-avoidance behavior in the revised manuscript. This study showed that rapid cooling-avoidance behavior is triggered by distinct neural circuits, different from those known for rapid-heating avoidance (AFD, AWC, FLP, PVC, and PHA sensory circuits).

Additionally, compared to the previous script, the explanation of the experimental method, result, and discussion is more specific, and additional experiments were conducted to persuade the readers. The analysis of the AVA interneuron network, including the AIZ and AIB interneurons related to Figures 9 and 10 is highly assessed. These intricate circuits in response to rapid-cooling stimuli, as mentioned in the discussion, play a crucial role in inducing animals to quickly avoid harmful stimuli, ultimately contributing to their survival.

However, there are still parts that remain unclear or ambiguous, so it would be beneficial if they were revised.

To Authors

Minor comments

1. Page 9, "Furthermore, the calcium signal defects in *unc-7* or *unc-9* mutants could be partially rescued by respective expression of *unc-7* or *unc-9* genomic DNA specific in both ASH and AIZ neurons (Fig. s9c, Fig. 3g). "

The meaning is ambiguous. Are the authors referring to the cDNA of *unc-7* or *unc-9*?

2. Page 17, "Based on the structural map and previous reports^{10,41,60}, "

The meaning is ambiguous. Neural connectome?

Reviewer #3 (Remarks to the Author):

In the revised manuscript, the authors have provided results from new experiments and added discussions on various comments raised by the previous reviews. These additions have further strengthened the findings of the original study and improved the clarity of the paper. However, there are a few places where either the authors did not understand the review comments correctly or some clarification is needed.

1. The authors did not correctly understand question 1 in comment 5 ("Comment 5:For AIB and RIM, previous studies have shown that these neurons display spontaneous activity even when the worms are not stimulated with sensory cues (Gordus et al Cell 2015). However, these spontaneous activity patterns were not shown in this study. This difference needs to be explained (1)"). This comment raised questions on the calcium activity patterns of several neurons, AIB, AVA, RIM, that are characterized in this study and previous studies. These neurons display spontaneous activity even when the worms are exposed to buffer. However these spontaneous activity patterns were not found

in this study, as shown in the heatmaps of Figure 3h, 7e-m, 8c, 8e and others. The authors provided baseline reversals and turns for this comment, which is not what the review asked for. Some discussions are needed to address the comment.

2. In Discussion, the authors state that "In our study, we found that the cell body of RIA has the same calcium activity as the axon (Fig. 4d, Movie s6)." This statement should be more specific for this study and indicate that "In our study, we found that the cell body of RIA has the same calcium activity as the axon in response to rapid cooling". In addition, in authors' Point-to-point Response to the comment for question 3 in comment 5, the authors indicated that "To explore whether axon of RIA responds to rapid cooling, we examined the calcium activity in axon during the rapid cooling, and found robust calcium transients in axon of RIA. Moreover, we compared the cooling-evoked calcium activities between axon and cell body of RIA when they were in the same focal plane, and found that both showed similar calcium activity during rapid cooling. These results suggest that the soma and axon of RIA have same modulation in response to rapid cooling, which are different from previous reports¹⁴. These differences imply that RIA may exist new response pattern to rapid cooling..." I would respectfully point out that it is not correct to say that the findings in this study on RIA are different from previous reports because previous reports did not examine the calcium response of RIA cell body in response to rapid cooling. But we agree with the authors that their findings suggest that RIA may display new response patterns to rapid cooling.

3. To address the function of ASH, the authors indicated that "As expected, if the cilia of ASH exhibit functional deficiencies, the cooling-elicited calcium transients in ASH and the avoidance behaviors of worms showed severe defects (Fig. 2d-f, Fig. s7a-b)," Did the authors imply that they examined the defects in ASH cilia? If yes, they need to clarify how the experiment was done and where they included the results. But, if this is not what the authors suggested, they need to clarify this point in their writing.

4. The writing of the manuscript needs to be improved. For example, "Sensory neuron ASHs respond to the cooling rate and exhibit rapid cooling evoked avoidance behavior." - it is not the sensory neurons ASH that exhibit avoidance behavior. Another example "We first examined the ASH calcium transients in these neuron-inhibited worms by expression of HisCl1 plus exogenous histamine or expression of PHminiSOG with blue light illumination." This sentence needs to be improved by re-writing.

Point-by-point response to the comments and suggestions of the reviewers

Responses to the reviewer #1:

The authors have satisfactorily addressed all my concerns, with an extensively revised manuscript including a lot of new supporting data. I congratulate the authors for this terrific study on cool-evoked avoidance in the worm that will be highly influential in the field. The level of mechanistic detail gained in a single study on a so far largely under-studied sensory modality is impressive.

Response: We are very happy to receive your recognition and encouragement for our work, and thank you again for your valuable comments on our manuscript, which helped us a lot.

Responses to the reviewer #2:

The authors provided sufficient motivation for this research on rapid cooling-avoidance behavior in the revised manuscript. This study showed that rapid cooling-avoidance behavior is triggered by distinct neural circuits, different from those known for rapid-heating avoidance (AFD, AWC, FLP, PVC, and PHA sensory circuits).

Additionally, compared to the previous script, the explanation of the experimental method, result, and discussion is more specific, and additional experiments were conducted to persuade the readers. The analysis of the AVA interneuron network, including the AIZ and AIB interneurons related to Figures 9 and 10 is highly assessed. These intricate circuits in response to rapid-cooling stimuli, as mentioned in the discussion, play a crucial role in inducing animals to quickly avoid harmful stimuli, ultimately contributing to their survival.

However, there are still parts that remain unclear or ambiguous, so it would be beneficial if they were revised.

Response: We are grateful to the reviewer's positive evaluation of our work and

pointing out the deficiencies in our work.

Minor comments

1. Page 9, “Furthermore, the calcium signal defects in *unc-7* or *unc-9* mutants could be partially rescued by respective expression of *unc-7* or *unc-9* genomic DNA specific in both ASH and AIZ neurons (Fig. s9c, Fig. 3g).” The meaning is ambiguous. Are the authors referring to the cDNA of *unc-7* or *unc-9*?

Response: Thanks for your nice comment. We have rephased this sentence in resubmitted manuscript. In this study, the genes of *unc-7* (5753bp) or *unc-9* (3383 bp) were PCR amplified from N2 genomic DNA library but not N2 cDNA, which including the intron of genomic.

2. Page 17, “Based on the structural map and previous reports^{10,41,60}”, The meaning is ambiguous. Neural connectome?

Response: We are sorry for this ambiguous writing. Actually, the structural map and previous reports are related to the neurons that have structural connection with the command AVA interneurons. We have rephased this sentence in our revised version.

Responses to the reviewer #3:

In the revised manuscript, the authors have provided results from new experiments and added discussions on various comments raised by the previous reviews. These additions have further strengthened the findings of the original study and improved the clarity of the paper. However, there are a few places where either the authors did not understand the review comments correctly or some clarification is needed.

Response: We thank the reviewer #3 sincerely for the positive comments on our study, the constructive suggestions and pointing out the deficiencies in the revised manuscript.

1. The authors did not correctly understand question 1 in comment 5 (“Comment 5: ...For AIB and RIM, previous studies have shown that these neurons display spontaneous activity even when the worms are not stimulated with sensory cues (Gordus et al Cell 2015). However, these spontaneous activity patterns were not shown in this study. This difference needs to be explained (1)”). This comment raised questions on the calcium activity patterns of several neurons, AIB, AVA, RIM, that are characterized in this study and previous studies. These neurons display spontaneous activity even when the worms are exposed to buffer. However, these spontaneous activity patterns were not found in this study, as shown in the heatmaps of Figure 3h, 7e-m, 8c, 8e and others. The authors provided baseline reversals and turns for this comment, which is not what the review asked for. Some discussions are needed to address the comment.

Response: We are extremely sorry for not understanding your comment correctly. Indeed, interneurons AIB, RIM and AVA, and sensory neuron AWC have a highly spontaneous calcium activity when the worms are not stimulated with isoamyl alcohol (IAA)¹. To explore whether these interneurons have the same spontaneous calcium activity in our study, we tested these interneurons calcium transients under the same recording condition but did not application of rapid cooling stimulation. We found that interneurons AIB, RIM and AVA showed almost no spontaneous calcium activity (Fig. s18a-c), which are totally different from in olfactory circuit. Due to the interneurons AIB, RIM and AVA can receive the inputs from sensory neurons either AWC or ASH²⁻⁴, and AWC neurons showed a highly calcium activities without odor stimulation in olfactory circuit¹, whereas in our study, ASH displayed no spontaneous calcium response without cooling stimulation (Fig. 11, Fig. s18d). We thus speculate that these differences occur because the different of spontaneous calcium activities between the sensory neuron ASH and AWC. Under the higher activity inputs from AWC, interneurons AIB, RIM and AVA showed obvious spontaneous calcium activity, while under the lower activity inputs from ASH in cooling sensation, interneurons AIB, RIM

and AVA displayed almost no spontaneous calcium response. We have added these data in Fig. s18a-e and Discussion in resubmitted manuscript.

2. In Discussion, the authors state that “In our study, we found that the cell body of RIA has the same calcium activity as the axon (Fig. 4d, Movie s6).” This statement should be more specific for this study and indicate that “In our study, we found that the cell body of RIA has the same calcium activity as the axon in response to rapid cooling”. In addition, in authors’ Point-to-point Response to the comment for question 3 in comment 5, the authors indicated that “To explore whether axon of RIA responds to rapid cooling, we examined the calcium activity in axon during the rapid cooling, and found robust calcium transients in axon of RIA. Moreover, we compared the cooling-evoked calcium activities between axon and cell body of RIA when they were in the same focal plane, and found that both showed similar calcium activity during rapid cooling. These results suggest that the soma and axon of RIA have same modulation in response to rapid cooling, which are different from previous reports¹⁴. These differences imply that RIA may exist new response pattern to rapid cooling...” I would respectfully point out that it is not correct to say that the findings in this study on RIA are different from previous reports because previous reports did not examine the calcium response of RIA cell body in response to rapid cooling. But we agree with the authors that their findings suggest that RIA may display new response patterns to rapid cooling.

Response: We are sorry for our inaccurate expression in previous Revised Manuscript and Point-to-point Response, and thank you very much for pointing out this error. According to your suggestion, we have added “in response to rapid cooling” at the end of sentence “In our study, we found that the cell body of RIA has the same calcium activity as the axon” to make RIA response more specific in this study.

3. To address the function of ASH, the authors indicated that “As expected, if the cilia of ASH exhibit functional deficiencies, the cooling-elicited calcium transients in ASH

and the avoidance behaviors of worms showed severe defects (Fig. 2d-f, Fig. s7a-b),” Did the authors imply that they examined the defects in ASH cilia? If yes, they need to clarify how the experiment was done and where they included the results. But, if this is not what the authors suggested, they need to clarify this point in their writing.

Response: Thanks for your nice comment. Actually, we didn’t examine the defects in ASH cilia. To test whether OSM-6 is expressed in ASH, we examined expression pattern of *osm-6* and found OSM-6 is expressed in ASH. We have added this results in Fig. s7c. In addition, according to your suggestion, we have rephrased this point in our revised manuscript with colour highlighting.

4. The writing of the manuscript needs to be improved. For example, “Sensory neuron ASHs respond to the cooling rate and exhibit rapid cooling evoked avoidance behavior.” - it is not the sensory neurons ASH that exhibit avoidance behavior. Another example “We first examined the ASH calcium transients in these neuron-inhibited worms by expression of HisC11 plus exogenous histamine or expression of PHminiSOG with blue light illumination.” This sentence needs to be improved by re-writing.

Response: We are extremely grateful to you for pointing out this problem. We have rewritten these sentences with colour highlighting in the revised manuscript.

- 1 Gordus, A., Pokala, N., Levy, S., Flavell, S. W. & Bargmann, C. I. Feedback from network states generates variability in a probabilistic olfactory circuit. *Cell* **161**, 215-227, doi:10.1016/j.cell.2015.02.018 (2015).
- 2 White, J. G., Southgate, E., Thomson, J. N. & Brenner, S. The structure of the nervous system of the nematode *Caenorhabditis elegans*. *Philos Trans R Soc Lond B Biol Sci* **314**, 1-340, doi:10.1098/rstb.1986.0056 (1986).
- 3 Gray, J. M., Hill, J. J. & Bargmann, C. I. A circuit for navigation in *Caenorhabditis elegans*. *P Natl Acad Sci USA* **102**, 3184-3191, doi:10.1073/pnas.0409009101 (2005).
- 4 Cook, S. J. *et al.* Whole-animal connectomes of both *Caenorhabditis elegans* sexes. *Nature* **571**, 63, doi:10.1038/s41586-019-1352-7 (2019).

REVIEWERS' COMMENTS

Reviewer #2 (Remarks to the Author):

The authors provided clear explanations for the ambiguities I pointed out.

Reviewer #3 (Remarks to the Author):

The authors addressed all the comments in the revised manuscript.